# Acidity suppresses CD8 + T-cell function by perturbing IL-2, mTORC1, and c-Myc signaling

Romain Vuillefroy de Silly [ID] ✉, Laetitia Pericou [ID], Bili Seijo, Isaac Crespo [ID] & Melita Irving [ID] ✉

## Abstract

CD8 + T cells have critical roles in tumor control, but a range of factors in their microenvironment such as low pH can suppress their function. Here, we demonstrate that acidity restricts T-cell expansion mainly through impairing IL-2 responsiveness, lowers cytokine secretion upon re-activation, and reduces the cytolytic capacity of CD8 + T cells expressing low-affinity TCR. We further find decreased mTORC1 signaling activity and c-Myc levels at low pH. Mechanistically, nuclear/cytoplasmic acidification is linked to mTORC1 suppression in a Rheb-, Akt/TSC2/PRAS40-, GATOR1- and Lkb1/AMPK-independent manner, while c-Myc levels drop due to both decreased transcription and higher levels of proteasome-mediated degradation. In addition, lower intracellular levels of glutamine, glutamate, and aspartate, as well as elevated proline levels are observed with no apparent impact on mTORC1 signaling or c-Myc levels. Overall, we suggest that, due to the broad impact of acidity on CD8 + T cells, multiple interventions will be required to restore T-cell function unless intracellular pH is effectively controlled.

**Keywords** Acidity; CD8+ T cell; mTOR; c-Myc; IL-2
**Subject Categories** Immunology; Signal Transduction

## Introduction

Patient responses to cancer immunotherapies typically rely upon the re-invigoration of cytolytic T lymphocytes (CTLs) within a hostile tumor microenvironment (TME) (Lanitis et al, 2017). Along with powerful immune checkpoints such as the PD-1/PD-L1 axis (Postow et al, 2015) and the presence of potently suppressive soluble factors like TGF-β, adenosine (Vigano et al, 2019) and PGE-2, the inherent physicochemical properties of the TME including nutrient starvation, hypoxia, and acidity, represent major barriers to T-cell function (Chang et al, 2015; Huber et al, 2017; Salmon et al, 2012; Vuillefroy de Silly et al, 2016). Extracellular acidification is a common property of tumors. This is due to the reliance of tumor cells upon aerobic glycolysis (i.e., the Warburg effect) (Tannock and Rotin, 1989; Warburg, 1956) to meet their metabolic needs leading to coupled efflux of lactate and protons, as well the tendency of tumor cells to over-express enzymes like carbonic anhydrase (Swietach et al, 2007). Interestingly, it has recently been shown that lymph nodes comprise acidic regions in which T cells themselves contribute to acidification of the extracellular milieu due to enhanced aerobic glycolysis upon activation (Wu et al, 2020).

It is well known that low pH can cause profound dysfunction of many immune-cell types in the TME including effector CD8+ T cells (Huber et al, 2017; Lardner, 2001; Tannock and Rotin, 1989; Taylor, 1962; Wu et al, 2020), but the impact of acidity on cellular signaling and metabolism has not been comprehensively characterized in CTLs. Elucidating mechanism(s) of action of low pH on T cells may contribute towards the development of more effective combinatorial immunotherapies and/or gene-engineering strategies (Irving et al, 2017) that can overcome this common barrier to T-cell activity in the TME. This notion is supported by the demonstration of improved tumor control in some pre-clinical models upon modulation of acidity by administration of proton pump inhibitors (Calcinotto et al, 2012) and bicarbonate (Pilon-Thomas et al, 2016).

Here, we have comprehensively evaluated the impact of pH on effector CD8+ T-cell function upon re-activation in vitro, including of expansion/proliferation as this is a major obstacle to most T-cell centric immunotherapies (Dudley and Rosenberg, 2003). We identified defects in IL-2 responsiveness under acidic conditions and subsequently dissected how pH influences IL-2R signaling, the mTORC1 pathway, c-Myc activity, and metabolomic profiles. Overall, at low pH we reveal important changes to T-cell effector function and to IL-2 responsiveness, including intracellular signaling and transcription factor levels, and amino acid metabolism. Together, these findings implicate low pH as a major barrier to the physiological function of CD8+ T-cells and as a critical target for improving responses to cancer immunotherapy.

## Results

### Low pH impairs the function of effector CD8+ T cells upon re-activation

Within the TME anti-tumor CD8+ T cells are for the most part fully differentiated into an effector state (Fig. EV1A) (Dudley and Rosenberg, 2003). Hence, in order to explore the impact of low pH

---

Ludwig Institute for Cancer Research, University of Lausanne and Department of Oncology, Lausanne University Hospital (CHUV), Lausanne, Switzerland.
✉E-mail: Romain.VuillefroydeSilly@unil.ch; Melita.Irving@unil.ch

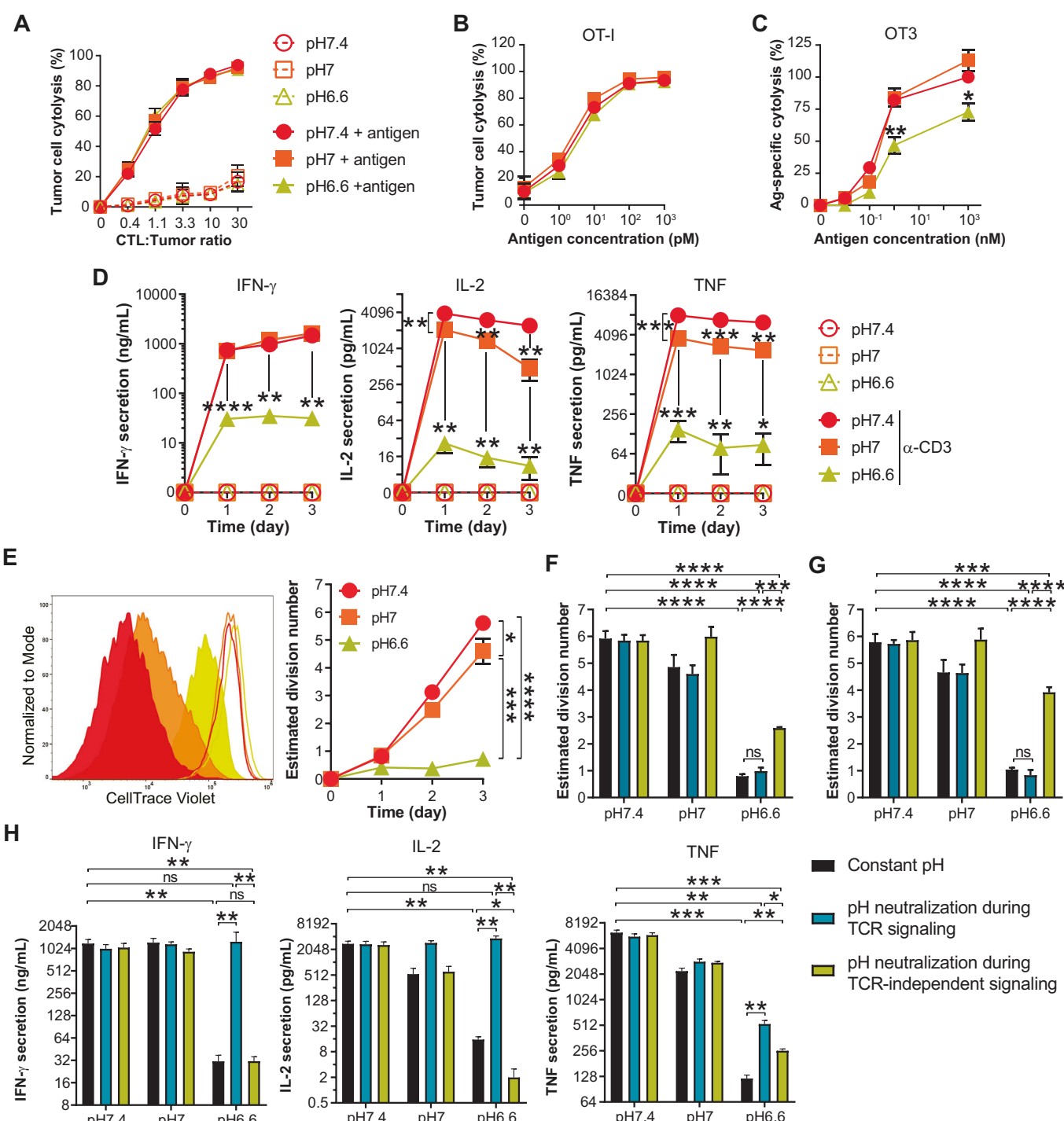

on T cells in the condition in which they would be found in tumors, we established an in vitro expansion protocol giving rise to effector murine CTLs (Fig. EV1B–D). Briefly, naive CD8[+] T cells were activated for two days with anti-CD3 and anti-CD28 antibodies and expanded over several days in the presence of high-dose murine IL-2 (200 IU/mL) to favor effector differentiation (Ross and Cantrell, 2018). Tumor pH is highly variable (Feng et al, 2024; Tannock and Rotin, 1989) and we thus set out to assess the impact of pH 7.4 as a physiological control, pH 7, and pH 6.6, a level of

acidity that can be found in tumors (and often explored in pH studies). For OT-I TCR transgenic T cells which recognize the MHC class I/SIINFEKL peptide complex with high affinity, we observed no difference in cytolytic capacity at the different pH, even when modulating antigen density on tumor cells (Fig. 1A,B). In contrast, weaker affinity OT-3 TCR CTLs demonstrated decreased cytolytic function at pH 6.6 (Fig. 1C). We further evaluated cytokine production and proliferation upon re-activation with anti-CD3 antibodies, as opposed to with tumor cells in order to exclude any

◀ **Figure 1. Low pH impairs the function of CTLs upon re-activation.**

(A) Impact of pH on CTL killing capacities of OT-I as a function of CTL-to-tumor ratio. C1498 tumor cells were pulsed, or not, with 1 μM of antigen (minimal ovalbumin peptide epitope, SIINFEKL). Results show the mean percentage of tumor cell lysis ± SD of at least three (or two for 0.4 and 1.1 CTL:Tumor ratios) biological replicates from at least two independent experiments. (B) Impact of pH on CTL killing capacities of OT-I as a function of antigen density. C1498 tumor cells were pulsed with varying amounts of antigen and co-cultured with CTLs at a 3.3 CTL-to-tumor ratio. Results show the mean percentage of tumor cell lysis ± SD of three biological replicates from two independent experiments. (C) Impact of pH on CTL killing capacities of OT-3 as a function of antigen density. The methodology is the same as in (B). Results show the mean percentage of antigen-specific tumor cell lysis, normalized to the condition pH 7.4 with 1 μM of antigen, ±SEM of three biological replicates from two independent experiments. Statistic comparisons between pH 6.6 and pH 7.4: *$P < 0.0248$, **$P = 0.0088$ (Student's paired $t$ test). (D) Impact of pH on cytokine secretion profiles of CTLs. OT-I CTLs were co-cultured in the presence, or absence, of an agonistic anti-CD3 antibody (1 μg/mL pre-coated plates). Results show the mean secretion of IFN-γ, IL-2 and TNF-α ± SEM of at least three biological replicates from two independent experiments, as detected by ELISA from supernatants. IFN-γ (pH 6.6 vs pH 7.4) – D1: ****$P < 0.0001$. D2: **$P = 0.003$. D3: **$P = 0.0015$. IL-2 – D1: pH 7 vs pH 7.4 **$P = 0.0085$, pH 6.6 vs pH 7.4 **$P = 0.0047$. D2: pH 7 vs pH 7.4 **$P = 0.004$, pH 6.6 vs pH 7.4 **$P = 0.0059$. D3: pH 7 vs pH 7.4 **$P = 0.0037$, pH 6.6 vs pH 7.4 **$P = 0.0016$. TNF – D1: pH 7 vs pH 7.4 ***$P = 0.0008$, pH 6.6 vs pH 7.4 ***$P = 0.0004$. D2: pH 7 vs pH 7.4 ***$P = 0.0008$, pH 6.6 vs pH 7.4 **$P = 0.0012$. D3: pH 7 vs pH 7.4 **$P = 0.004$, pH 6.6 vs pH 7.4 *$P = 0.028$. (Student's paired $t$ test). (E) Impact of pH on CTL proliferation upon anti-CD3 re-activation. Histograms show one representative experiment of CellTrace Violet dilution upon re-activation with anti-CD3 (solid histograms), or without stimulation (empty histograms) at day 3. Bar graph shows the estimated division number ± SEM of four biological replicates from two independent experiments. *$P = 0.0498$, ***$P = 0.0002$, ****$P < 0.0001$ (one-way repeated measures ANOVA, Tukey post-hoc test). (F, G) Acidity affects CTL proliferation during TCR/CD3-independent signaling. OT-I CTLs were cultured for 1 day in anti-CD3-coated plates (TCR/CD3 signaling step). The resulting cells and supernatants were transferred to an anti-CD3-free plate and cultured for two further days (TCR/CD3-independent signaling step; cf. Fig. EV1G). Cells were either: cultured during the TCR/CD3 signaling step and the TCR/CD3-independent signaling at constant pH ("Constant pH"), cultured at pH 7.4 during the TCR/CD3 signaling step then at various pH during the TCR/CD3-independent signaling step ("pH neutralization during TCR signaling"), or cultured at various pH during TCR/CD3 signaling step then at pH 7.4 during the TCR/CD3-independent signaling step ("pH neutralization during TCR-independent signaling"). Cell proliferation was measured by flow cytometry. (G) Same methodology as in (F), but adding exogenous murine IL-2 (200 IU/mL). Bar graphs show the estimated division number + SEM of four biological replicates from two independent experiments. ns: not statistically significant, ***$P = 0.0003$ in (F), ***$P = 0.0006$ in (G), ****$P < 0.0001$ (one-way repeated measures ANOVA, Tukey post-hoc test). (H) Acidity influences cytokine secretion profile of CTLs during TCR signaling. The methodology was the same as in (F). Cytokine secretion was measured by ELISA. Bar graph shows the mean cytokine secretion of IFN-γ, IL-2 and TNF-α + SEM of four biological replicates from two independent experiments. ns not statistically significant. IFN-γ: pH 6.6 vs pH 7.4 **$P = 0.0052$, pH 6.6 neutralization during TCR-independent vs pH 7.4 **$P = 0.0054$, pH 6.6 neutralization during TCR vs pH 6.6 **$P = 0.0018$, pH 6.6 neutralization during TCR-independent vs pH 6.6 neutralization during TCR **$P = 0.0017$. IL-2: pH 6.6 vs pH 7.4 **$P = 0.0077$, pH 6.6 neutralization during TCR-independent vs pH 7.4 **$P = 0.0077$, pH 6.6 neutralization during TCR vs pH 6.6 **$P = 0.0065$, pH 6.6 neutralization during TCR-independent vs pH 6.6 *$P = 0.0352$, pH 6.6 neutralization during TCR-independent vs pH 6.6 neutralization during TCR **$P = 0.0066$. TNF: pH 6.6 vs pH 7.4 ***$P = 0.0008$, pH 6.6 neutralization during TCR-independent vs pH 7.4 **$P = 0.0011$, pH 6.6 neutralization during TCR vs pH 7.4 ***$P = 0.001$, pH 6.6 neutralization during TCR vs pH 6.6 **$P = 0.0061$, pH 6.6 neutralization during TCR-independent vs pH 6.6 **$P = 0.003$, pH 6.6 neutralization during TCR-independent vs pH 6.6 neutralization during TCR *$P = 0.0232$ (one-way repeated measures ANOVA, Tukey post-hoc test). Source data are available online for this figure.

CTL-extrinsic impact of acidity. Upon re-stimulation at pH 6.6 we observed a strong decrease in the production of all cytokines assessed (IFN-γ, IL-2 and TNF; Fig. 1D). At low pH, CTL proliferation was also dampened (Fig. 1E) independently of the stimulation strength (Fig. EV1E). We further observed that acidity prevented the increase in T-cell size and granularity normally induced upon activation at pH 7.4 (Fig. EV1F).

Next, we sought to determine at which stage proliferation is impaired upon CTL re-activation at low pH. CTL re-activation can be roughly divided into two sequential steps: (i) a first TCR/CD3-dependent step leading to the production of cytokines, and, (ii) a second one that relies mainly upon autocrine IL-2 production and IL-2R signaling (Fig. EV1G). In order to determine the stage at which low pH impacts CTL proliferation, we acidified and subsequently neutralized the pH at one step or the other (Fig. EV1H). We observed that pH neutralization during the IL-2R-dependent step improved CTL proliferation (Fig. 1F), even more so in the presence of exogenous IL-2 (Fig. 1G). Whereas, neutralizing pH during the TCR/CD3 signaling step could restore both the cytokine secretion profile (Fig. 1H) and cell-size augmentation (Appendix Fig. S1A). Taken together, our data show that acidity impacts both steps of CTL reactivation; low pH during TCR/CD3 triggering blunts cytokine secretion and cytolysis (the latter for low-affinity TCR T cells), and acidic conditions at the IL-2R signaling phase impairs proliferation.

## Acidity abrogates IL-2-mediated proliferation of CTLs

To further validate that acidity dampens IL-2R signaling-driven proliferation we took advantage of the fact that CTLs maintain responsiveness to IL-2 without re-activation via TCR/CD3 triggering. We observed that even upon high dose IL-2 stimulation (200 IU/mL) acidity lowered CTL proliferation and viability as well as cell-size and granularity (Fig. 2A; Appendix Fig. S1B,C). Notably, pH 6.6 did not preferentially block T cells in a particular phase of the cell cycle (Appendix Fig. S1D).

We subsequently sought to evaluate potential changes to the cell-surface expression of the IL-2R complex at low pH. Briefly, the IL-2R comprises an α (CD25), β (CD122), and γ (CD132) chain and can engage IL-2 by a βγ (intermediate affinity) or αβγ (high affinity) complex. The latter complex can be found on activated T cells upon upregulation of the α-chain. For both complexes, IL-2R signaling is mediated by the cytoplasmic domains of the IL-2Rβ and IL-2Rγ chains (Liao et al, 2013; Ross and Cantrell, 2018; Wang et al, 2005). After 24 h incubation at pH 6.6 we observed a decrease in the β- and γ-chains at the cell-surface, and this was even more pronounced at pH 6.2 (Fig. 2B). Notably, we also detected lower levels of both chains when the cells were pre-incubated with IL-2, presumably due to steric hindrance of antibody-binding and/or internalization of the chains (Fig. 2B). To evaluate if there was also a decrease in functional IL-2R complexes at low pH, we incubated the cells with biotinylated IL-2 (bio-IL-2) to allow detection of its binding (Fig. 2C). For this assay, competition by previously bound non-bio-IL-2 probably precludes detection of all cell-surface IL-2R complexes. We observed a decrease in bio-IL-2/IL-2R binding capacity at pH 6.6, which was even more pronounced at pH 6.2 (Fig. 2D). Overall, since only the most extreme acidity tested (i.e., pH 6.2) was associated with substantially decreased IL-2R levels and highly impaired IL-2/IL-2R binding, we conclude that the

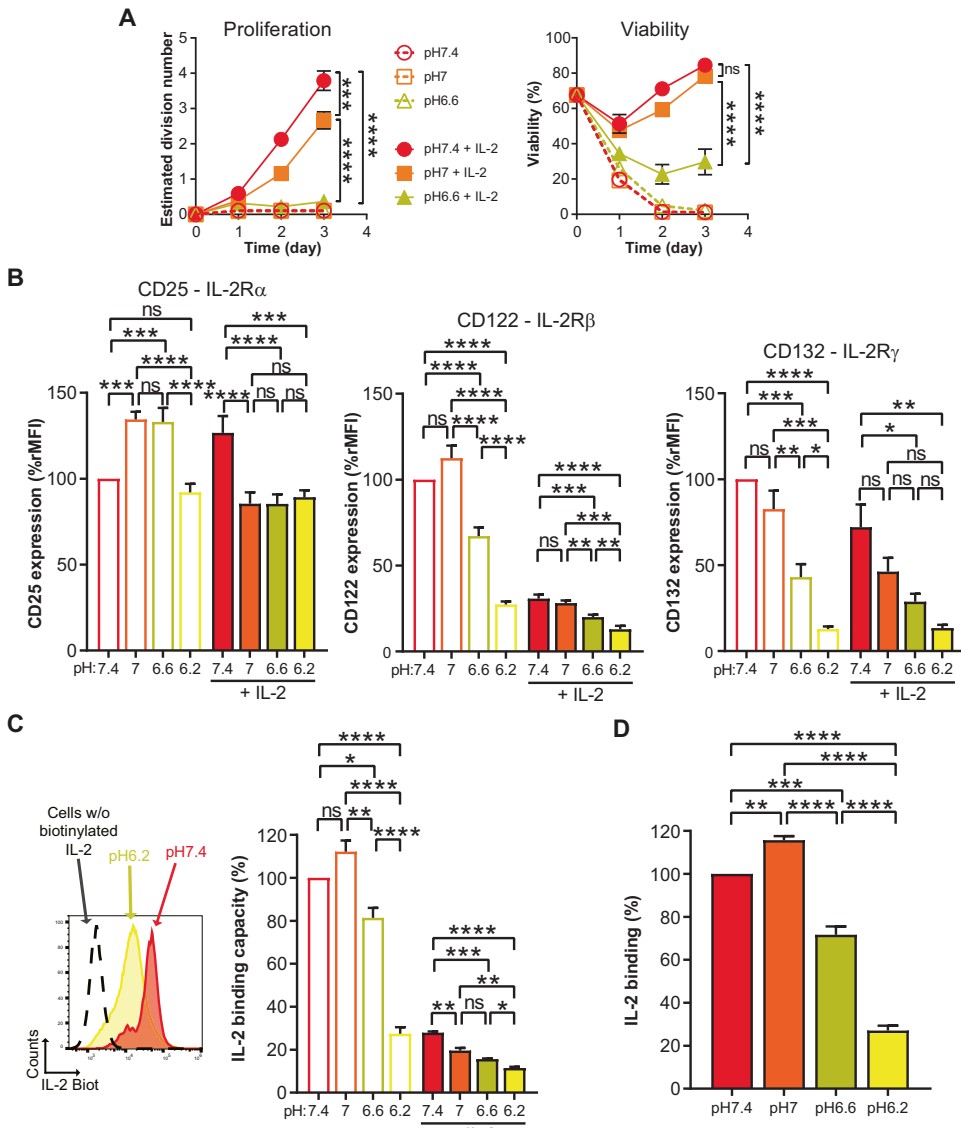

**Figure 2. Acidity lowers IL-2-dependent proliferation of CTLs.**

(A) Time-course of the pH impact on proliferative and survival responses of CTLs to IL-2. OT-I CTLs were cultured in the presence, or the absence, of exogenous murine IL-2 (200 IU/mL). Results show the estimated division number, or viability, ± SEM of at least three biological replicates from at least two independent experiments. ns: not significant, ***P = 0.0007, ****P < 0.0001 (one-way repeated measures ANOVA, Tukey post-hoc test). (B) Impact of pH on protein expression of IL-2R subunits. OT-I CTLs were cultured for one day. Results show the mean expression of the indicated subunits + SEM of three biological replicates from two independent experiments. Expression was calculated as ratio median fluorescence intensity and was normalized to the condition pH 7.4 without exogenous IL-2. ns not significant, ****P < 0.0001. CD25: pH 7 vs pH 7.4 ***P = 0.0003, pH 6.6 vs pH 7.4 ***P = 0.0004, pH 6.2 + IL-2 vs pH 7.4 + IL-2 ***P = 0.001. CD122: pH 6.6 + IL-2 vs pH 7+IL-2 **P = 0.0044, pH 6.2 + IL-2 vs pH 7+IL-2 ***P = 0.0001, pH 6.2 + IL-2 vs pH 6.6 + IL-2 **P = 0.0093. CD132: pH 6.6 vs pH 7.4 ***P = 0.0009, pH 6.6 vs pH 7 **P = 0.0061, pH 6.2 vs pH 7 ***P = 0.0003, pH 6.2 vs pH 6.6 *P = 0.0219, pH 6.6 + IL-2 vs pH 7.4 + IL-2 *P = 0.0184, pH 6.2 + IL-2 vs pH 7.4 + IL-2 **P = 0.0042. (one-way repeated measures ANOVA, Tukey post-hoc test). (C) Impact of pH on the levels of IL-2R complexes that can bind IL-2. OT-I CTLs were cultured for one day at three different pH in the presence, or absence, of exogenous murine IL-2. One representative experiment is shown as flow cytometry histograms: dashed line show CTLs cultured at pH 7.4 without IL-2 for one day and not stained with biotinylated-IL-2, while red and yellow histograms represent CTLs cultured without IL-2 for one day at pH 7.4 and pH 6.2 and stained with biotinylated IL-2, respectively. Bar graph shows the mean IL-2 binding capacity +SEM of three biological replicates from two independent experiments. Binding capacity was calculated as ratio median fluorescence intensity of surface-detected biotinylated IL-2, normalized to the condition pH 7.4 without exogenous IL-2. ns not significant, ****P < 0.0001. pH 6.6 vs pH 7.4 *P = 0.0162, pH 6.6 vs pH 7 **P = 0.0012, pH 7+IL-2 vs pH 7.4 + IL-2 **P = 0.0019, pH 6.6 + IL-2 vs pH 7.4 + IL-2 ***P = 0.0002, pH 6.2 + IL-2 vs pH 7+IL-2 **P = 0.0023, pH 6.2 + IL-2 vs pH 6.6 + IL-2 *P = 0.0357. (one-way repeated measures ANOVA, Tukey post-hoc test). (D) Impact of pH on the binding of IL-2 to IL-2R. OT-I CTLs stained with biotinylated IL-2 for 30 min at 4 °C in PBS at various pH. Bar graph shows the mean IL-2 binding +SEM of three biological replicates from two independent experiments. **P = 0.0096, ***P = 0.0004, ****P < 0.0001 (one-way repeated measures ANOVA, Tukey post-hoc test). Source data are available online for this figure.

profound decrease in proliferation and viability of T cells cultured at pH 6.6 cannot be attributed to lower IL-2/IL-2R engagement.

Finally, in order to exclude that the impact of acidity on T cells resulted from indirect effects such as precipitation/inactivation of components in the medium, we analyzed T-cell proliferation with medium previously acidified and then re-adjusted to pH 7.4 with NaOH. No proliferation defect was observed for T cells upon restoration of physiologic pH. In addition, because the experimental set-up involves HCl to reach the desired pH, we further tested if an increase in osmolarity could impact T-cell proliferation (by addition of NaCl) but this was not the case (Appendix Fig. S1E).

## Acidity blunts IL-2R signaling and specifically lowers the mTORC1 pathway and c-Myc levels

Given that IL-2/IL2-R complex formation is only modestly reduced at pH 6.6 but IL-2-mediated proliferation is nonetheless significantly impaired, we next sought to evaluate changes to intracellular signaling components. Briefly, IL-2/IL-2R complex formation first triggers the phosphorylation of Janus Kinases (JAK), JAK1 and JAK3. The JAKs then phosphorylate tyrosine residues in the intracellular domains of the IL-2R to generate docking sites for the signal transducer and activator of transcription (STAT) factors (e.g., STAT5) which dimerize and translocate to the nucleus and induce gene expression programs. The JAKs also initiate PI3K/Akt/mTORC1 and MAPK/ERK signaling pathways, and promote c-Myc transcriptional activity (Fig. EV2A) (Liao et al, 2013). While we observed an important pH-dependent reduction in the phosphorylation of JAK1, JAK3 and STAT5 (Figs. 3A and EV2B; Appendix Fig. S2A) at pH 6.2, the decrease was only modest at pH 6.6. However, although upstream transducers were slightly altered at pH 6.6, we observed a strong impact on both the mTORC1 pathway (as evidenced by phosphorylation of p70S6K, S6 and 4E-BP1) (Laplante and Sabatini, 2009; Saxton and Sabatini, 2017) and c-Myc levels (Figs. 3B and EV2C; Appendix Fig. S2B). In contrast, the MAPK/ERK pathway was not suppressed and even increased at a later time-point, presumably due to cell-cycle differences.

We subsequently assessed the activation of various pathways that could lead to mTORC1 inhibition, including (i) stress response (p38, eiF2α), (ii) energy homeostasis (AMPK), (iii) Wnt/Hedgehog signaling (GSK-3β), (iv) amino acid starvation (eiF2α) and, (v) cAMP response (CREB/CREM) (Aramburu et al, 2014; Nikonorova et al, 2018; Xie et al, 2011), but we did not observe any notable differences to any of them at low pH (Appendix Figs. S3 and S4). In line with the described role of mTORC1 as a major regulator of protein synthesis, we observed that acidic pH was associated with lower protein content in the cells (Appendix Fig. S5A). We further performed RNA sequencing on CTLs cultured for 24 h at pH 7.4 versus pH 6.6 in the presence of IL-2, and gene set enrichment analysis identified c-Myc targets and the mTORC1 pathway as downregulated at low pH (Appendix Fig. S5B).

Next, we evaluated the impact of low pH on IL-2R signaling as well as mTORC1 and c-Myc at 4 h, a time point at which cell viability and apoptosis are not impacted by acidic conditions (Appendix Fig. S6A). We sought to elucidate whether the modest decrease in IL-2/IL-2R binding and JAK/STAT phosphorylation at pH 6.6 could account for the significant decreases in phosphorylation of molecules downstream of mTORC1 and to lower c-Myc

levels. We performed an IL-2 dose-response experiment at pH 7.4 versus pH 6.6 and observed impaired mTORC1 activation and lower c-Myc at levels seemingly disproportionate to a global decrease in IL-2R signaling (Fig. 3C). Indeed, by plotting individual values of mTORC1 pathway activation (p-p70S6K as a surrogate) and c-Myc levels as a function of JAK1/JAK3/STAT5 phosphorylation (surrogates of IL-2R signaling) as correlation curves (Fig. EV2D), we observed no overlap for OT-I TCR T cells cultured at pH 6.6 versus pH 7.4. These data suggest that the decreases in mTORC1 signaling, and to a lesser extent in c-Myc levels, are not due to lower IL-2R signaling. The same was observed for wild-type polyclonal C57BL/6 CTLs (Appendix Fig. S6B,C).

Recently, Gaggero et al, reported the development of a human IL-2 mutein (Switch-2) able to preferentially engage IL-2R at low pH, thereby restoring signaling and proliferation (Gaggero et al, 2022). We generated Switch-2 in-house but could not reproduce their findings for either murine (Appendix Fig. S7A,B) or human CD8+ T cells (Appendix Fig. S8A,B). Switch-2 is described to specifically rescue binding to the IL-2Rα subunit at low pH. However, we observed that although knockout of IL-2Rα lowered all signaling parameters analyzed upon IL-2 stimulation, acidity still had a negative impact (Appendix Fig. S9A–D).

Finally, we performed a dose-response curve for anti-CD3 antibody stimulation to determine if mTORC1 and c-Myc are also altered at low pH during CTL re-activation. Both mTORC1 signaling and c-Myc levels were blunted at low pH, but at high levels of anti-CD3 antibody c-Myc levels could be restored (Fig. 4A). We thus investigated TCR/CD3 signaling pathway molecules SLP-76, ERK and PLCγ1 (Fig. 4B) and found that at low pH the $EC_{50}$ to reach maximum phosphorylation levels (as achieved at pH 7.4) increased. This indicates that low pH augments the activation threshold, resulting in the c-Myc response observed.

## Low pH leads to decreased *Myc* transcription and to proteasome-mediated c-Myc degradation

We next questioned whether mTORC1 and c-Myc cross-regulate one another and thus performed a kinetic analysis. We observed a drop in mTORC1 as early as 5 to 15 min post low-pH exposure (even in the absence of exogenous IL-2) while c-Myc only decreased after an hour in an IL-2-dependent manner (Fig. 5A). Although mTORC1 is a major regulator of protein synthesis (i.e., its inhibition at low pH could account for the decrease in c-Myc protein levels), its inhibition by rapamycin was associated with only a minimal decrease in c-Myc (Fig. 5B) thereby indicating that there is no interplay between mTORC1 and c-Myc under acidic conditions (at least during the first four hours). We further explored *Myc* mRNA levels and found them decreased at pH 6.6 (Fig. 5C), at least in part due to lower p-JAK1/JAK3/STAT5 signaling (Fig. 5D).

A kinetic study of *Myc* mRNA and c-Myc protein levels at pH 6.6 (Fig. 5E) revealed that, in the presence of IL-2, both were lowered within the first hour and that c-Myc protein had maximally decreased by two hours, matching levels observed in the absence of IL-2. These observations suggested that c-Myc protein levels are controlled by *Myc* mRNA levels. After 4 h at low pH, however, *Myc* mRNA levels were higher in the presence of IL-2 than in its absence despite equivalent c-Myc protein levels, thus indicating a role for post-transcriptional regulation (Farrell and Sears, 2014). The post-

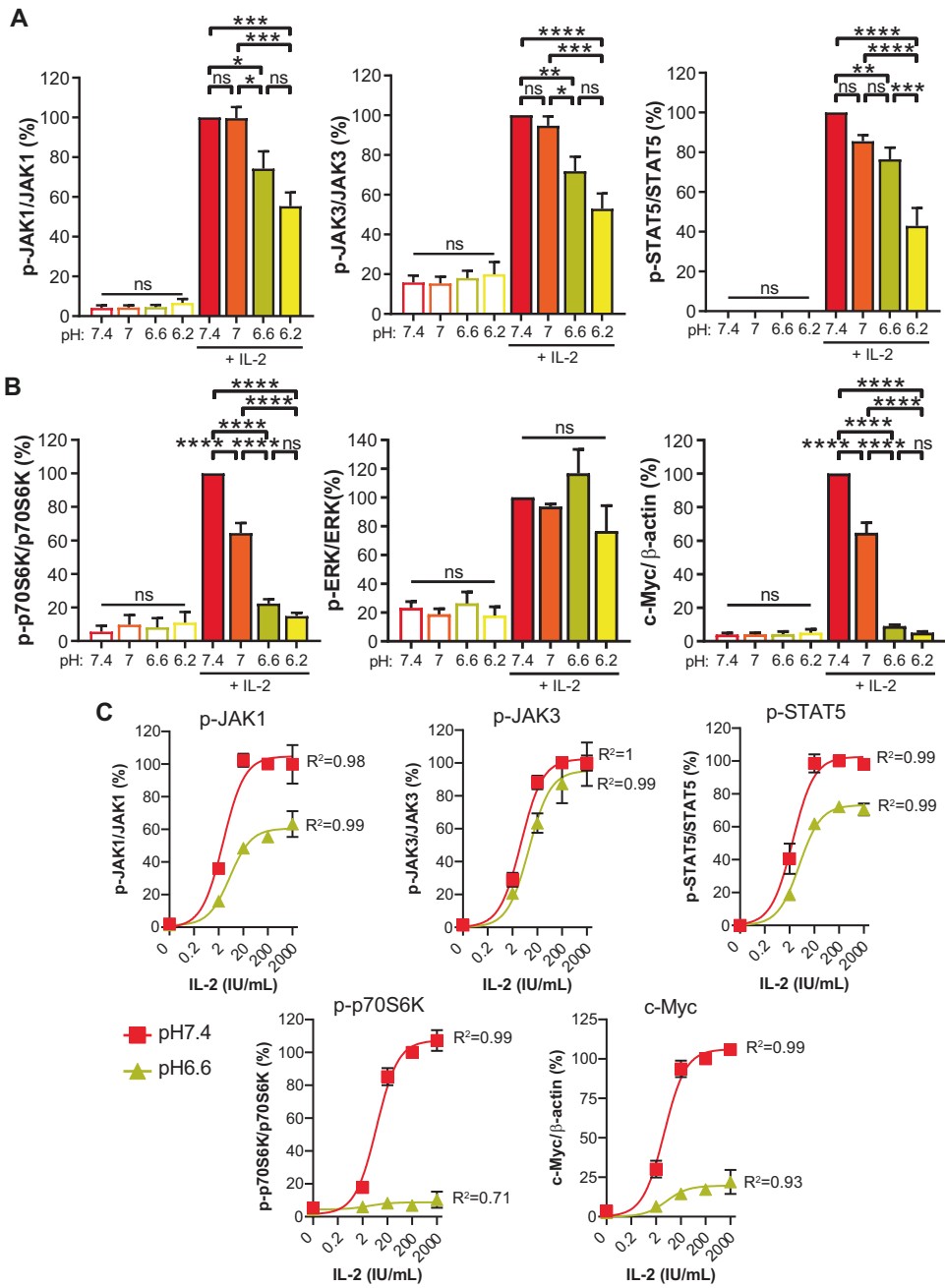

**Figure 3. Low pH disrupts IL-2 responsiveness, mTORC1 signaling and c-Myc levels.**

(A) IL-2R signaling is disrupted at lower pH. OT-I CTLs were cultured for 4 h (without prior starving). Bar graphs show the mean levels normalized to the condition "pH 7.4 + IL-2" +SEM of at least five biological replicates from at least two independent experiments. ns not significant, ****$P < 0.00001$. p-JAK1: pH 6.6 + IL-2 vs pH 7.4 + IL-2 *$P = 0.0138$, pH 6.2 + IL-2 vs pH 7.4 + IL-2 ***$P = 0.0002$, pH 6.6 + IL-2 vs pH 7+IL-2 *$P = 0.015$, pH 6.2 + IL-2 vs pH 7+IL-2 ***$P = 0.0002$. p-JAK3: pH 6.6 + IL-2 vs pH 7.4 + IL-2 **$P = 0.0076$, pH 6.6 + IL-2 vs pH 7+IL-2 *$P = 0.0306$, pH 6.2 + IL-2 vs pH 7+IL-2 ***$P = 0.0002$. p-STAT5: pH 6.6 + IL-2 vs pH 7.4 + IL-2 **$P = 0.0093$, pH 6.2 + IL-2 vs pH 6.6 + IL-2 ***$P = 0.0003$ (one-way repeated measures ANOVA, Tukey post-hoc test). (B) Low pH disturbs IL-2 –induced mTORC1 pathway and c-Myc levels. The methodology was the same as in (A). Bar graphs show the mean levels normalized to the condition "pH 7.4 + IL-2" +SEM of at least five biological replicates from at least two independent experiments. ns: not significant, ****$P < 0.0001$ (one-way repeated measures ANOVA, Tukey post-hoc test). (C) IL-2R signaling dose-response. OT-I CTLs were for 4 h. Results show the mean levels normalized to the condition "pH 7.4 + IL-2 200 IU/mL" ± SEM of four biological replicates from two independent experiments. Corresponding correlation curves together with associated $R^2$ are displayed. Source data are available online for this figure.

transcriptional regulation of c-Myc is oftentimes dependent upon its phosphorylation status at T58 (GSK-3β-dependent; pT58 promotes proteasomal degradation) and S62 (ERK-dependent; pS62 improves stability). However, while we did not observe an alteration in pT58/pS62 at low pH, and GSK-3β inhibition (CHIR99021/GSKi) did not upregulate c-Myc levels, we found that proteasome inhibition (MG-132) augmented c-Myc levels (Fig. 5F). Hence, we next explored the half-life of c-Myc using cycloheximide

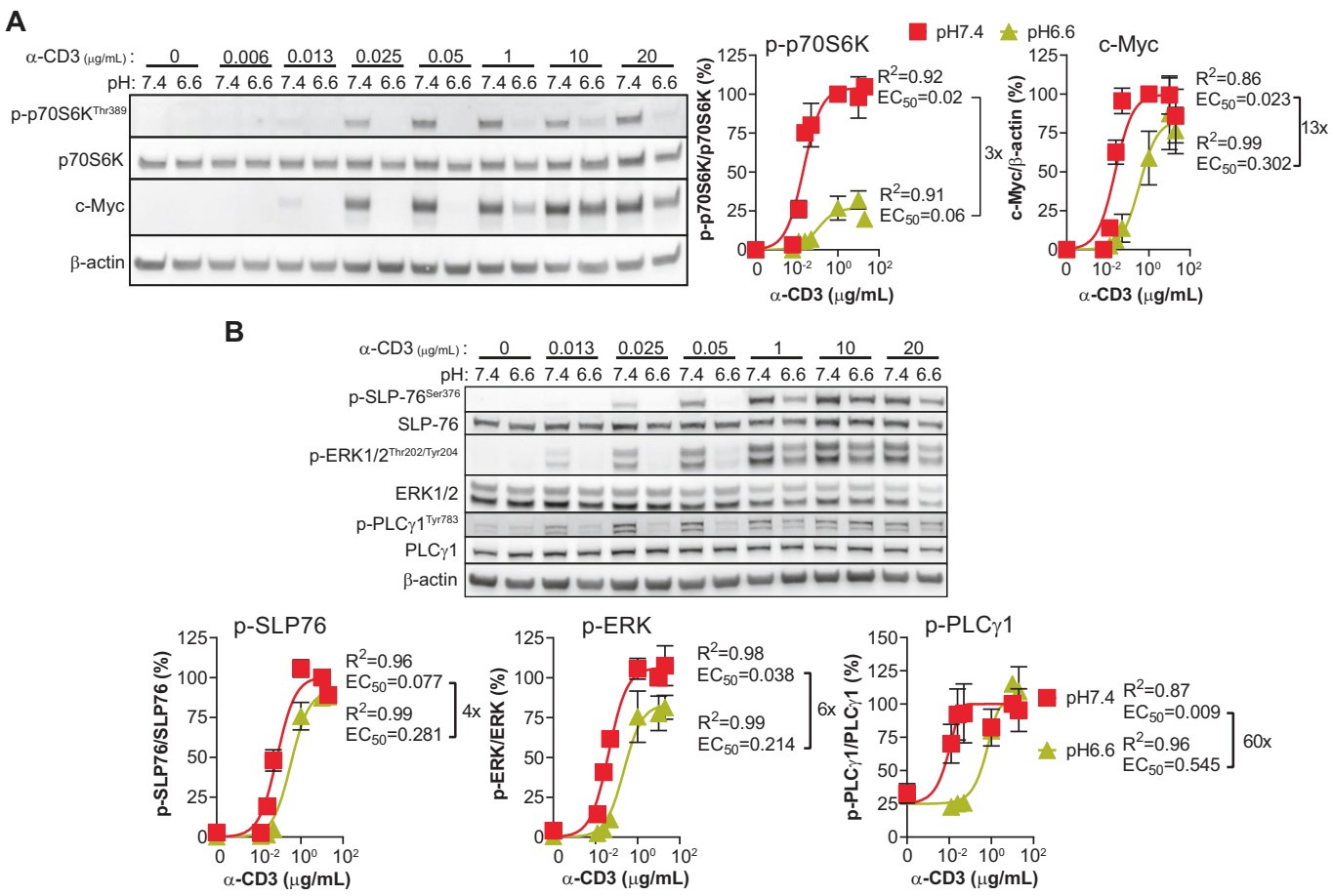

**Figure 4. Impact of low pH on TCR/CD3 signaling during CTL re-activation.**

(A) mTORC1 activity and c-Myc levels upon TCR/CD3 signaling dose-response at low pH. OT-I CTLs were cultured in the presence of various doses of coated- anti-CD3 antibodies for 4 h. One representative western blot experiment is shown. Results show the mean levels, normalized to the condition "pH 7.4 + anti-CD3 1 μg/mL" ± SEM of three biological replicates from two independent experiments. Corresponding correlation curves together with associated $R^2$ and EC50 are displayed. (B) TCR/CD3 signaling dose-response at low pH. The methodology was the same as in (A). One representative western blot experiment is shown. Results show the mean levels normalized to the condition "pH 7.4 + anti-CD3 10 μg/mL" ± SEM of three biological replicates from two independent experiments. Corresponding correlation curves together with associated $R^2$ and EC50 are displayed. Source data are available online for this figure.

(CHX) (Fig. 5G), a protein synthesis inhibitor that can preclude any transcriptional bias. Based on c-Myc kinetics (Fig. 5E), we pretreated CTLs for 1.5 h before CHX addition so that the assay was conducted at the time point at which there is the full impact of acidity on c-Myc levels. Interestingly, we observed that acidity lowered c-Myc half-life from 37 min to 23 min. Taken together, our results indicate that the drop in c-Myc levels under acidic conditions is caused by lower *Myc* transcription at least in part as a result of weaker IL-2R signaling, and also by increased proteasomal degradation which lowers the half-life of c-Myc.

## TSC2 knockout augments mTORC1 activity at low pH but does not restore CTL proliferation, even in combination with c-Myc overexpression

We next sought to explore mechanisms limiting mTORC1 activity at low pH. Interestingly, although mTORC1 inhibition with rapamycin recapitulated proliferative defects observed under acidic conditions (Fig. EV3A), the phosphorylation status of Akt

(an important activator of mTORC1; Fig. EV3B), was not altered at pH 6.6 (Fig. 6A). Moreover, although Akt inhibition lowered mTORC1 activation and c-Myc accumulation, at low pH we did not detect a decrease in the phosphorylation status of direct Akt targets GSK-3β and FoxO, while PRAS40 phosphorylation was only modestly lowered. Subsequently, by CRISPR/Cas9 we individually knocked out PRAS40 and TSC2, the latter of which is a major negative regulator of mTORC1 that is also controlled by Akt (Fig. 6B). PRAS40 was not found to play a role in the regulation of mTORC1 activity, but loss of TSC2 resulted in mTORC1 activation, even in the absence of IL-2 and regardless of pH. The addition of IL-2 further increased mTORC1 activity in TSC2 knockout CTLs at pH 7.4 in an Akt-independent manner (Fig. EV3C), but had little impact on mTORC1 at pH 6.6. Despite the elevated mTORC1 activity at low pH in TSC2 knockout cells, there was not an increase in CTL proliferation (Fig. 6C).

Rheb is directly inhibited by TSC2, and Rheb overexpression has been reported to facilitate unhindered mTORC1 encounter and activation (Angarola and Ferguson, 2020; Garami et al, 2003).

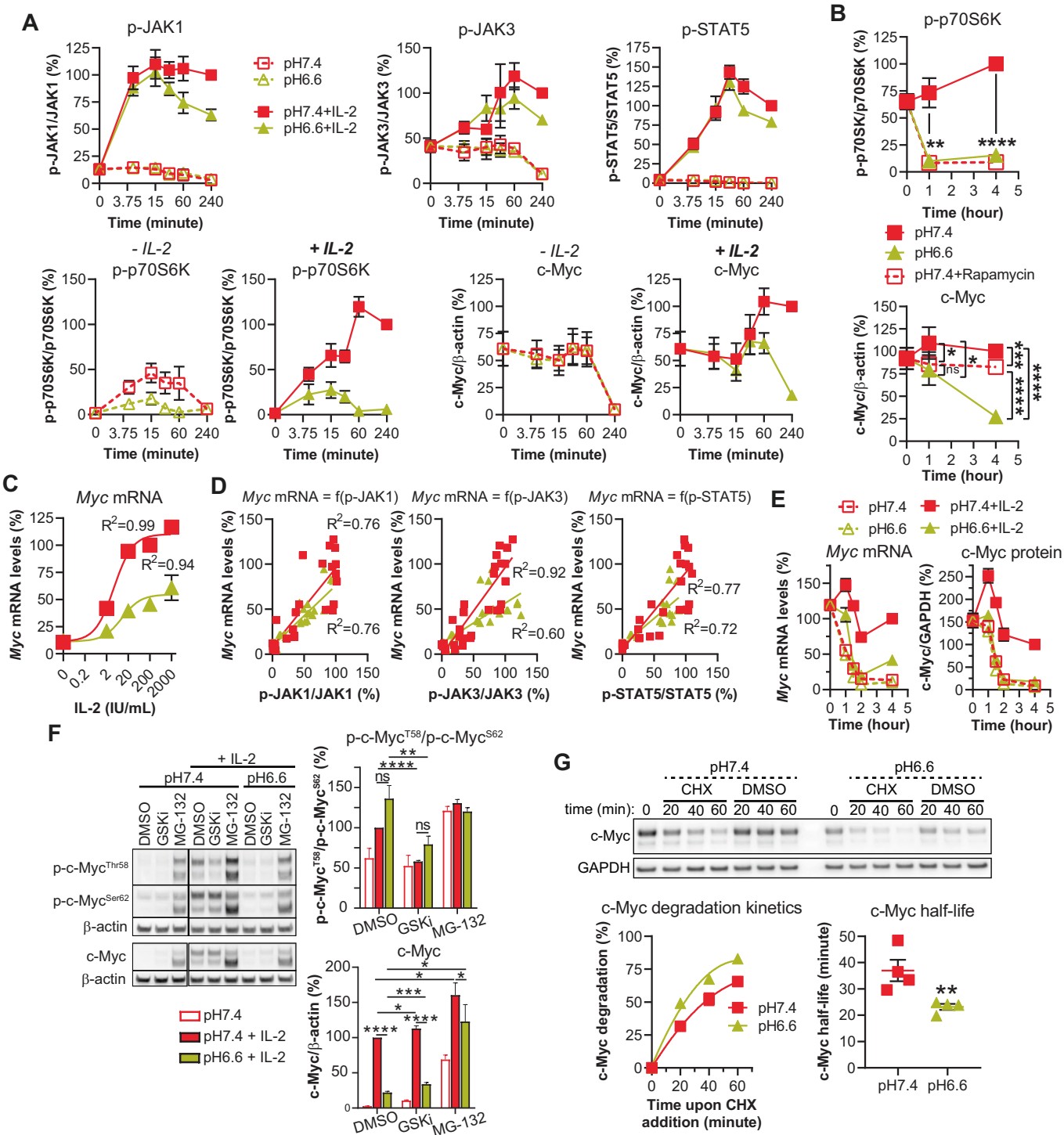

We overexpressed Rheb and observed similar effects as for TSC2 knockout cells, albeit less pronounced (Fig. EV3D). In addition to growth factors and Akt, mTORC1 activity is regulated by amino acid insufficiency and energy stress via GATOR1 and the Lkb1/AMPK axis, respectively (Fig. EV3B). Interestingly, knockout of Nprl2 which blocks the GATOR1 axis led to activation of mTORC1 even in the absence of IL-2, but not at pH 6.6 (Fig. EV3E). Also, despite increasing mTORC1 activity upon addition of IL-2 at pH

7.4, Nprl2 and Lkb1 knockouts minimally augmented mTORC1 activity under acidic conditions.

Despite promoting mTORC1 activity at low pH, knocking out TSC2 is probably not sufficient to restore cell expansion because other pathways like c-Myc are disturbed in a mTORC1-independent manner. Hence, we next overexpressed c-Myc in conjunction to knocking out TSC2 (Fig. 6D). Importantly, retroviral transfer of the wild-type c-Myc transgene allowed protein

Figure 5.   Low pH leads to decreased *Myc* transcription and to proteasome-mediated c-Myc degradation in CTLs.

(A) Time-course of IL-2R signaling. Results show the mean ± SEM of four biological replicates from two independent experiments. (B) Impact of mTOR inhibition of c-Myc levels. OT-I CTLs were cultured with exogenous murine IL-2 in the absence, or in the presence of rapamycin 10 nM. Results show the mean ± SEM of four biological replicates from two independent experiments. ns not statistically significant, ****$P < 0.0001$. p-p70S6K: **$P = 0.0043$. c-Myc: 1 h 7.4 Rapa vs pH 7.4 *$P = 0.0134$, 1 h pH 6.6 vs pH 7.4 *$P = 0.0383$, 4 h pH 7.4 Rapa vs pH 7.4 ***$P = 0.0005$ (one-way repeated measures ANOVA, Tukey post-hoc test). (C) *Myc* mRNA transcription. OT-I CTLs were cultured for 4 h. Line graph displays the mean percentage of *Myc* mRNA levels relative to *Actb* as a function of extracellular IL-2, normalized to the condition "pH 7.4 + IL-2 200 IU/mL" ± SEM of four biological replicates from two independent experiments. A correlation curve and $R^2$ of four pH is displayed. (D) *Myc* mRNA levels as a function of IL-2R signaling. OT-I CTLs were cultured for 4 h with various concentrations of exogenous murine IL-2 at pH 7.4. Dot plots display the individual percentage of *Myc* mRNA levels relative to *Actb* as a function of extracellular JAK1, JAK3 or STAT5 phosphorylation, normalized to the condition "pH 7.4 + IL-2 200 IU/mL" from four biological replicates from two independent experiments. A correlation curve and $R^2$ for each pH is displayed. (E) Kinetics of *Myc* mRNA and c-Myc protein levels. OT-I CTLs were cultured for 1, 1.5, 2 or 4 h. Results show the mean *Myc* mRNA or c-Myc protein levels ± SEM normalized to the condition "pH 7.4 + IL-2, 4 h" from four biological replicates from two independent experiments. (F) Acidity does not modify c-Myc phosphorylation but decreases c-Myc levels via the proteasome. OT-I CTLs were cultured for 4 h with, or without, the proteasome inhibitor MG-132 (10 μM) or the GSK inhibitor CHIR99021 (2 μM - GSKi). A representative western blot is shown. Bar graph on the left hand displays the mean ratio p-c-MycT58/p-c-MycS62 normalized to the condition "pH 7.4 + IL-2 + DMSO" +SEM of four biological replicates from two independent experiments. Bar graph on the right hand shows the total levels of c-Myc relative to β-actin, normalized to the condition "pH 7.4 + IL-2 + DMSO" +SEM of four biological replicates from two independent experiments. ns not significant, ****$P < 0.0001$. p-c-Myc ratio: **$P = 0.0052$. c-Myc: pH 7.4+GSKi vs pH 7.4 + DMSO *$P = 0.0415$, pH 6.6 + MG-132 vs pH 7.4 + DMSO *$P = 0.0393$, pH 6.6 + MG-132 vs pH 6.6 + DMSO *$P = 0.0288$, pH 6.6+GSKi vs pH 6.6 + DMSO ***$P = 0.0003$, pH 6.6 + MG-132 vs pH 7.4 + MG-132 *$P = 0.0166$. (Student's paired $t$ test). (G) Acidity lowers c-Myc half-life. OT-I CTLs were pre-cultured for 1.5 h with exogenous murine IL-2 at pH 7.4 or 6.6. Cycloheximide (CHX, 50μg/mL) was then added for 20, 40 or 60 min. A representative western blot to detect c-Myc degradation is shown. Line graph on the left hand displays the mean c-Myc degradation ± SEM (as compared to time 0 per pH) together with the corresponding correlation curve. c-Myc half-life (individual plots with mean ± SEM) calculated from c-Myc degradation kinetics is displayed on the right hand. Results are from four biological replicates from two independent experiments. **$P = 0.0085$ (Student's paired $t$ test). Source data are available online for this figure.

accumulation in the absence of IL-2 stimulation and restored its levels at low pH, thus indicating that under acidic conditions c-Myc is mostly controlled at the mRNA level. Notably, despite a three- to fourfold increase in mRNA levels upon c-Myc overexpression as compared to endogenous *Myc* mRNA induction by IL-2, c-Myc protein levels remained almost identical to those promoted by IL-2. This suggests that the c-Myc degradation process becomes saturated upon its overexpression. Cell expansion at low pH (Fig. 6E) was not restored upon overexpression of c-Myc, even if TSC2 was also knocked out. At neutral pH, whereas, c-Myc overexpression enhanced cell expansion, but in combination with TSC2 knockout this effect was abrogated as a result of increased cell death.

## Low extracellular pH rapidly drives cytoplasmic and nuclear acidification in CTLs

Having thus far demonstrated that low pH blunts T-cell function including IL-2 mediated proliferation with modest impairment to IL-2/IL-2R binding and signaling but strong independent reduction in mTORC1 activity and c-Myc levels, we reasoned that there must be important intracellular acidification disruptive to multiple independent cellular processes. We sought an experimental approach for which it would be possible to distinguish acidification within the cytoplasm versus nucleus, a large organelle in CTLs. Briefly, we engineered T cells to express SEpHluorin/mCherry (Koivusalo et al, 2010), a fusion protein comprising SEpHluorin (a pH-sensitive mutant of GFP) and mCherry (pH-insensitive) and allowing one to infer intracytoplasmic and nuclear pH in parallel by ratiometric measurements via confocal microscopy by comparison with a standard curve (Fig. 7A). Interestingly, we observed that intracellular pH was rapidly (≤20 min) acidified (Fig. 7B) down to ~pH 7.0 (vs ~pH 7.3 in the control group; corresponding to a H+ increase of more than 80%) upon pH 6.6 treatment, and that it dropped to ~pH 6.90 (corresponding to a H+ increase of more than 140%) by the end of the experiment (4 h). Acidic conditions also lowered nuclear pH to the same extent and with the same kinetics

(Fig. 7C) despite that the nucleus had a slightly more alkaline pH at baseline ( + 0.1 pH unit). We questioned if low pH may directly impair the enzymatic activity of mTORC1, thereby accounting for lower downstream signaling. However, we found that the ability of mTORC1 complexes to phosphorylate purified 4E-BP1 was in fact higher in vitro at pH 6.6 than at pH 7.4 (Fig. 7D). Indeed, mTORC1 functions in close proximity to lysosomes which are highly acidic (pH 4.5–5) (Rogala et al, 2019) which may explain our observation.

## Acidity lowers intracellular glutamine/glutamate/ aspartate levels and promotes proline accumulation

Finally, we sought to explore differences in the metabolome of T cells under acidic conditions which could potentially play a role in impaired cellular function (Buck et al, 2017). Indeed, both mTORC1 and c-Myc are known regulators of amino acid homeostasis and can themselves be controlled by amino acid levels (Loftus et al, 2018; Wolfson and Sabatini, 2017). Interestingly, under acidic conditions, we observed a consistent decrease in the intracellular levels of glutamine/glutamate, along with an increase in proline content (Figs. EV4A and 8A). Since mTORC1 activation and c-Myc levels have been previously described to be modulated by glutamine (Jewell et al, 2015; Loftus et al, 2018), we assessed whether low glutamine levels could be causing the effects observed at low pH. We confirmed that glutamine deprivation from the medium mimics impaired CTL proliferation and mTORC1/c-Myc patterns observed under acidic conditions (Fig. EV4B,C). However, a glutamine dose-response experiment revealed that mTORC1/c-Myc patterns at low pH were matched at 1000-fold less (2 μM) glutamine in the culture medium (Fig. 8B) but merely a tenfold decrease (200 μM) in extracellular glutamine completely abrogated intracellular glutamine detection in CTLs (Fig. 8C). Plotting mTORC1 and c-Myc as a function of intracellular glutamine or glutamate (Fig. EV4D) further indicates that low intracellular glutamine/glutamate under acidic conditions does not dictate mTORC1/c-Myc patterns but rather could be a consequence of them. Whereas not detected in previous assays, it is

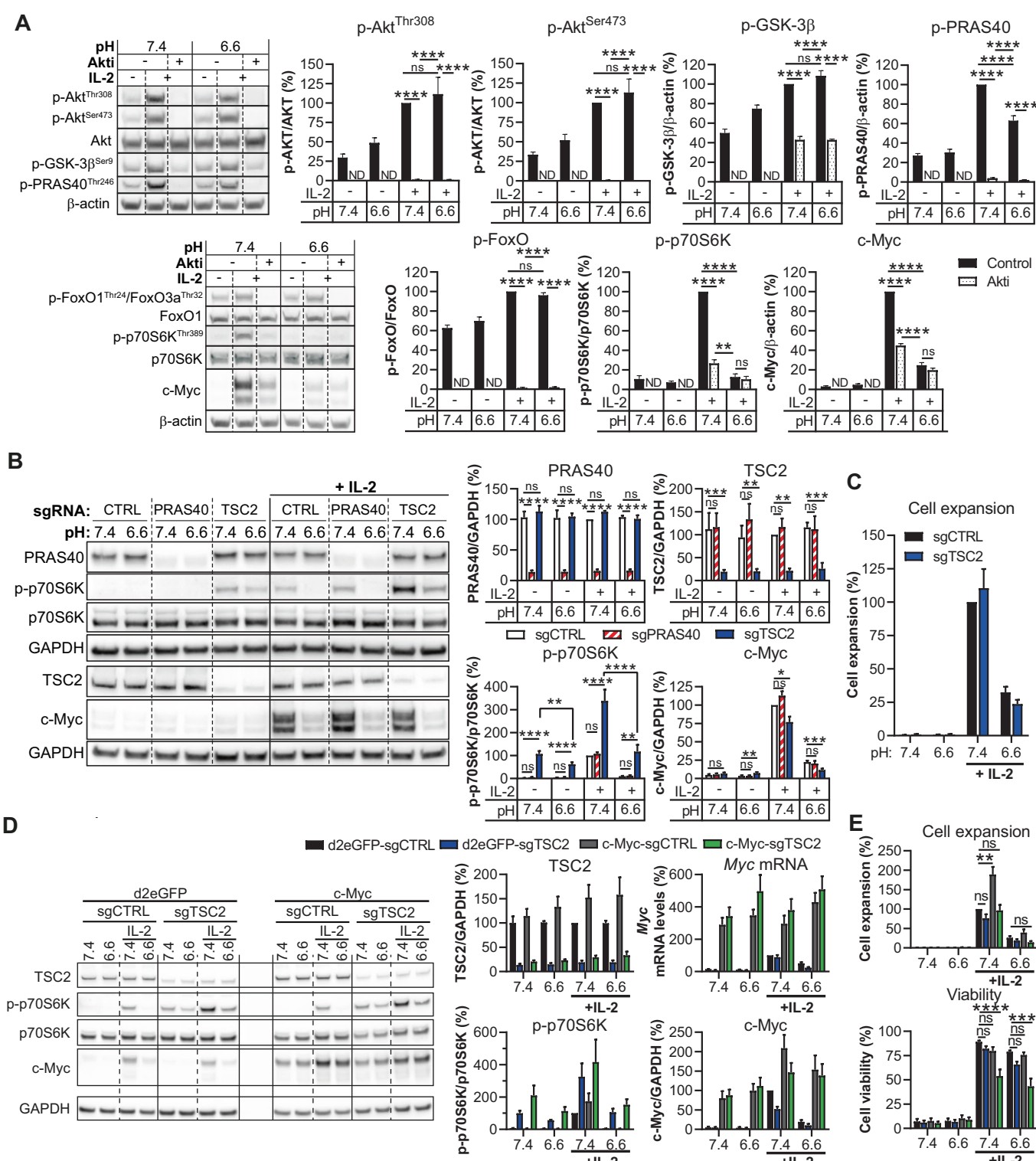

worth noting that intracellular aspartate content was also lowered by acidity and was dependent upon extracellular glutamine levels (Fig. 8C). Interestingly, we observed that increasing doses of extracellular glutamine lowered intracellular levels of serine and threonine (Appendix Fig. S10A) which might reflect CTL reliance on the neutral amino acid transporter alanine serine cysteine transporter 2 (ASCT2), a transporter which imports neutral amino acids such as alanine and glutamine (its primary role) in exchange for intracellular amino acids (Fuchs and Bode, 2005; Scalise et al, 2018).

Although glutamine is considered the major source of glutamate (Altman et al, 2016), proline and aspartate can arise from

**Figure 6. TSC2 knockout augments mTORC1 activity at low pH but does not restore CTL proliferation, even in combination with c-Myc overexpression.**

(A) Acidity does not inhibit phosphorylation of direct Akt targets. OT-I CTLs were cultured for 4 h in the presence, or absence ("Control", DMSO), of Akt1/2 inhibitor (10 μM- "Akti"). One representative western blot is shown. Bar graphs show the mean levels normalized to the condition "pH 7.4 + IL-2" +SEM of four biological replicates from two independent experiments. ND not determined. ns: not significant, **$P = 0.0029$, ****$P < 0.0001$ (one-way repeated measures ANOVA, Tukey post-hoc test). (B) TSC2 knockout improves mTORC1 activity. OT-I x CRISPR/Cas9 CTLs were transduced with retroviruses encoding a negative control, a PRAS40 or a TSC2 sgRNA, and were cultured for 4 h. One representative western blot from two membranes of the same samples is shown. Bar graphs show the mean phosphorylation status, or total levels, of the indicated molecule relative to the total protein of interest, or to GAPDH, normalized to the condition "pH 7.4 + IL-2 sgCTRL" +SEM of four biological replicates from two independent experiments. ns: not significant, ****$P < 0.0001$. TSC2: pH 7.4 sgTSC2 vs sgCTRL ***$P = 0.0005$, pH 6.6 sgTSC2 vs sgCTRL **$P = 0.0042$, pH 7.4 + IL-2 sgTSC2 vs sgCTRL **$P = 0.0026$, pH 6.6 + IL-2 sgTSC2 vs sgCTRL ***$P = 0.0007$. p-p70S6K: pH 6.6 sgTSC2 vs pH 7.4 sgTSC2 **$P = 0.001$, pH 6.6 + IL-2 sgTSC2 vs sgCTRL **$P = 0.0075$. c-Myc: pH 6.6 sgTSC2 vs sgCTRL **$P = 0.0011$, pH 7.4 + IL-2 sgTSC2 vs sgCTRL *$P = 0.0241$, pH 6.6 + IL-2 sgTSC2 vs sgCTRL ***$P = 0.0005$ (one-way repeated measures ANOVA, Tukey post-hoc test). (C) TSC2 knockout does not improve CTL proliferation. The methodology was the same as in (B). CTLs were cultured 4 days. Results show the relative expansion (normalized to the condition "pH 7.4 + IL-2 sgCTRL") +SEM of at least four biological replicates from at least two independent experiments. (D) TSC2 knockout and c-Myc overexpression improves mTORC1 activity and c-Myc levels. OT-I x CRISPR/Cas9 CTLs encoding a negative control protein (d2eGFP) or c-Myc, together with a negative control (d2eGFP-sgCTRL; c-Myc-sgCTRL) or a TSC2 sgRNA (d2eGFP-sgTSC2; c-Myc-sgTSC2) were cultured 4 h. One representative western blot is shown. Bar graphs show the mean levels normalized to the condition "pH 7.4 + IL-2 d2eGFP-sgCTRL" + SEM of four biological replicates from four independent experiments. (E) TSC2 knockout and c-Myc overexpression do not improve CTL expansion at low pH. The methodology was the same as in (D). CTLs were cultured 3 days. Results show the relative expansion (normalized to the condition "pH 7.4 + IL-2 d2eGFP-sgCTRL"), or viability, + SEM of four biological replicates from four independent experiments. ns not significant, **$P = 0.0063$, ***$P = 0.0002$, ****$P < 0.0001$ (one-way repeated measures ANOVA, Tukey post-hoc test). Source data are available online for this figure.

glutamine/glutamate conversion (Yoo et al, 2020). Thus, we next performed tracer experiments with ¹³C isotopic glutamine. Of note, within four hours upwards of 50% of some TCA cycle metabolites (e.g., citrate, fumarate, and malate) were generated from extracellular glutamine (Appendix Fig. S10B). Importantly, we confirmed that more than 80% of glutamate, more than 70% of aspartate, and at least 65% of the proline pool were derived from glutamine (Figs. EV4E and 8D). We questioned if the decrease in intracellular levels of glutamine/glutamate/aspartate could be due to increased conversion to proline. While a miR-based knockdown of more than 80% of P5CS, a rate-limiting enzyme involved in proline conversion (Fig. 8E), blocked an increase in proline levels at low pH, there was no rescue of glutamine/glutamate/aspartate levels (Fig. 8F), mTORC1/c-Myc patterns (Fig. 8G), or cell proliferation and viability (Fig. EV4F). Upon P5CS knockdown there was, however, an increase in alanine accumulation (Fig. 8F). Finally, utilizing tritiated glutamine, we determined that its uptake was pH-dependent (Fig. 8H). Notably, the assay had to be performed using complete medium at 37 °C (instead of HBSS at room temperature) (Cormerais et al, 2018) in order to observe a clear impact, thus suggesting that acidity disturbs amino acid uptake and/or export via competition for transporters. All perturbations to CTLs at low pH are summarized in a graphical abstract in Fig. EV5.

## Discussion

Here, we have comprehensively examined the impact of acidity, a common suppressive feature of solid tumors, on effector CD8+ T cells. It has been previously shown that the cytolytic capacity of CTLs is blunted at low pH (Fischer et al, 2007; Nakagawa et al, 2015) and we confirmed this to be the case for T cells expressing weak but not high-affinity TCRs. In line with previous studies, we also observed reduced cytokine secretion (IFN-γ, IL-2 and TNF) and proliferation upon CTL re-activation under acidic conditions. Several mechanisms have been proposed to explain how acidity impairs CTLs including CD3ζ downregulation, lower CD25 (IL-2Rα) levels, upregulation of CTLA-4 and PD-1, lower

phosphorylation of Akt, ERK, STAT5, p38 and JNK, and/or activation of proton-sensing receptors (e.g., ASICs, TDGA8, OGR1, TRPV) (Bosticardo et al, 2001; Brand et al, 2016; Calcinotto et al, 2012; Mendler et al, 2012; Nakagawa et al, 2015; Pilon-Thomas et al, 2016), but previous studies are awash with contradictory findings.

To gain deeper mechanistic insight into acidity-induced T-cell dysfunction, we considered CTL re-activation in two stages, a first TCR/CD3-dependent step during which cytokines are generated, and a second one that relies mainly upon autocrine IL-2 production and IL-2R signaling. We found that although acidity inhibited IL-2 production during CTL re-activation, impaired proliferation was mostly due to diminished IL-2 responsiveness in a TCR/CD3-independent manner. We observed that levels of the IL-2R subunits (mostly β and γ), functional IL-2R complex, IL-2 binding capacity, and the phosphorylation of upstream signaling molecules (i.e., JAK1, JAK3 and STAT5) dropped under acidic conditions. However, at pH 6.6, IL-2R signaling was only modestly impacted, and the MAPK/ERK pathway not at all. In contrast, at pH 6.6 we observed that both mTORC1 activation/signaling (as evaluated by phosphorylation of p70S6K, S6, 4E-BP1 and ULK1) and c-Myc accumulation were reduced, apparently independently of one other and of a global decrease in IL-2R signaling.

Notably, it has been recently reported that acidity preserves stemness (Cheng et al, 2023), in line with the implicated roles of IL-2R signaling, c-Myc and mTORC1 in effector cell versus stem cell/memory differentiation (Ross and Cantrell, 2018; Verbist et al, 2016), all of which were found to be perturbed in our study investigating changes in CTLs under acidic conditions. The impact of low pH on T cells may be even more pronounced in the presence of lactate (Feng et al, 2022), a metabolite often secreted at the same time as protons, and is of interest to explore in future studies.

mTOR is a central hub integrating growth factor signaling and nutrient availability to regulate critical cellular processes including protein and nucleotide synthesis, and ultimately T-cell fate (Saxton and Sabatini, 2017). Interestingly, although the mTORC1 inhibitor rapamycin (Pollizzi and Powell, 2015; Ross and Cantrell, 2018) blunted CTL proliferation (i.e., suggesting that impaired proliferation at low pH is associated with decreased mTORC1 activity), we

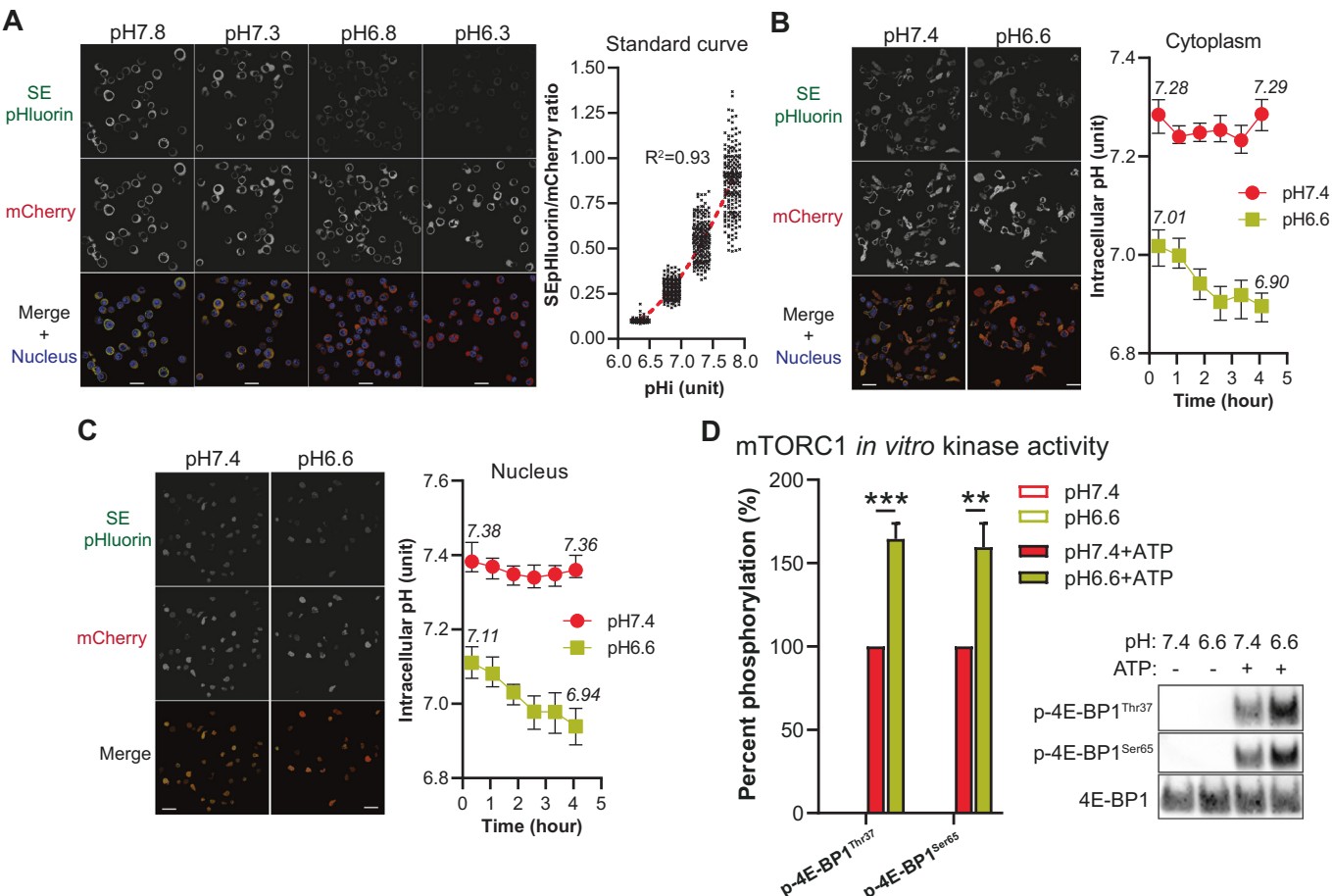

**Figure 7. Low extracellular pH rapidly drives cytoplasmic and nuclear acidification in CTLs.**

(A) Standard curve to detect intracellular pH. OT-I CTLs engineered to express SEpHluorin-mCherry were cultured in high $K^+$ buffer containing Valinomycin/Nigericin with defined pH, allowing to obtain equilibrium between extracellular and intracellular pH. Dot plot graph shows the ratio of fluorescence of SEpHluorin/mCherry per cell excluding nucleus at each pH acquired by live confocal microscopy. A correlation curve was obtained using the median fluorescence for each pH and was used as a standard curve. Representative images for each pH with a 20 μm scale bar are displayed. (B) Extracellular acidity lowers intracytoplasmic pH. OT-I CTLs engineered to express SEpHluorin-mCherry and cultured at pH 7.4 or pH 6.6 in the presence of exogenous murine IL-2 were acquired by live confocal microscopy. Line graph shows the median ratio of fluorescence of SEpHluorin/mCherry per cell excluding nucleus ± 95% CI of at least 150 cell per time-point per pH from one out of two independent experiments. Representative images for each pH with a 20 μm scale bar are displayed. (C) Extracellular acidity lowers nuclear pH. The methodology is the same as in (B). Line graph shows the median ratio of fluorescence of SEpHluorin/mCherry per cell excluding extra-nucleus signal ± 95% CI of at least 150 cell per time-point per pH from one out of two independent experiments. Representative images for each pH with a 20 μm scale bar are displayed. (D) mTORC1 kinase activity in vitro is not lowered at pH 6.6. Kinase activity of recombinant mTORC1 complexes (mTOR/RAPTOR/MLST8) at pH 7.4 or pH 6.6 was assessed by determining phosphorylation status of recombinant 4E-BP1 upon a 10 min reaction at 30 °C in the presence, or absence, of ATP. One representative Western blot is shown. Results show the mean phosphorylation status of 4E-BP1 normalized to the condition "pH 7.4 + ATP" + SEM of four independent experiments. **$P = 0.0058$, ***$P = 0.0004$ (Student's *t* test). Source data are available online for this figure.

observed that Akt activation (a kinase which can activate mTORC1) and its downstream targets including GSK-3β, FoxO1 and PRAS40 were not disrupted under acidic conditions. GATOR1 (involved in amino acid sensing) and Lkb1/AMPK pathways were similarly not involved in the reduced mTORC1 activation at low pH. Moreover, we observed no phosphorylation or accumulation of the transcription factors CREB and CREM, suggesting that proton-sensing receptors, which often signal through the cAMP/PKA pathway and can lead to mTORC1 inhibition (Damaghi et al, 2013; Jewell et al, 2019), are not involved in reduced mTORC1 activity under acidic conditions. Similarly, Wu et al. recently ruled out participation of several proton-sensing receptors in acidity-mediated suppression of T cells (Wu et al, 2020).

The inhibition of mTORC1 at low pH has been described in fibroblasts and tumor cells (Balgi et al, 2011; Faes et al, 2016; Walton et al, 2018) and it has been proposed that acidification within the cytoplasm causes lysosome dispersion thereby preventing mTORC1 co-localization with its activator Rheb (Walton et al, 2018). We observed that CTL culture at low pH causes rapid acidification within both the cytoplasmic and nuclear compartments. However, the fact that TSC2 (a Rheb inhibitor) knockout promoted mTORC1 activity argues against lysosome dispersion taking place in CTLs. While TSC2 knockout enforced mTORC1 activity at low pH, this was not sufficient to restore CTL proliferation upon IL-2 stimulation. Finally, we questioned if low pH could directly inhibit mTORC1, but the kinase activity of the

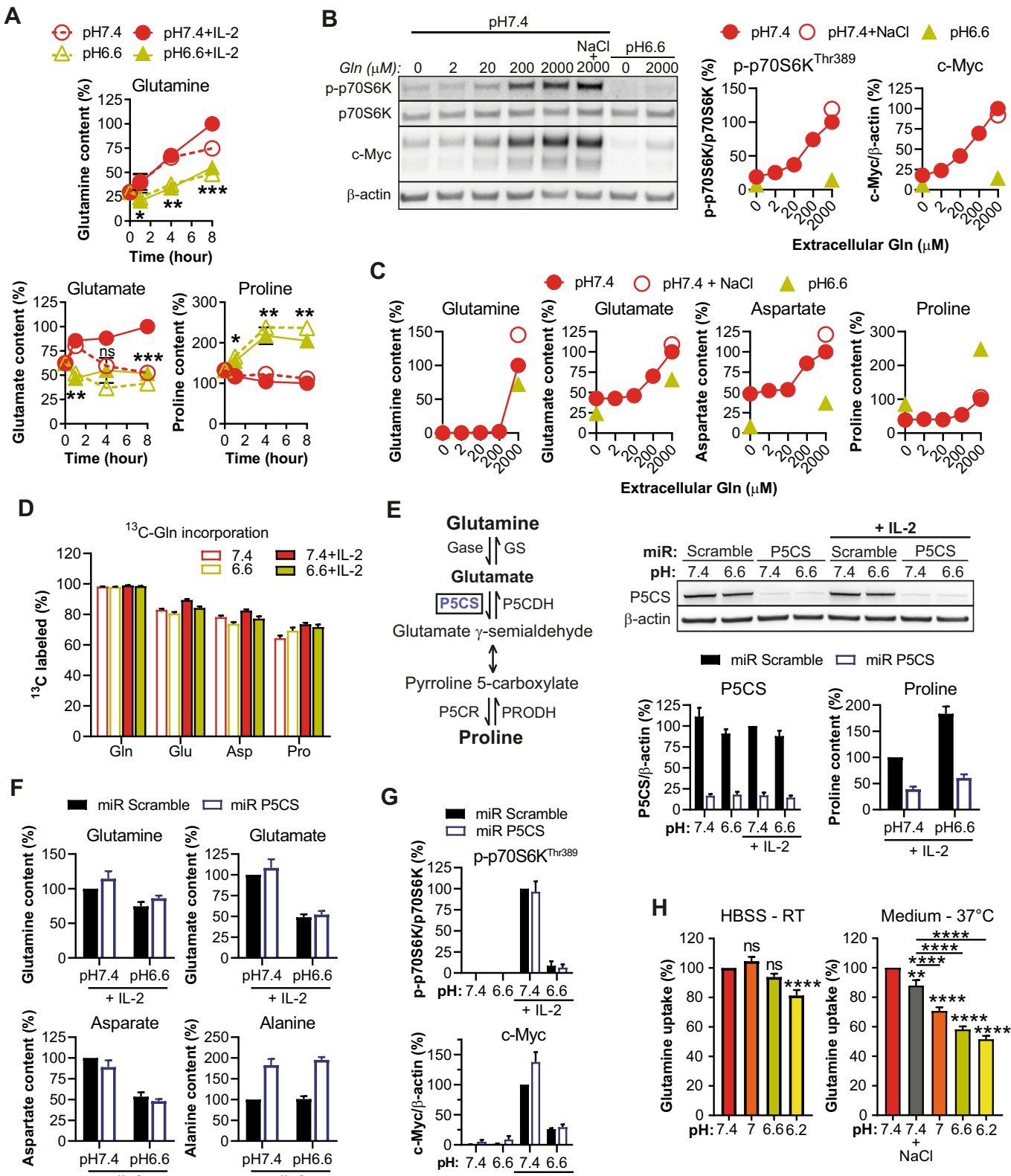

purified complex in vitro was not disrupted at pH 6.6. Hence, we conclude that along with its well-known roles in sensing nutrient and energy sufficiency, our data suggest that mTORC1 also acts as a sensor of pH (within just minutes of CTL exposure to an acidic milieu), an important parameter for integration in order to adapt cellular physiology to the microenvironment.

The transcription factor c-Myc plays critical roles during T-cell activation and IL-2-mediated proliferation, it regulates metabolic

**Figure 8. Acidity lowers intracellular glutamine/glutamate/aspartate levels and promotes proline accumulation in CTLs.**

(A) Time-course of intracellular levels of glutamine, glutamate, and proline in OT-I CTLs. Results show the mean intracellular level of the amino acid of interest (normalized to the condition "pH 7.4 + IL-2 8 h") ± SEM of four biological replicates from two independent experiments. ns: not significant. Gln: pH 6.6 + IL-2 vs pH 7.4 + IL-2 1 h *$P$ = 0.0498, 4 h **$P$ = 0.0015, 8 h ***$P$ = 0.0001. Glu: pH 6.6 + IL-2 vs pH 7.4 + IL-2 1 h **$P$ = 0.0066, 4 h ns $P$ = 0.0672, 8 h ***$P$ = 0.0008. Pro: pH 6.6 + IL-2 vs pH 7.4 + IL-2 1 h *$P$ = 0.0178, 4 h **$P$ = 0.0085, 8 h **$P$ = 0.0017 (Student's paired $t$ test). (B) Dose-dependent impact of glutamine on phosphorylation of p70S6K and c-Myc levels. OT-I CTLs were cultured for 4 h with exogenous murine IL-2 and in presence or absence of NaCl 31 mM. One representative western blot out of four biological replicates from two independent experiments is shown. Line graphs display the mean relative levels ± SEM of the pooled data, as a function of extracellular glutamine content. (C) Dose-dependent impact of exogenous glutamine on amino acid levels. The methodology was the same as in (B). Results show the mean intracellular amino acid content normalized to the condition "pH 7.4 + 2000 µM Gln" ± SEM of four biological replicates. (D) Intracellular glutamine, glutamate, aspartate and proline are coming from extracellular glutamine. OT-I CTLs were cultured for 4 h with isotopic $^{13}$C-glutamine (2 mM). Results show the mean proportion of the indicated intracellular amino acid that incorporated $^{13}$C + SEM of four biological replicates from two independent experiments. (E) Knockdown of P5CS prevents proline accumulation. OT-I CTLs transduced with a control miR (Scramble) or a P5CS-targeting miR were cultured for 4 h. One representative western blot displaying P5CS protein levels is shown. Bar graph on the left hand displays the mean P5CS protein levels normalized to the condition "pH 7.4 + IL-2 miR Scramble" + SEM of four biological replicates from two independent experiments. Bar graph on the right hand shows proline levels normalized to the condition "pH 7.4 + IL-2 miR Scramble" + SEM of four biological replicates from two independent experiments. (F) P5CS knockdown does not improve glutamine/glutamate/aspartate levels at low pH, but increases alanine accumulation. The methodology was the same as in (E). Results show the mean amino acid level normalized to the condition "pH 7.4 + IL-2 miR Scramble" + SEM of four biological replicates from two independent experiments. (G) P5CS knockdown does not improve mTORC1 activity and c-Myc levels at low pH. The methodology was the same as in (E). Results show the mean levels normalized to the condition "pH 7.4 + IL-2 miR Scramble" + SEM of four biological replicates from two independent experiments. (H) Acidity lowers glutamine uptake in CTLs. OT-I CTLs were cultured for 20 min either in HBSS at room temperature, or in CTL medium with IL-2 at 37 °C. Amino acid uptake assay was carried out by adding tritiated glutamine at 10 µCi/mL (equivalent to 165 nM). Results show the mean glutamine uptake normalized to the condition "pH 7.4" + SEM of eight biological replicates from at least three independent experiments. ns: not significant, **$P$ = 0.0019, ****$P$ < 0.0001. Comparisons to pH 7.4 are performed when not indicated. (one-way repeated measures ANOVA, Tukey post-hoc test). Source data are available online for this figure.

programs including glutaminolysis, and its inactivation prevents the proliferation of T cells (Gnanaprakasam and Wang, 2017; Marchingo et al, 2020; Wang et al, 2011). Under acidic conditions we observed a slight decrease in IL-2/IL-2R binding and in JAK/STAT activation, and a consequent decrease in *Myc* transcription. Interestingly, we also found that c-Myc half-life was lowered at low pH. Accordingly, although we did not observe changes in phosphorylation of T58 or S62 (the residues mostly commonly involved in driving proteasomal degradation or stabilization of c-Myc, respectively, (Farrell and Sears, 2014), we observed higher c-Myc accumulation upon proteasome inhibition. Our data indicate that acidity might improve proteasome activity, as previously proposed by others (Rackova and Csekes, 2020; Zund et al, 1997). Notably, while both transcriptional and post-transcriptional mechanisms regulated c-Myc levels, c-Myc over-expression was sufficient to restore its levels at low pH. Although this suggests a major role of *Myc* mRNA transcription for controlling protein levels at low pH, it does not rule out the importance of a lowered protein half-life. Indeed, c-Myc over-expression may lead to a highly active and potentially saturated c-Myc degradation process, as suggested by the high mRNA-to-protein c-Myc ratio observed, thus negating the impact of lowered c-Myc half-life. Nevertheless, despite significantly enhancing CTL proliferation at neutral pH, c-Myc overexpression was not sufficient to restore CTL proliferation at low pH.

As observed during IL-2R signaling, acidity lowered mTORC1 activity and c-Myc levels upon re-activation of CTLs. Notably, in contrast to mTORC1, we found that lowered c-Myc levels were related to a higher re-activation threshold under acidic conditions and could be compensated by increasing the activation stimulus. Given the multifunctional roles of both c-Myc and mTORC1 including in protein synthesis which is needed for the generation of cytokines, granzymes and perforin, etc., it is likely that impaired CTL effector function at low pH is at least in part due to their perturbation. Further work is warranted to delineate the precise mechanisms by which c-Myc and mTORC1 are inhibited under acidic conditions.

Finally, we questioned if low pH could cause changes in amino acid metabolism. Indeed, amino acids play a crucial role in T-cell function and they are tightly connected to mTORC1 and c-Myc activities. Under acidic conditions we observed a drop in intracellular levels of glutamine/glutamate/aspartate whereas proline accumulated. Glutamine is a nonessential amino acid but can be conditionally essential under catabolic conditions, and its uptake and catabolism are highly induced in active T cells to provide intermediate molecules for different pathways of biosynthesis as well as substrates for the mitochondria (Newsholme et al, 1985). For example, during glutaminolysis its carbon backbone can be converted to α-ketoglutarate to maintain homeostasis of the tricarboxylic-acid cycle (TCA), or to lactate that generates nicotinamide adenine dinucleotide (NAD) and NADPH (DeBerardinis et al, 2007). Notably, insufficient glutamine can inhibit T-cell proliferation, growth and cytokine production (Hammami et al, 2012).

Using isotopic glutamine, we determined that glutamate, aspartate and proline in CTLs are mostly derived from extra-cellular glutamine. We also found that glutamine deprivation in the culture media recapitulated the attenuated CTL proliferation and mTORC1/c-Myc patterns generated under acidic conditions. However, while a tenfold decrease in extracellular glutamine concentration resulted in the same intracellular levels measured at low pH, the impact on mTORC1 and c-Myc was only marginal. We observed that glutamine uptake by CTLs was lower under acidic conditions, at least in part due to competition with other amino acids for import. We also found that increased glutamine levels in the culture media was associated with lower intracellular levels of serine and threonine suggesting that low pH may disrupt substrate specificity and/or activity of ASCT2 (SLC1A5) (Fuchs and Bode, 2005; Scalise et al, 2018). Taken together, our results suggest that lower levels of glutamine/glutamate/aspartate at low pH are consequences of either disturbed mTORC1/c-Myc status (especially considering c-Myc is a known regulator of glutamine metabolism) and/or of the activity of glutamine transporters.

Proline accumulation is a conserved stress response mechanism across many organisms including plants, bacteria, protozoa and marine invertebrates. In plants, for example, drought, cold, radiation, hyperosmolarity or pH changes drive proline build-up which can serve as an osmoprotectant and to quench reactive oxygen species (ROS) (Hayat et al, 2012; Szabados and Savoure, 2010). The role of elevated proline in CTLs, if any, is unclear based on our data. Indeed, the knockdown of P5CS (an enzyme involved in glutamate conversion to proline) prevented proline accumulation but it did not restore glutamine/glutamate/aspartate levels and there was no impact on mTORC1, c-Myc, CTL proliferation or viability.

We set out in this work with the aim of comprehensively elucidating the mechanism(s) by which acidity, a common suppressive feature of tumors, dampens T-cell function (Lanitis et al, 2017). While previous studies have demonstrated improved tumor control with proton pump inhibitors and bicarbonate (Calcinotto et al, 2012; Pilon-Thomas et al, 2016), such approaches may not be universally effective as tumor cells might also benefit from alkalization. Here, we have demonstrated that low pH causes profound disruptive changes in CTLs affecting responsiveness to IL-2, different intracellular signaling pathways, transcription factors, and amino acid metabolism, which all together severely impair proliferation, cytokine production and, for lower affinity TCR T cells, cytotoxicity. Importantly, we show that concomitantly improving mTORC1 activity and c-Myc levels by gene engineering is not sufficient to restore CTL proliferation under acidic conditions. In summary, we conclude that aside from developing a strategy that can specifically and effectively restore intracellular CTL pH, multiple interventions will be required to enable T cells to overcome suppression caused by acidity in the TME.

# Methods

### Reagents and tools table

| Reagent/ resource | Reference or source | Identifier or catalog number |
|---|---|---|
| **Experimental models** | | |
| 293T cells (*Homo sapiens*) | ATCC | CRL-3216 |
| C1498 cells (*M. musculus*) | ATCC | TIB-49 |
| CTLL-2 (*M. musculus*) | ATCC | TIB-214 |
| C57BL/6 J (*M. musculus*) | Harlan | N/A |
| OT-I (*M. musculus*) | Charles River | N/A |
| OT-3 (*M. musculus*) | Enouz et al (2012) | Prof. Dietmar Zehn, TUM, Germany |
| OT-I x CRISPR/Cas9 (*M. musculus*) | This study | N/A |
| **Recombinant DNA** | | |
| MSGV2-Thy1.1-2A-Rheb | This study | N/A |

| Reagent/ resource | Reference or source | Identifier or catalog number |
|---|---|---|
| MSGV2-Thy1.1-2A-mCherry-SEpHluorin | This study | N/A |
| RSV-U6-sgCD25-PGK-Thy1.1 | This study | N/A |
| RSV-U6-sgCTRL-PGK-Thy1.1 | This study | N/A |
| RSV-U6-sgCTRL-PGK-d2eGFP-IRES-Thy1.1 | This study | N/A |
| RSV-U6-sgCTRL-PGK-c-Myc-IRES-Thy1.1 | This study | N/A |
| RSV-U6-sgLkb1-PGK-Thy1.1 | This study | N/A |
| RSV-U6-sgNprl2-PGK-Thy1.1 | This study | N/A |
| RSV-U6-sgPras40-PGK-Thy1.1 | This study | N/A |
| RSV-U6-sgTsc2-PGK-Thy1.1 | This study | N/A |
| RSV-U6-sgTsc2-PGK-d2eGFP-IRES-Thy1.1 | This study | N/A |
| RSV-U6-sgTsc2-PGK-c-Myc-IRES-Thy1.1 | This study | N/A |
| pCL-Eco | Addgene | # 12371 |
| pQG-U6L-miRCTRL-PGK-Thy1.1 | This study | N/A |
| pQG-U6L-miRP5CS-PGK-Thy1.1 | This study | N/A |
| **Antibodies** | | |
| Goat anti-mouse IgG HRP | Thermo Fisher | A16072 |
| Donkey anti-rabbit IgG HRP | Thermo Fisher | 31458 |
| Rabbit anti-Akt | CST | 4691 |
| Rabbit anti-p-Akt_Ser | CST | 4060 |
| Rabbit anti-p-Akt_Thr | CST | 13038 |

| Reagent/resource | Reference or source | Identifier or catalog number |
|---|---|---|
| Rabbit anti-ALDH18a1/P5CS | Sigma-Aldrich | HP008333 |
| Rabbit anti-AMPKα | CST | 5831 |
| Rabbit anti-AMPKβ | CST | 4150 |
| Rabbit anti-p-AMPKα | CST | 2535 |
| Rabbit anti-p-AMPKβ | CST | 4186 |
| Mouse-anti-β-Actin | Santa Cruz | sc-47778 |
| Rabbit anti-c-Myc | CST | 5605 |
| Rabbit anti-p-c-MycS62 | CST | 13748 |
| Rabbit anti-p-c-MycT58 | Abcam | ab28842 |
| Rabbit anti-CREB | Bethyl | A301-669A |
| Rabbit anti-p-CREB | CST | 9198 |
| Mouse anti-CREM | Santa Cruz | sc-390426 |
| Rabbit anti-p-eIF2α | CST | 3398 |
| Rabbit anti-ERK1/2 | CST | 4695 |
| Rabbit anti-p-ERK1/2 | CST | 4377 |
| Rabbit anti-FoxO1 | Thermo Fisher | MA5-32114 |
| Rabbit anti-p-FoxO1/3 | CST | 9464 |
| Mouse anti-GAPDH | Santa Cruz | sc-32233 |
| Rabbit anti-p-GSK-3β | CST | 9323 |
| Rabbit anti-JAK1 | CST | 3344 |
| Rabbit anti-JAK3 | CST | 8863 |
| Rabbit anti-p-JAK1 | CST | 74129 |
| Rabbit anti-p-JAK3 | CST | 5031 |
| Rabbit anti-Lkb1 | CST | 3047 |
| Rabbit anti-Nprl2 | CST | 37344 |
| Rabbit anti-p62 | CST | 5114 |
| Rabbit anti-PLCγ1 | CST | 2822 |

| Reagent/resource | Reference or source | Identifier or catalog number |
|---|---|---|
| Rabbit anti-p-PLCγ1 | CST | 2821 |
| Rabbit anti-PRAS40 | CST | 2691 |
| Rabbit anti-p-PRAS40 | CST | 2997 |
| Rabbit anti-p-p38 | CST | 4511 |
| Rabbit anti-p62 | CST | 5114 |
| Rabbit anti-p70S6K | CST | 9202 |
| Rabbit anti-p-p70S6K | CST | 9205 |
| Rabbit anti-Rheb | CST | 13879 |
| Rabbit anti-STAT5 | Bethyl | 9359 |
| Rabbit anti-p-STAT5 | CST | A303-494A |
| Mouse anti-S6 | Santa Cruz | sc-74459 |
| Rabbit anti-p-S6 | CST | 4858 and 5364 |
| Rabbit anti-SLP-76 | CST | 4958 |
| Rabbit anti-p-SLP-76 | CST | 92711 |
| Rabbit anti-TSC2 | CST | 4308 |
| Rabbit anti-ULK1 | CST | 8054 |
| Rabbit anti-p-ULK1 | CST | 6888 |
| Rabbit anti-4E-BP1 | CST | 9644 |
| Rabbit anti-p-4E-BP1_Ser | CST | 9451 |
| Rabbit anti-p-4E-BP1_Thr | CST | 2855 |
| Hamster anti-CD3 | Biolegend | 100331 |
| Hamster anti-CD28 | Biolegend | 302934 |
| Rat anti-CD8α-AF488 | Biolegend | 100723 |
| Rat anti-CD8α-PECy7 | Biolegend | 100722 |
| Rat anti-CD8α-BV421 | Biolegend | 100738 |
| Rat anti-CD25-PECy7 | Biolegend | 102016 |
| Rat anti-CD25-APC | Biolegend | 102012 |
| Rat anti-CD44-PECy7 | Biolegend | 103030 |

| Reagent/resource | Reference or source | Identifier or catalog number |
| --- | --- | --- |
| Rat anti-CD62L-BV570 | Biolegend | 104433 |
| Mouse anti-CD90.1/Thy1.1-APC | Biolegend | 202526 |
| Mouse anti-CD90.1/Thy1.1-Biotin | Biolegend | 202510 |
| Mouse anti-CD90.1/Thy1.1-BV421 | Biolegend | 202529 |
| Rat anti-CD122-BV421 | BD Biosciences | 564925 |
| Rat anti-CD132-APC | Biolegend | 132308 |
| Mouse anti-GrB-AF647 | Biolegend | 515406 |
| Goat anti-streptavidin-biotin | Vector laboratories | BA-0500 |
| **Oligonucleotides and other sequence-based reagents** | | |
| Actb-F | This study | CTAAGGCCAACCGTGAAAAGAT |
| Actb-R | This study | CACAGCCTGGATGGCTACGT |
| Myc-F | This study | TTGATGTGGTGTCTGTGGAGAAGAG |
| Myc-R | This study | CGTAGTTGTGCTGGTGAGTGGA |
| sgCTRL | This study | AAACCTAGCGTAGATTCGGC |
| sgCD25 | This study | AACCCCAACATCAGCAAGCG |
| sgLkb1 | This study | TCCTTAGCGCCCTACGTATA |
| sgNprl2 | This study | GCAGAGGCGGCCGTACCAAT |
| sgPRAS40 | This study | ACGACATCGCACAGGCGCAC |
| sgTSC2 | This study | TCTCATACACTCGAGTGGCG |
| shCTRL | This study | CAGGCAGAAGTATGCAAAGCA |
| shP5CS | This study | TCGACATGTAATTTCATTTCT |
| **Chemicals, enzymes and other reagents** | | |
| DMEM high glucose | Thermo Fisher | 31966021 |
| DMEM low glucose | Thermo Fisher | 21885025 |
| DMEM low glucose, no glutamine | Thermo Fisher | 11880028 |
| DMEM low glucose, no carbonate | Sigma-Aldrich | D5523 |
| Fetal Bovine Serum | Sigma-Aldrich | F7524 |
| Dialyzed fetal bovine serum | Thermo Fisher | 26400044 |
| Penicillin-streptomycin | Thermo Fisher | 15140122 |
| HEPES | Bioconcept | 5-31F00-H |
| MES | Sigma | M1317 |

| Reagent/resource | Reference or source | Identifier or catalog number |
| --- | --- | --- |
| Glutamine | Bioconcept | 5-10K00-H |
| Non-essential amino acids | Thermo Fisher | 11140035 |
| Phenol red | Sigma | P0290 |
| Glutamax | Thermo Fisher | 35050061 |
| HBSS | Thermo Fisher | 24020117 |
| Rapamycin | Sigma-Aldrich | 553211 |
| MG-132 | Sigma-Aldrich | 474791 |
| Cycloheximide | Sigma-Aldrich | C4859 |
| CHIR99021 | Sigma-Aldrich | SML1046 |
| Akt1/2 inhibitor | Sigma-Aldrich | A6730 |
| Mouse IL-2 | Peprotech | 212-12 |
| Human IL-2 | Peprotech | 200-02 |
| Ficoll-Paque Plus | Cytiva | 17-1440.02 |
| T cell TrasAct | Miltenyi | 130-111-160 |
| Retro-concentin | SBI | RV100A |
| Protamine sulfate | Sigma-Aldrich | P4020 |
| Retronectin | Takara | T100B |
| CellTrace Violet | Thermo Fisher | C34557 |
| Precision count beads | Biolegend | 424902 |
| Live/dead fixable near-IR dead cell stain | Thermo Fisher | L34976 |
| Hoechst 34580 | Thermo Fisher | H21486 |
| Nigericin and valinomycin | Thermo Fisher | P35379 |
| RNAse A | Sigma-Aldrich | 10109142001 |
| Propidium iodide | Thermo Fisher | P3566 |
| Streptavidin-PE | Biolegend | 405204 |
| Rosetta 2 - Novagen competent cells | Merck | 71397 |
| AnnexinV-FITC | BD Biosciences | 556547 |
| Kapa Sybr Fast Rox Low | Roche | KK4621 |
| Halt phosphate/protease inhibitors | Thermo Fisher | 78440 |
| RIPA lysis buffer | Thermo Fisher | 89900 |

| Reagent/resource | Reference or source | Identifier or catalog number |
|---|---|---|
| LDS Sample Buffer | Thermo Fisher | B0007 |
| Sample reducing agent | Thermo Fisher | B0009 |
| L-Glutamine-13C5 | Sigma-Aldrich | 605166 |
| Glutamine, L-[2,3,4-3H] | Hartmann Analytic | ART0149 |
| Microscint-40 | PerkinElmer | 6013641 |
| mTOR/RAPTOR/MLST8 | Sigma-Aldrich | SRP0364 |
| Recombinant Human 4EBP1 | Abcam | ab89849 |
| **Software** | | |
| FlowJo | https://www.flowjo.com/ | N/A |
| GraphPad Prism 8 | https://www.graphpad.com/ | N/A |
| ImageJ | https://imagej.net/ | N/A |
| **Other** | | |
| Mouse CD8 + T cell isolation kit | Miltenyi | 130-104-075 |
| EasySep human CD8 + T cell isolation kit | STEMCELL | 17953 |
| CELLection Biotin Binder Kit | Thermo Fisher | 11533D |
| Mouse IFN-γ ELISA MAX | Biolegend | 430802 |
| Mouse IL-2 ELISA MAX | Biolegend | 431002 |
| Mouse TNF-α ELISA MAX | Biolegend | 430902 |
| Labtek-I | Nunc | 055083 |
| EZ-Link Sulfo-NHS-LC-Biotin kit | Thermo Fisher | 21327 |
| Primescript first strand cDNA synthesis kit | Takara | 6110A |
| RNeasy kit | Qiagen | 74104 |
| DNase set | Qiagen | 79254 |
| Bolt 4-12% Bis-Tris Plus Gels | Thermo Fisher | NW04127BOX |
| NuPAGE 4-12% Bis-Tris Midi gels | Thermo Fisher | WG1403BOX |

| Reagent/resource | Reference or source | Identifier or catalog number |
|---|---|---|
| iBlot 2 Transfer Stacks PVDF | Thermo Fisher | IB24002 and IB24001 |
| 7900 HT Fast real-time PCR | Applied Biosystems | N/A |
| CytoFlex | Beckman Coulter | N/A |
| FACSCanto | BD Biosciences | N/A |
| Gallios | Beckman Coulter | N/A |
| iBlot 2 dry blotting system | Thermo Fisher | N/A |
| Fusion FX imager | Vilber Lourmat | N/A |
| LSM 800 confocal microscope | Zeiss | N/A |
| pH meter Orion A Star A111 | Thermo Fisher | N/A |
| Scintillation counter TopCount NXT | PerkinElmer | N/A |
| Spectramax M3 | Molecular Devices | N/A |

## Methods and protocols

### Mice

Mice were housed at the University of Lausanne (UNIL, Epalinges, Switzerland) animal facility. All experiments were conducted in accordance and approval from the Service of Consumer and Veterinary Affairs (SCAV) of the Canton of Vaud. Female C57BL/6 were purchased from Harlan (Harlan, Netherlands). OT-I (Charles River) and OT-3 mice (kindly provided by Prof. Dietmar Zehn) carry a TCR transgene specific for ovalbumin. Homozygous OT-I x CRISPR/Cas9 mice were obtained upon crossing OT-I with B6J Rosa26-Cas9 mice (The Jackson Laboratory).

### Cell lines

The 293T and C1498 cell lines (ATCC) were grown in DMEM medium containing 4.5 g/L glucose, sodium pyruvate, and glutamax (Gibco), supplemented with 10% heat-inactivated fetal bovine serum (Sigma-Aldrich), 10 mM HEPES, 50 U/mL penicillin, and 50 µg/mL streptomycin (Gibco). The CTLL-2 cell line (ATCC) was grown in the same medium as primary CTLs and was used to titrate the human IL-2 "switch" variant.

### Splenocyte isolation, CD8+ selection and CTL generation

Splenocytes were extracted from OT-I, OT-3, or C57BL/6 mice. CD8+ T cells were purified by negative selection using the Miltenyi CD8+ T cell isolation kit (130-104-075), and were primed by culture on plates pre-coated with anti-CD3 antibodies (5 µg/mL;

clone 145-2C11, Biolegend) in the presence of anti-CD28 antibodies (1 μg/mL; clone 37.51, Biolegend) and recombinant murine IL-2 (rmIL-2, 200 IU/mL; Peprotech) at 37 °C with 5% $CO_2$. Every two or three days, cells were diluted, and expanded until nine to twelve days with fresh medium containing 200 IU/mL rmIL-2. During most CTL assays with IL-2, and when not provided, the dose used is 200 IU/mL.

PBMCs were extracted from buffy coats of healthy donors (Transfusion Interregionale CRS, Switzerland) using Ficoll-Paque Plus (Cytiva) density gradient centrifugation and CD8+ T cells were negatively selected using EasySep human CD8 T+ cell isolation kit (STEMCELL Technologies). CD8+ T cells were activated for three days using CD3/CD28 -stimulation reagent (T Cell TransAct, Miltenyi Biotec), and expanded for four days using 2000 IU/mL recombinant human IL-2 (Peprotech).

### CTL culture to analyze pH impact
Most of the experiments were carried out in DMEM, containing 1 g/L glucose, sodium pyruvate and Glutamax (Gibco), supplemented with 10% heat-inactivated fetal bovine serum, non-essential amino acids (Gibco), penicillin-streptomycin (Gibco), and 50 μM β-mercaptoethanol. When needed (e.g. for glutamine deprivation experiments), DMEM containing 1 g/L glucose and sodium pyruvate (Gibco), supplemented with 10% dialyzed and heat-inactivated fetal bovine serum (Gibco), penicillin-streptomycin, phenol red (Sigma-Aldrich) and 50 μM β-mercaptoethanol was used. Where specified, medium was complemented with 2 mM glutamine (Bioconcept) or glutamax (Gibco). The medium was supplemented with HCl 1 M and pre-incubated before use for 2 h at 37 °C, 5% $CO_2$ in order to reach the desired pH ($\pm$ 0.1 pH unit). pH in the medium was measured with a pH meter (Orion Star, Thermo Fisher Scientific). In some experiments, the pH was restored to 7.4 with NaOH 1 M. mTORC1 inhibitor (rapamycin), CHX protein synthesis inhibitor (cycloheximide), GSK-3β inhibitor (CHIR99021), Akt inhibitor (Akt1/2) and proteasome inhibitor (MG-132) were from Sigma-Aldrich. When used, DMSO was added as a vehicle in the control condition.

### Plasmid constructions
Mouse Rheb and mCherry-SEpHluorin (gift from Sergio Grinstein (Addgene plasmid # 32001; http://n2t.net/addgene:32001; RRID: Addgene_32001) (Koivusalo et al, 2010) were codon optimized (Thermo Fisher Scientific) and cloned into a MSGV retroviral vector (gift from David Ott [Addgene plasmid # 64269; http://n2t.net/addgene:64269; RRID: Addgene_64269]) (Coren et al, 2015), straight after a Thy1.1 (CD90.1) reporter gene, a furin cleavage and a T2A ribosomal skipping sites. Using this plasmid, gene transcription upon virus integration is dependent on the LTR promoter constitutive activity. "miR Scramble" and "miR P5CS" are miR-30-based shRNA (Transomic Technologies) that were cloned in a modified version of the pQCXIP (Takara) plasmid under the control of the human U6 promoter along with the U6 snRNA leader sequence (Chang et al, 2006) (this plasmid is a self-inactivating retroviral vector modified to contain a Thy1.1 reporter gene under the control of the human phosphoglycerate kinase promoter, a woodchuck posttranscriptional regulatory element, and a bovine growth hormone polyA signal following the 3' LTR). P5CS shRNA sequence was obtained with http://splashrna.mskcc.org/ online website prediction (Pelossof et al, 2017). sgRNA obtained

from http://chopchop.cbu.uib.no/ (Labun et al, 2016) were cloned in a modified self-inactivated MSGV retroviral vector. Briefly, the vector was as follows: 5'LTR U3 promoter was replaced with a RSV promoter, the splicing donor site and gag sequence were removed, a U6 promoter, the sgRNA sequence including an optimized CRISPR tracrRNA sequence (Chen et al, 2013), poly-T terminator, a human PGK promoter, two copies of Thy1.1 split by furin and T2A sequences, and a WPRE sequence were added, the 3'LTR was modified in order to keep a minimal U3 sequence that is devoid of promoter activity and a bovine growth hormone polyA sequence was added after the U5. d2eGFP or wild-type (and non-codon optimized) mouse c-Myc transgene, including a stop codon, was inserted downstream of the PGK promoter, followed by an IRES of the EMCV and an in-frame Thy1.1 cassette. Constructs were validated by Sanger sequencing. The ecotropic packaging plasmid pCL-Eco was a gift from Inder Verma (Addgene plasmid # 12371; http://n2t.net/addgene:12371; RRID: Addgene_12371) (Naviaux et al, 1996).

### Retrovirus production and transduction
293T were transfected using the calcium phosphate technique (Jordan et al, 1996) with a 1:1 retroviral vector to packaging plasmid ratio. Two days post-transfection, supernatants were 0.45μm-filtered and concentrated with the retro-concentin virus precipitation solution (SBI). Retroviral content was titrated with the C1498 cell line in the presence of protamine sulfate (Sigma-Aldrich), based on CD90.1/Thy1.1 positivity by flow cytometry. Two days post-activation using the aforementioned protocol, CD8+ T cells from OT-I or OT-I x CRISPR/Cas9 backgrounds were transduced (MOI of 2) with 10 μg/cm² of Retronectin (Takara) following manufacturer's instructions. Resulting cells were expanded, as previously mentioned, with 200 IU/mL rmIL-2. Three days post-infection, transduced cells were selected by magnetic separation (CELLection Biotin Binder Kit, ThermoFisher Scientific) via CD90.1/Thy1.1 surface labeling. Five to seven days post-transduction, purity (>90%) was confirmed by flow cytometry.

### Assessment of CTL proliferation and expansion
CTLs were labeled with the CellTrace Violet (CTV) Cell Proliferation Kit (5 μM; Thermo Fisher), re-activated with plates pre-coated with anti-CD3 antibodies (1 μg/mL), and/or stimulated with various doses of IL-2, and cell division number was tracked by flow cytometry in the viable fraction. Importantly, since CTLs do not represent a uniform cell size population, CTV incorporation does not give well resolved peaks. Consequently, we adapted a methodology to estimate cell division number by creating gates that approximatively identify each cell division number. Briefly, considering that CTLs below a certain value of fluorescence (that is obtained with unstimulated CTLs) are proliferating, we create a gate that represents one division by dividing this value by two (since a division should dilute CTV fluorescence by two); we then iteratively halve fluorescence values to create the other gates allowing to identify the other division numbers. Finally, estimated mean cell division number is obtained by summing up each percentage of cells belonging to a particular gate multiplied by the corresponding cell division number. Cell expansion was obtained either by counting cell number by trypan blue exclusion using a Neubauer chamber, or by flow cytometry using Precision count beads (Biolegend).

### Cytotoxicity assay

As target cells, C1498 cells were labeled with 2.5 μM CTV for subsequent discrimination, and pulsed for one hour with different concentrations of SIINFEKL peptide (Protein and Peptide Chemistry Facility of the University of Lausanne). Resulting cells were co-cultured with CTLs for four hours at various CTL-to-tumor ratios. C1498 cell death was quantified by flow cytometry with the live/dead fixable near-IR dead cell stain kit (Thermo Fisher). CTL-mediated lysis (%) was calculated as follows: (%CTL-induced cell death − % spontaneous cell death)/(100 − %spontaneous cell death). Antigen-specific lysis (%) was calculated as: (%CTL-induced cell death of Ag-pulsed C1498 − %CTL-induced cell death of unpulsed C1498)/ (100 − %CTL-induced cell death of unpulsed C1498).

### Cytokine secretion analyses

Cytokine content (IFN-γ, IL-2 and TNF) from supernatant of re-activated CTLs was determined using the mouse ELISA MAX Sets from Biolegend according to the manufacturer's instructions.

### Intracellular pH determination by confocal microscopy

SEpHluorin-transduced CTLs were pre-stained with Hoechst 34580 (Thermo Fisher Scientific) to stain nuclei. For standard curves, CTLs were seeded on poly-L-Lysine-coated Labtek I culture chamber slides (Nunc, 055083) in high K$^+$ buffer (145 mM KCl, 1 mM MgCl$_2$, 0.5 mM EGTA, 10 mM MES, 10 mM HEPES) containing nigericin and valinomycin (10 μM each, Thermo Fisher Scientific), and adjusted to pH 7.8, pH 7.3, pH 6.8 and pH 6.3 with NaOH. This protocol allows to equilibrate intracellular pH to extracellular pH. In parallel, CTLs were seeded on poly-L-Lysine-coated Labtek I culture chamber slides at pH 7.4 or pH 6.6 in bicarbonate and phenol-free media containing 15 mM HEPES and 15 mM MES in the presence of rmIL-2 (200 IU/mL). CTLs were imaged by confocal microscopy each 40 min.

Confocal fluorescence images were acquired with a Zeiss LSM 800 confocal microscope using a ×40 objective without immersion. Experiments were carried out at 37 °C in a humidified atmosphere without CO$_2$ using an incubation chamber. At least four positions per well were acquired. Resulting images were processed with *in-house* ImageJ/Fiji macros in order to obtain nucleus-free or cytoplasm-free pictures and fluorescence values. At least 100 cells were analyzed per condition.

### Cell cycle analysis

Briefly, CTLs were fixed and permeabilized with 70% ethanol, depleted from endogenous RNA using RNAse A (Sigma-Aldrich), and DNA was stained using propidium iodide (ThermoFisher Scientific). Resulting cells were analyzed by flow cytometry.

### IL-2R complexes quantification and binding capacities of IL-2 to IL-2R

rmIL-2 was biotinylated using the EZ-Link Sulfo-NHS-LC-Biotin kit (ThermoFisher Scientific) at a 100 biotin to 1 IL-2 molar ratio, as previously described by De Jong et al (De Jong et al, 1995), following manufacturer's instructions, and was serially filtered with an AMICON centrifugal unit (Merck) to remove unbound biotin. Following culture of CTLs under various pH, IL-2R complexes were determined by staining the cells with the biotinylated-rmIL-2 (equivalent to 200 IU/mL) at 4 °C for 30 min. Cells were stained with PE-conjugated streptavidin (Biolegend), the signal was further amplified with another round of staining with biotinylated anti-

streptavidin (Vector Laboratories) and PE-conjugated streptavidin, and analyzed by flow cytometry. The impact of pH on binding capacities of IL-2 to IL-2R was determined by incubating biotinylated-rmIL-2 (200 IU/mL) with CTLs under various pH in PBS at 4 °C for 30 min. IL-2 binding was determined following the same aforementioned protocol.

### Production and purification of human IL-2 "switch"

Rosetta pRAREII (Novagen) were transformed with the pET-22b plasmid encoding the IL-2 "switch" sequence (Gaggero et al, 2022) and protein was extracted from inclusion bodies. Non-aggregated protein was enriched by preparative size-exclusion column chromatography (Superdex75) followed by Amicon filtration (3KDa Cutoff).

### Staining for flow cytometry

For all the experiments, but intracellular pH detection, the buffer used was composed of PBS, 1% BSA, 0.1% NaN$_3$, and the staining procedure included the use of a live/dead fixable cell stain kit (Molecular Probes) to analyze viable cells. Cells were stained with antibodies targeting the protein of interest or appropriate isotype-matched controls. For apoptosis determination, cells were stained with AnnexinV-FITC (BD Biosciences) and the live/dead fixable near-IR dead cell stain kit. Apoptosis was analyzed amongst viable (live/dead marker$^-$) cells, and AnnexinV$^+$ cells were considered as apoptotic. Samples were run using a Gallios, a CytoFlex (Beckman Coulter) or a BD FACSCanto flow cytometer. Analyses were carried out on singlets using FlowJo software, and marker expression was calculated from the Median Fluorescence Intensity (MFI) by calculating the ratio MFI (rMFI) as: MFI marker/MFI isotype (or, MFI unstained in case isotype was not available, e.g., for biotinylated-IL-2 staining, MFI biot-IL-2/MFI w/o biot-IL-2). Then, the background was removed (rMFI-1), and the result was normalized to the "control" condition (i.e., the "pH 7.4" condition, which is considered to be equivalent to 100%), such as: normalized rMFI (%) = (rMFI[test] − 1) × 100/(rMFI[control] − 1).

### mRNA expression analyses by real-time quantitative PCR

DNA-free RNA was extracted from dry pellets using the Qiagen RNeasy kit and DNAse Set according to the manufacturer's instructions. Equal amounts of RNAs were used to synthesize cDNA with the oligodT primer from the Primescript first strand cDNA synthesis kit (Takara Bio). *Myc* expression was analyzed by normalization with *Actb* through real-time quantitative PCR (SDS 7900 HT instrument; Applied Biosystems) using the Kapa Sybr Fast mix (Roche). The following parameters were used: 95 °C for 3 min, and 40 cycles of 95 °C 3 s, 60 °C 30 s and 72 °C 1 s. Each reaction was performed in duplicate. The 2$^{\Delta Ct}$ method was used to normalize and linearize the results, but was adjusted to the observed primer efficacy (i.e., *Myc*: 1.80; *Actb*: 1.98).

### RNA sequencing

RNA quality was assessed on a Fragment Analyzer (Agilent Technologies) and all RNAs had a RQN between 7.5 and 10. RNA-seq libraries were prepared using 350 ng of total RNA and the Illumina TruSeq Stranded mRNA reagents (Illumina) on a Sciclone liquid handling robot (PerkinElmer) using a PerkinElmer-developed automated script. Cluster generation was performed with the resulting libraries using the Illumina HiSeq 2500 SR Cluster Kit

v4 reagents and sequenced on the Illumina HiSeq 2500 using TruSeq SBS Kit v4 reagents. Sequencing data were demultiplexed using the bcl2fastq Conversion Software (v. 2.20, Illumina).

### Immunoblotting

Unless specified, prior to the assay cells were starved for one hour and a half by culture at 37 °C in cytokine- and serum-free medium. For the assays, the resulting cells were cultured (1–2 million cells per mL) with rmIL-2 at various doses for different times. Cells were lysed in RIPA buffer supplemented with Halt phosphate/protease inhibitors (ThermoFisher Scientific) and were boiled at 97 °C for 10 min with Bolt LDS sample buffer and reducing agent (Thermo-Fisher Scientific). Protein samples were separated by SDS-PAGE and transferred to PVDF membranes using the iBlot2 system (Thermo Fisher Scientific). Antibody staining of the different molecules of interest was carried out according to the manufacturer's instructions. Images were acquired with a western blot imager (Fusion, Vilber Lourmat) and protein levels were quantified using the ImageJ software by analyzing pixel intensity. Membranes were stripped (Tris 62.5 mM pH 6.7, SDS 2%, 0.1 M β-mercaptoethanol, 30 min at 50 °C) in order to stain the same membrane for different proteins. Quantification was carried out using the ImageJ/Fiji software. For most of the molecules, phosphorylation status was calculated by dividing the signal of the phosphorylated protein by the signal of the matched total protein (except phosphorylation of eIF2α, GSK-3β, and p38, for which the signal was divided by the β-actin signal), which were acquired from the same membrane. Total levels of the other molecules assessed were calculated by dividing their signal to the β-actin signal. Results were normalized to the control condition (i.e., the "pH 7.4 + IL-2" condition yields 100%).

### Glutamine isotopic trace analysis

CTLs were pre-cultured 2 h in the absence of serum, IL-2 and glutamine a 37 °C. Upon culture for 4 h at 37 °C in CTL medium containing dialyzed serum, IL-2 and a glutamine isotope (2 mM—Sigma-Aldrich, 605166) for which all carbon are $^{13}C$-labeled, cell pellets were stored at −80 °C. CTL lysates (2 million cells) were extracted and homogenized by the addition of 500 µL of MeOH 80% and ceramic beads, in the air-cooled Cryolys Precellys Homogenizer (2 ×20 s at 10,000 rpm, Bertin Technologies, Rockville, MD, USA). Homogenized extracts were centrifuged for 15 min at $21,000 \times g$ and 4 °C, and the resulting supernatant was collected and evaporated to dryness in a vacuum concentrator (LabConco, Missouri, USA). Dried sample extracts were re-suspended in MeOH:H$_2$O (4:1, v/v) prior to LC-HRMS analysis according to the total protein content (to normalize for sample amount).

Cell extracts were analyzed by Hydrophilic Interaction Liquid Chromatography coupled to high resolution mass spectrometry (HILIC - HRMS) in negative ionization mode using a 6550 Quadrupole Time-of-Flight (Q-TOF) system interfaced with 1290 UHPLC system (Agilent Technologies) as previously described (Gallart-Ayala et al, 2018).

Raw LC-MS files were processed in Profinder B.08.00 software (Agilent Technologies) using the targeted data mining in isotopologue extraction mode. The metabolite identification was based on accurate mass and retention time matching against an *in-house* database. The Extracted Ion Chromatogram areas (EICs) of

each isotopologue (M + 0, M + 1, M + 2, M + 3,…) were corrected for natural isotope abundance (Midani et al, 2017) and the label incorporation or $^{13}C$ enrichment was calculated based on relative isotopologue abundance (in %), in each one of two analyzed conditions (Roci et al, 2016).

In addition, pooled QC samples (representative of the entire sample set) were analyzed periodically (every 4 samples) throughout the overall analytical run in order to assess the quality of the data, correct the signal intensity drift and remove the peaks with poor reproducibility. In addition, a series of diluted quality controls (dQC) were prepared by dilution with methanol: 100% QC, 50% QC, 25%QC, 12.5%QC and 6.25%QC and analyzed at the beginning and at the end of the sample batch. This QC dilution series served as a linearity filter to remove the features that do not respond linearly or correlation with dilution factor is <0.65.

### Amino acid quantification

CTL lysates (1–2 million cells) were extracted and homogenized by the addition of 750 µL of MeOH 80% and ceramic beads, in the air-cooled Cryolys Precellys Homogenizer (2 ×20 s at 10,000 rpm, Bertin Technologies, Rockville, MD, USA). Homogenized extracts were centrifuged for 15 min at $21,000 \times g$ and 4 °C. The 50 µL of extract were mixed with 250 µL of Methanol containing isotopic labeled internal standards. Sample extracts were vortexed and centrifuged (15 min, $2700 \times g$ at 4 °C) and the resulting supernatant was collected and injected into the HILIC-HRMS (Q Exactive Focus instrument interfaced with the a HESI source) (Thermo Fisher Scientific) for amino acid quantification (Teav et al, 2019).

### Total protein quantification

After metabolite extraction with organic solvents (MeOH) the obtained protein pellets were evaporated and lysed in 20 mM Tris-HCl (pH 7.5), 4 M guanidine hydrochloride, 150 mM NaCl, 1 mM Na$_2$EDTA, 1 mM EGTA, 1% Triton, 2.5 mM sodium pyrophosphate, 1 mM β-glycerophosphate, 1 mM Na$_3$VO4, 1 µg/ml leupeptin using the Cryolys Precellys 24 sample Homogenizer (2 ×20 s at 10,000 rpm, Bertin Technologies, Rockville, MD, USA) with ceramic beads. BCA Protein Assay Kit (Thermo Scientific, Massachusetts, USA) was used to measure (A562nm) total protein concentration (Hidex, Turku, Finland).

### Amino acid uptake assay

CTLs were incubated at 5 million per mL in HBSS buffer (Invitrogen) at various pH for 20 min at room temperature in the presence of 10 µCi/mL of tritiated glutamine (equivalent to 165 nM; Hartmann Analytic). Alternatively, CTLs were pre-incubated (37 °C, 20 min) under various pH in CTL medium containing IL-2 with dialyzed serum without glutamine and were further incubated for 20 min in the presence of 10 µCi/mL of tritiated glutamine. Cells were washed, lysed with 0.3% final SDS, and radioactivity incorporation was determined with a scintillation counter (PerkinElmer) after MicroScint 40 (PerkinElmer) addition.

### In vitro mTORC1 kinase activity

Kinase activity of recombinant mTORC1 complexes (human mTOR/RAPTOR/MLST8, Sigma-Aldrich) was assessed by determining phosphorylation status of recombinant human 4E-BP1 (Abcam). Kinase assay was carried out with 20 mM HEPES, 20 mM MES, 10 mM MgCl$_2$ and 50 mM KCl. In all, 10× reaction buffer

were prepared by adding NaOH to reach pH 7.4 and pH 6.6. mTORC1, 4E-BP1 and reaction buffer were mixed and preincubated at room temperature for 20 min. Upon addition of 200 μM ATP, samples were incubated at 30 °C for 10 min. Thereafter, samples were boiled at 97 °C for 10 min with Bolt LDS sample buffer and reducing agent, and were separated by SDS-PAGE and transferred to PVDF membranes. Staining was carried out according to the beforehand described section for western blot.

### Statistical analyses

Statistical significance was evaluated using the two-tailed paired Student's *t* test for comparison of two groups, or a one- or two-way ANOVA with a Tukey post-hoc test for comparison of multiple groups using the GraphPad Software (Prism). For correlation curves, linear and non-linear correlation curves were modelled with GraphPad according to plot patterns.

## Data availability

The datasets produced in this study are available in the following databases: RNA-seq data: Gene Expression Omnibus GSE269681. Metabolomics data for 13C-glutamine incorporation: https://doi.org/10.5281/zenodo.11568730 (https://doi.org/10.5281/zenodo.11568730). Flow cytometry data of Figs. 1E–G and 2C: https://doi.org/10.5281/zenodo.12124559 (https://doi.org/10.5281/zenodo.12124559).

The source data of this paper are collected in the following database record: biostudies:S-SCDT-10_1038-S44318-024-00235-w.

## Peer review information

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

## Acknowledgements

This work was generously supported by the Ludwig Institute for Cancer Research, the Swiss National Science Foundation (SNSF# 310030_204326 to MI), the ISREC Foundation, the Prostate Cancer Foundation, Cancer and the Biltema and Carigest Foundations to MI. We thank the Flow Cytometry Facility (Francisco Sala de Oyanguren), the Metabolomics Platform (Julijana Ivanisevic, Hector Gallart Ayala and Tony Teav), the Lausanne Genomic Technologies Facility and the Cellular Imaging Facility (Florence Morgenthaler Grand) of the University of Lausanne for their expertise, Marjorie Decroux for technical help, and Patrick Reichenbach for useful scientific discussions. We also thank Prof. George Coukos (Ludwig Institute for Cancer Research of the University of Lausanne), Prof. Chi V Dang (Johns Hopkins University School of Medicine) and Prof. Christoph Hess (University of Basel, University of Cambridge) for insightful scientific discussions earlier in the project.

## Author contributions

**Romain Vuillefroy de Silly**: Conceptualization; Data curation; Software; Formal analysis; Supervision; Investigation; Visualization; Methodology; Writing—original draft; Project administration; Writing—review and editing. **Laetitia Pericou**: Investigation. **Bili Seijo**: Resources. **Isaac Crespo**: Resources; Data curation; Software; Formal analysis. **Melita Irving**: Conceptualization; Resources; Supervision; Funding acquisition; Writing—original draft; Project administration; Writing—review and editing.

Source data underlying figure panels in this paper may have individual authorship assigned. Where available, figure panel/source data authorship is listed in the following database record: biostudies:S-SCDT-10_1038-S44318-024-00235-w.

## Disclosure and competing interests statement

The authors declare no competing interests.

# Expanded View Figures

**Figure EV1. Impact of acidity on CTL size, granularity and proliferation in response to anti-CD3 stimulation.**

(A) Illustration of the main steps and functions of anti-tumor CD8 + T cell migrating to the tumor microenvironment. Previously differentiated anti-tumor CD8 + T cells (i.e. effector cytotoxic T lymphocytes; CTLs) encounter acidic conditions at the tumor site when they exert their activities. (B) Schematic of the in vitro methodology used to deconvolve the impact of pH on CTLs function in tumors. (C) Effector/memory phenotype of primed and expanded OT-I CD8$^+$ T cells used in this study. Naive OT-I CD8$^+$ T cells were activated for two days with pre-coated anti-CD3 and soluble anti-CD28 antibodies, and expanded for further ten days in the presence of murine IL-2. Dot plots show one representative experiment of background staining ("Isotype", top graph), and of CD44 CD62L staining (bottom graph) obtained by flow cytometry. Histograms show one representative experiment of background staining ("Isotype", black dashed line), and of Granzyme B staining (red solid line) obtained by flow cytometry. Almost 80% of cells are effectors (CD44$^+$ CD62L$^-$), while all express Granzyme B to relatively high levels. (D) Effector generation protocol gives efficient killing capacities of the resulting OT-I T cells. Varying numbers of OT-I CTLs were co-cultured with C1498 tumor cells pulsed (solid lines and symbols), or not (dashed lines and empty symbols), with 1 µM of antigen (minimal ovalbumin peptide epitope, SIINFEKL). Results show the mean percentage of tumor cell lysis ± SD of at least three (or two for 0.4 and 1.1 CTL:Tumor ratios) biological replicates from at least two independent experiments. (E) Impact of pH on CTL proliferation following re-activation with increasing doses of anti-CD3. OT-I CTLs were cultured for three days in the presence of an agonistic anti-CD3 antibody (pre-coated plates). Line graph show the estimated division number ± SEM of at least three biological replicates from at least two independent experiments. (F) Low pH prevents the increase of CTL size and granularity following anti-CD3 re-activation. OT-I CTLs were cultured in the presence of an agonistic anti-CD3 antibody (1 µg/mL pre-coated plates). Bar graph shows the mean cell size or granularity ± SEM of at least four biological replicates from at least two independent experiments. Density plots shows one representative experiment obtained by flow cytometry one day post re-stimulation. ns: not statistically significant, ****$P < 0.0001$. Cell size: ***$P = 0.0008$. Granularity: **$P = 0.0018$, ***$P = 0.0002$ (one-way repeated measures ANOVA, Tukey post-hoc test). (G) Cartoon depicting the two main steps involved in cytokine secretion and proliferation following anti-CD3 re-activation of CTLs. The TCR/CD3-independent signaling allowing CTL proliferation is mostly mediated by IL-2/IL-2R signaling. (H) Design of the experiments in order to analyze at which step pH impacts cytokine secretion and proliferation during anti-CD3-mediated CTL re-activation. pH was neutralized or acidified at step 1, or 2, in order to know whether cytokine secretion or proliferation can be restored.

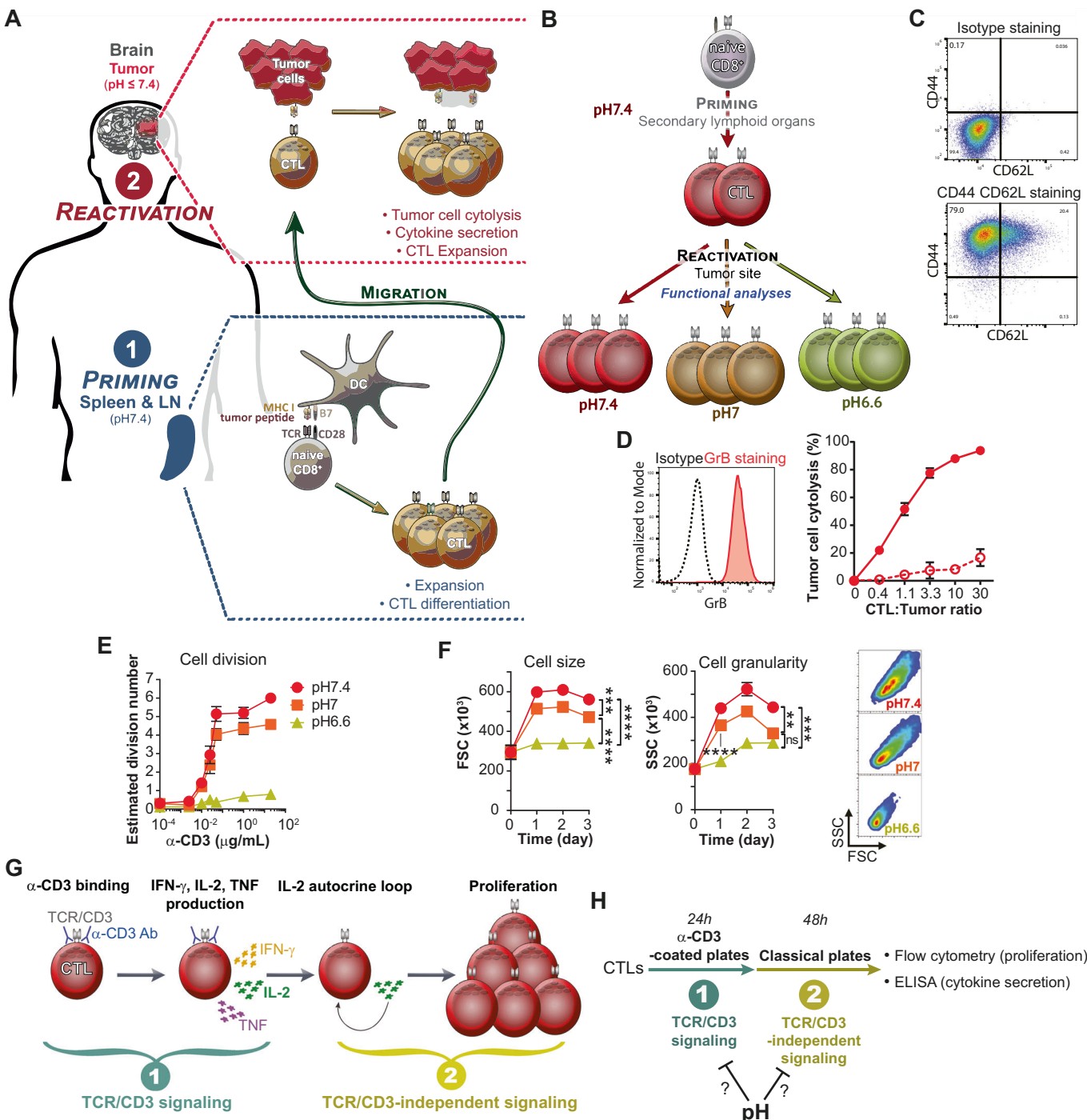

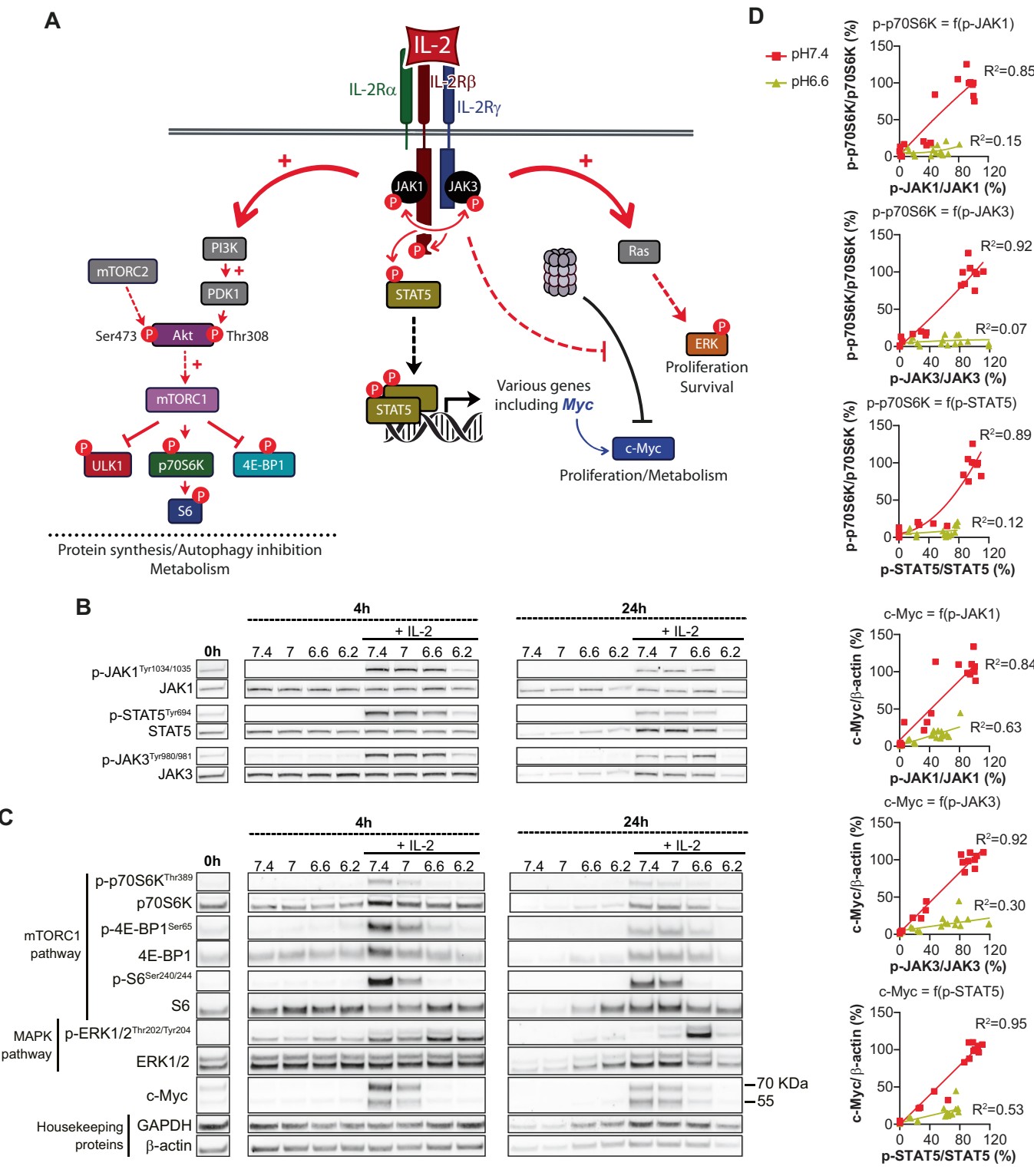

**Figure EV2. Acidity lowers IL-2R signaling.**

(**A**) Simplified scheme of the IL-2R signaling. (**B**) IL-2R signaling is disturbed at lower pH. OT-I CTLs were cultured at various pH in the presence, or the absence of exogenous murine IL-2 for 4 or 24 h. One representative western blot is shown. (**C**) Low pH disturbs IL-2 –induced mTORC1 pathway and c-Myc levels. The methodology is the same as in (**B**). One representative western blot is shown. (**D**) Correlation between the activation of first signaling transducers and mTORC1 pathway targets, or c-Myc levels. Results show individual values obtained from the experiments displayed in Fig. 3C. Corresponding correlation curves together with associated $R^2$ are shown.

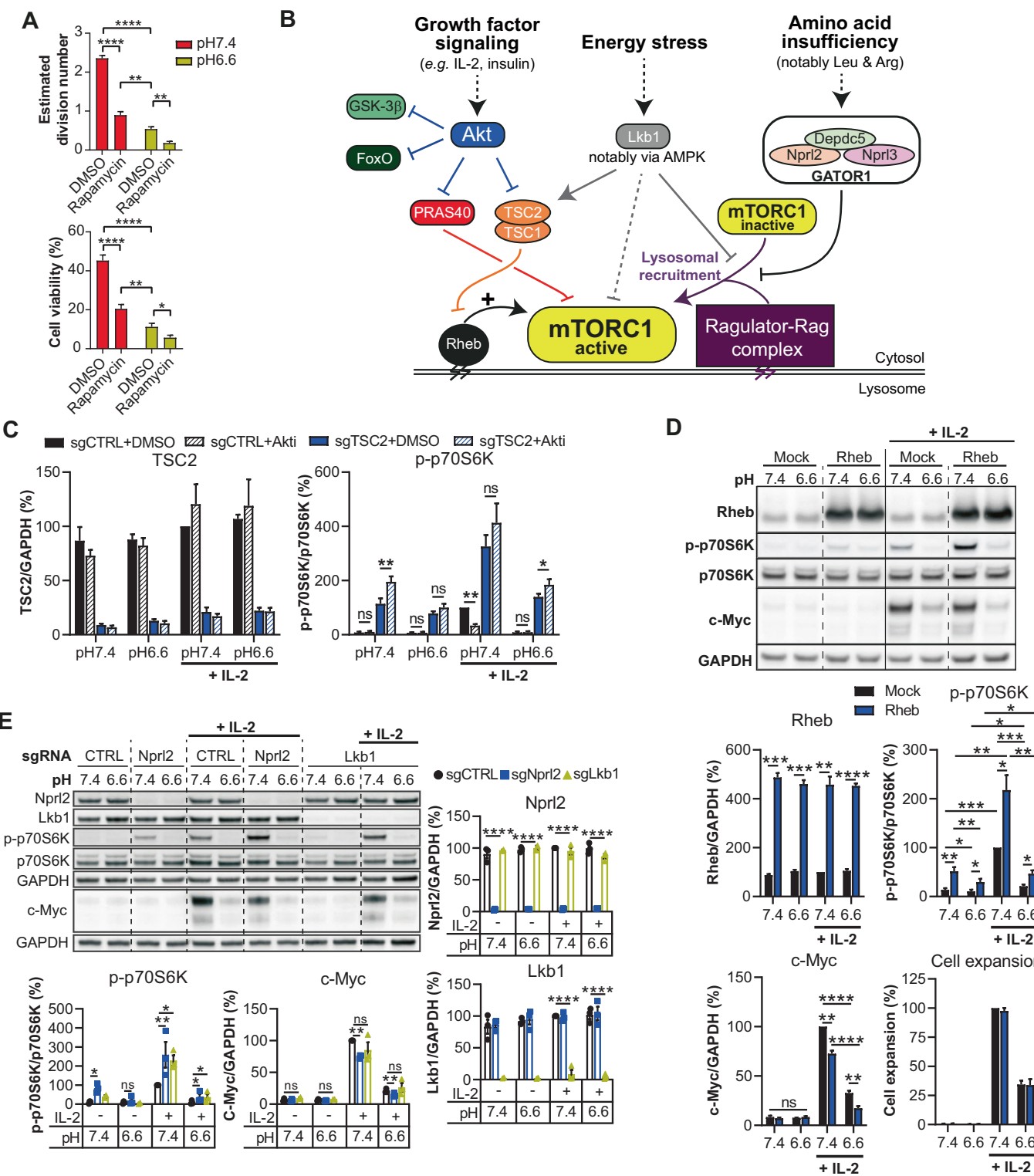

◀ **Figure EV3.   mTORC1 pathway exploration.**

(A) mTORC1 inhibition leads to proliferation and viability defects following IL-2 stimulation. OT-I CTLs were cultured for three days in the presence ("Rapamycin"), or absence ("DMSO"), of rapamycin 10 nM with exogenous murine IL-2. Results show the estimated division number, or viability, + SEM of three biological replicates from two independent experiments. ****$P < 0.0001$. Division number: pH 7.4+Rapa vs pH 6.6 + DMSO **$P = 0.0021$, pH 6.6+Rapa vs pH 6.6 + DMSO **$P = 0.0020$. Viability: pH 7.4+Rapa vs pH 6.6 + DMSO **$P = 0.0014$, pH 6.6+Rapa vs pH 6.6 + DMSO *$P = 0.0185$ (one-way repeated measures ANOVA, Tukey post-hoc test). (B) Simplified scheme of mTORC1 activity regulation. mTORC1 is recruited to the lysosome via the Ragulator-Rag complex where it can eventually be activated by Rheb. Growth factor signaling leads to Akt activation that phosphorylates PRAS40 and TSC2 allowing to block their capacity to inhibit mTORC1 activity. Amino acid sufficiency prevents GATOR1 complex from impeding mTORC1 recruitment to lysosomes. Energy stress leads to Lkb1 activation that can lower mTORC1 activity via several mechanisms (mostly involving AMPK) including TSC2 activation, Ragulator-Rag inhibition and mTORC1 inhibiting phosphorylation. (C) Impact of Akt inhibition on mTORC1 activity in TSC2 knockouts. OT-I x CRISPR/Cas9 CTLs were transduced with retroviruses encoding a negative control, a PRAS40 or a TSC2 sgRNA, and were cultured for 4 h in the presence, or absence ("DMSO"), of Akt1/2 inhibitor (10μM- Akti). Bar graphs show the mean levels normalized to the condition "pH 7.4 + IL-2 sgCTRL DMSO" + SEM of four biological replicates from two independent experiments. ns: not significant. pH 7.4 sgTSC2+Akti vs sgTSC2+DMSO **$P = 0.0037$, pH 7.4 + IL-2 sgCTRL+Akti vs sgCTRL+DMSO **$P = 0.0014$, pH 6.6 + IL-2 sgTSC2+Akti vs sgTSC2+DMSO *$P = 0.0195$ (Student's paired $t$ test). (D) Rheb overexpression improves mTORC1 activity. OT-I CTLs were transduced with a control (Mock) or a Rheb encoding retrovirus, and were cultured for 4 h (western blot analyses) or 4 days (cell expansion and viability). One representative western blot is shown. Bar graphs show the mean levels normalized to the condition "pH 7.4 + IL-2 Mock" + SEM of four biological replicates from two independent experiments. ns: not significant, ****$P < 0.0001$. Rheb: pH 7.4 Rheb vs Mock ***$P = 0.0001$, pH 6.6 Rheb vs Mock ***$P = 0.0002$, pH 7.4 + IL-2 Rheb vs Mock **$P = 0.0017$. p-p70S6K: pH 7.4 Rheb vs Mock **$P = 0.0099$, pH 6.6 Mock vs pH 7.4 Mock *$P = 0.0445$, pH 7.4 + IL-2 Mock vs pH 7.4 Mock ***$P = 0.0001$, pH 6.6 Rheb vs pH 7.4 Rheb **$p = 0.0019$, pH 7.4 + IL-2 Rheb vs pH 7.4 Rheb **$P = 0.0098$, pH 6.6 Rheb vs pH 6.6 Mock *$P = 0.0181$, pH 6.6 + IL-2 Mock vs pH 6.6 Mock *$P = 0.0143$, pH 6.6 + IL-2 Rheb vs pH 6.6 Rheb *$P = 0.0396$, pH 7.4 + IL-2 Rheb vs Mock *$P = 0.0307$, pH 6.6 + IL-2 Mock vs pH 7.4 + IL-2 Mock ***$P = 0.0002$, pH 6.6 + IL-2 Rheb vs pH 7.4 + IL-2 Rheb **$P = 0.0056$, pH 6.6 + IL-2 Rheb vs pH 6.6 + IL-2 Mock *$P = 0.0125$. c-Myc: pH 7.4 + IL-2 Rheb vs Mock **$P = 0.0032$, pH 6.6 + IL-2 Rheb vs Mock **$P = 0.0053$. (Student's paired $t$ test). (E) Nprl2 and Lkb1 knockouts do not improve mTORC1 activity at low pH. OT-I x CRISPR/Cas9 CTLs were transduced with retroviruses encoding a negative control, a Nprl2 or a Lkb1 sgRNA, and were cultured for 4 h. One representative western blot from two membranes of the same samples is shown. Bar graphs show the mean levels normalized to the condition "pH 7.4 + IL-2 sgCTRL" + SEM of three biological replicates (except for conditions pH 7.4 and pH 6.6 sgLkb1 without IL-2: two) from two independent experiments. ns: not significant, ****$P < 0.0001$. p-p70S6K: pH 7.4 sgNprl2 vs sgCTRL *$P = 0.0296$, pH 7.4 + IL-2 sgNprl2 vs pH 7.4 + IL-2 sgCTRL **$P = 0.0050$, pH 7.4 + IL-2 sgLkb1 vs pH 7.4 + IL-2 sgCTRL *$P = 0.0429$, pH 6.6 + IL-2 sgNprl2 vs pH 6.6 + IL-2 sgCTRL *$P = 0.0339$, pH 6.6 + IL-2 sgLkb1 vs pH 6.6 + IL-2 sgCTRL *$P = 0.0187$. c-Myc: pH 7.4 + IL-2 sgNprl2 vs sgCTRL **$P = 0.0042$, pH 6.6 + IL-2 sgNprl2 vs sgCTRL **$P = 0.0023$. (Student's paired $t$ test).

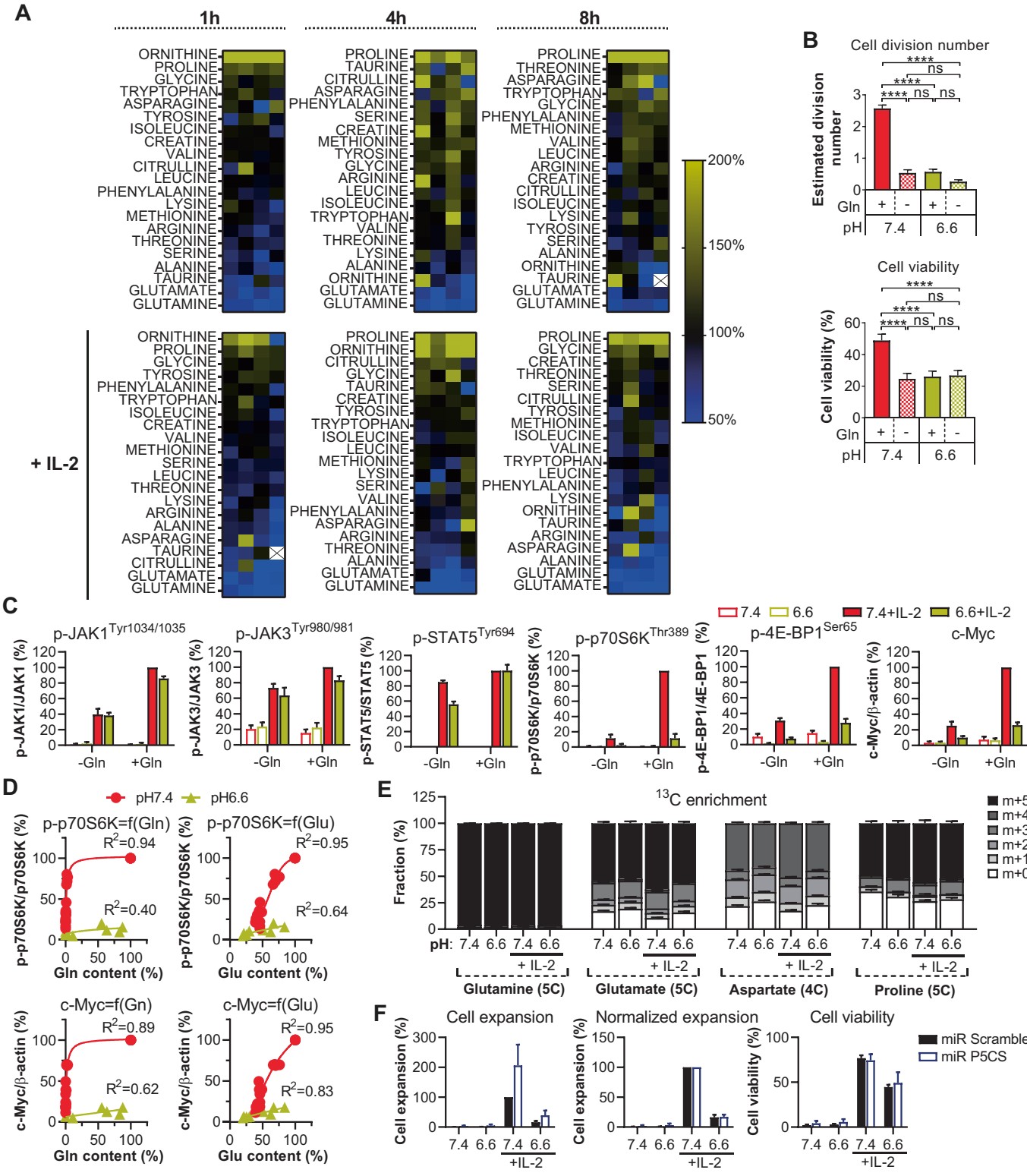

**Figure EV4. Exploration of glutamine metabolism at low pH.**

(A) Heat map of intracellular amino acid contents as a function of pH, time, and IL-2 stimulation. Intracellular levels of amino acids were quantified from OT-I CTLs. Results show the percent level of amino acid as compared to the matched pH 7.4 condition using a color code, as outlined on the ladder. Each square shows one biological replicate, with a total of four from two independent experiments. White square with a cross inside means the amino acid was not detected. (B) Glutamine deprivation inhibits CTL proliferation and viability. OT-I CTLs were cultured for three days with exogenous murine IL-2 in the presence, or absence, of glutamine (under the glutamax form, 4 mM). Results show the estimated division number, or cell viability, $+$ SEM of at least four biological replicates from at least two independent experiments. ns: not significant, $*P < 0.05$, $****P < 0.0001$ (one-way repeated measures ANOVA, Tukey post-hoc test). (C) Glutamine deprivation lowers mTORC1 activation and c-Myc accumulation. OT-I CTLs were cultured for 4 h in the presence, or absence, of glutamine (2 mM). Results show the mean levels normalized to the condition "pH 7.4 $+$ IL-2" $\pm$ SEM of four biological replicates from two independent experiments. (D) mTORC1/c-Myc as a function of intracellular levels of glutamine/glutamate. OT-I CTLs were cultured for 4 h with exogenous murine IL-2 in the presence of various exogenous quantities of glutamine. Results show the individual values of p-p70S6K or c-Myc as a function of intracellular amino acid content normalized to the condition "pH 7.4 $+$ 2000 μM Gln" from four biological replicates out of two independent experiments. A correlation curve and $R^2$ for each pH is displayed. (E) Intracellular glutamine, glutamate, aspartate and proline are coming from extracellular glutamine. OT-I CTLs were cultured for 4 h with isotopic $^{13}$C-glutamine (2 mM). Results show isotopologue distribution of the indicated intracellular amino acid $+$ SEM of four biological replicates from two independent experiments. The maximum number of carbons that can be labeled is indicated under bracket aside each amino acid. (F) P5CS knockdown does not restore CTL proliferation and viability at low pH. OT-I CTLs transduced with a control miR (Scramble) or a P5CS-targeting miR were cultured for 3 days. Bar graphs display mean CTL expansion normalized to the condition "pH 7.4 $+$ IL-2", mean CTL expansion normalized to the condition "pH 7.4 $+$ IL-2" per corresponding miR, or mean CTL viability $+$ SEM of four biological replicates from two independent experiments.

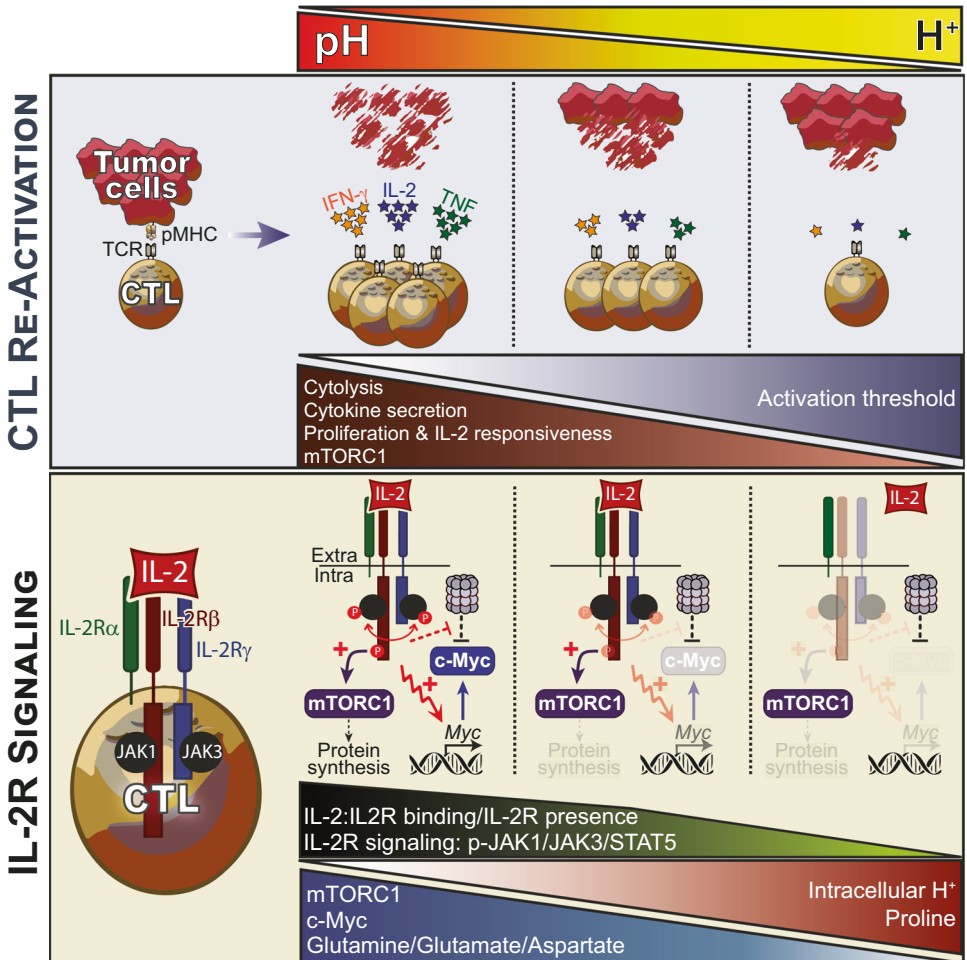

**Figure EV5.** Schematic overview of the impact of increasingly acidic conditions on CTLs.

