## [Peer Review File · The EMBO Journal]

Acidity suppresses CD8+ T-cell function by perturbing IL-2, mTORC1, and c-Myc signaling

Romain Vuillefroy de Silly, Laetitia Pericou, Bili Seijo, Isaac Crespo, and Melita Irving

Corresponding author(s): Romain Vuillefroy de Silly (romain.vuillefroydesilly@unil.ch) , Melita Irving (melita.irving@unil.ch)

Review Timeline:

Submission Date:	26th Feb 24
Editorial Decision:	3rd Apr 24
Revision Received:	1st Jul 24
Editorial Decision:	26th Jul 24
Revision Received:	13th Aug 24
Accepted:	20th Aug 24

Editor: Ioannis Papaioannou

Transaction Report:

Dear Dr. Vuillefroy de Silly,

Thank you again for submitting your manuscript EMBOJ-2024-117094 for consideration by The EMBO Journal. It has been seen by four experts in the field, and we have received the full set of their comments, which I have already shared with you (they are included again below). I would also like to thank you for your draft point-by-point response to their comments and for your provisional revision plan, which were very helpful for us to reach a fair and balanced editorial decision on the manuscript.

The referees recognize that this work adds to previous studies on T cell function in acidic conditions and mention that the findings are potentially interesting. However, they also identify a number of limitations in the study and the manuscript, and they raise several technical and conceptual concerns, while their well-informed reports list relevant suggestions for the improvement of the manuscript.

Given the referees' comments and recommendations, as well as your willingness to revise your study substantially and address their concerns, I would like to invite you to submit a revised version of the manuscript along with a detailed point-by-point response addressing all referees' comments. I should add that it is EMBO Journal policy to allow only a single round of major experimental revision, and acceptance of your manuscript will therefore depend on the completeness of your responses in this revised version. If you have any questions or comments, we can discuss further in a video call, if you like.

We generally allow three months as standard revision time (July 2, 2024). As a matter of policy, competing manuscripts published during this period will not negatively impact our assessment of the conceptual advance presented by your study. However, we request that you contact us as soon as possible upon publication of any related work, to discuss how to proceed. Should you foresee a problem in meeting this three-month deadline, please let us know in advance and we may be able to grant an extension.

Thank you for the opportunity to consider your work for publication in The EMBO Journal. I look forward to your revision.

Yours sincerely,

Instructions for preparing your revised manuscript

1. When you are ready to submit the revision, please upload:

- A Word file of the manuscript text (including legends of main Figures, EV Figures and Tables). Please make sure that changes are highlighted (or "tracked") to be clearly visible.

- Individual production-quality figure files (one file per figure). When assembling your figures, please refer to our figure preparation guidelines in order to ensure proper formatting and readability in print as well as on screen:

If the data shown in a figure are obtained from n {less than or equal to} 2, please use scatter plots showing the individual data points.

- i. the name of the statistical test used to generate error bars and P values
- ii. the number (n) of independent experiments (please specify technical or biological replicates) underlying each data point (discussion of statistical methodology can be reported in the Materials and Methods section, but figure legends should contain a basic description of n , P , and the test applied)
- iii. the nature of the bars and error bars (s.d., s.e.m.).

- A point-by-point response to the referees' comments, with a detailed description of the changes made (as a word file). All referees' concerns must be fully addressed and their suggestions taken on board. When preparing your letter of response to the referees' comments, please bear in mind that this will form part of the Review Process File and will therefore be available online to the community. Please note that you have the possibility to opt out of the transparent process at any stage prior to publication by letting the editorial office know (contact@embojournal.org); if you do opt out, the Review Process File link will point to the

following statement: "No Review Process File is available with this article, as the authors have chosen not to make the review process public in this case.". For more details on our Transparent Editorial Process, please visit our website: <https://www.embopress.org/page/journal/14602075/authorguide#transparentprocess>

- Expanded View (EV) files (replacing Supplementary Information) that are collapsible/expandable online. A maximum of 5 EV Figures can be typeset. EV Figures should be cited as "Figure EV1, Figure EV2" etc. in the text, and their respective legends should be included in the manuscript file after the legends of regular figures. See detailed instructions regarding Expanded View files here:

- For the figures that you do NOT wish to display as Expanded View figures, they should be bundled together with their legends in a single PDF file called "Appendix", which should start with a short Table of Contents (including page numbers). Appendix figures should be referred to in the main text as: "Appendix Figure S1, Appendix Figure S2" etc. Please see detailed instructions here: <https://www.embopress.org/page/journal/14602075/authorguide#expandedview>

- A complete author checklist, which you can download from our author guidelines (<https://www.embopress.org/page/journal/14602075/authorguide>). Please note that the checklist will also be part of the Review Process File.

2. Please note that no statistics should be calculated and shown in Figures if $n=2$.

3. Before submitting your revision, primary datasets (and computer code, where appropriate) produced in this study need to be deposited in appropriate public databases (see <https://www.embopress.org/page/journal/14602075/authorguide#dataavailability>). In particular, we would kindly ask you to deposit the reported RNA sequencing and mass spectrometry datasets. The accession numbers and databases should be listed in a formal "Data availability" section (placed after Materials and Methods) that follows the model below (see also <https://www.embopress.org/page/journal/14602075/authorguide#dataavailability>):

Data availability

- RNA-seq data: Gene Expression Omnibus GSE46843 (<https://www.ncbi.nlm.nih.gov/geo/query/acc.cgi?acc=GSE46843>)
- [data type]: [name of the resource] [accession number/identifier/doi] ([URL or identifiers.org/DATABASE:ACCESSION])

*** All links should resolve to a page where the data can be accessed. ***

*** Please remember to provide in the Data availability section of your revised manuscript reviewer passwords if the datasets are not yet public. ***

*** The Data Availability Section is restricted to new primary data that are part of this study. In case you have no data that require deposition in a public database, please state so instead of referring to the database: "Our study includes no data deposited in public repositories." under the heading "Data availability". ***

4. Please check that the title and the abstract of the manuscript are brief, yet explicit, even to non-specialists. The length of the title should not exceed 100 characters, and the abstract should be a single paragraph not exceeding 175 words.

5. Please also note our reference format: <https://www.embopress.org/page/journal/14602075/authorguide#referencesformat>.

7. Please remember: digital image enhancement is acceptable practice, as long as it accurately represents the original data and conforms to community standards. If a figure has been subjected to significant electronic manipulation, this must be noted in the figure legend or in the "Materials and Methods" section. The editors reserve the right to request original versions of figures and the original images that were used to assemble the figure.

8. Our journal encourages inclusion of data citations in the reference list to directly cite datasets that were obtained from public databases. Data citations in the article text are distinct from normal bibliographical citations and should directly link to the database records from which the data can be accessed. In the main text, data citations are formatted as follows: "Data ref: Smith et al, 2001" or "Data ref: NCBI Sequence Read Archive PRJNA342805, 2017". In the Reference list, data citations must be labeled with "[DATASET]". A data reference must provide the database name, accession number/identifiers, and a resolvable link to the landing page from which the data can be accessed at the end of the reference. Further instructions are available at: <https://www.embopress.org/page/journal/14602075/authorguide#referencesformat>.

9. We request authors to consider both actual and perceived competing interests. Please review our policy (<https://www.embopress.org/page/journal/14602075/authorguide#conflictsofinterest>) and update your competing interests statement if necessary. Please name this section 'Disclosure and competing interests statement' and place it after the Acknowledgements section.
10. Please note that all corresponding authors are required to provide an ORCID ID upon submission of a revised manuscript (<https://orcid.org/>). Please find instructions on how to link your ORCID ID to your account in our manuscript tracking system in our Author guidelines (<https://www.embopress.org/page/journal/14602075/authorguide#authorshipguidelines>).
11. We use CRediT to specify the contributions of each author in the journal submission system. CRediT replaces the author contribution section, which should be removed from the manuscript. Please use the free text box to provide more detailed descriptions. See also guide to authors: <https://www.embopress.org/page/journal/14602075/authorguide#authorshipguidelines>.
12. Further information is available in our Guide For Authors: <https://www.embopress.org/page/journal/14602075/authorguide>
13. We would also welcome the submission of cover suggestions or motifs to be used by our Graphics Illustrator in designing a cover.
14. Please use the link below to submit your revision:
<https://emboj.msubmit.net/cgi-bin/main.plex>

Referee #2:

The manuscript "Acidity perturbs IL-2 responsiveness, mTORC1 and c-Myc in CD8+ T cells" seeks to examine the impact of reducing culture media pH on in vitro differentiated cytotoxic T lymphocyte (CTL) function, signalling and amino acid content. The motivation behind this investigation was to understand how the specific variable of acidity, which is typically low in solid tumours, impacts CTL signalling, to better inform strategies looking to reinvigorate T cell responses against cancer.

This is not the first study to examine the impact of extracellular pH on T cell function, with most of the key CTL cellular phenotypes reported within this study (reduced IFN γ , IL2 production, reduced cell proliferation and blunted CTL killing, though only under select conditions here) initially reported in previous works on this topic, which is outlined and suitably referenced by the authors in the text. This study does however represent a more detailed and extensive attempt than prior work to measure and test the cell signalling changes downstream of the TCR and IL2 when CTL are placed into reduced pH media.

Investigation into the effect of low pH on IL2 signalling find that: 1) consistent with less detailed previous work, IL2-receptor expression is reduced and JAK/STAT signalling is decreased, 2) show that mTORc1 activity is substantially reduced, which is consistent with what known about the effects of pH on cell signalling but has not to my knowledge been shown in CTL and 3) that cMyc expression is substantially decreased. The authors also find that strong TCR stimulation can over-ride the effects of low pH conditions on Myc expression but not mTORc1 signalling.

The authors try to pull apart how change in mTORc1 activity and Myc expression is being regulated downstream of IL2. For Myc they ascribe this to a combination of transcript level and protein degradation rate. For mTORc1 they investigate several possible major upstream signalling pathways as well as control of amino acid content. While the authors put in considerable work on this question and are able to rule out a number of signalling pathways as individual mechanisms in their culture system, unfortunately they have not been able to pin down what the mechanism is here for pH control of mTORc1. To close the paper the authors look at changes in amino acid content due to low pH showing an increase in proline and decrease in glutamine/glutamate and find that glutamine uptake is reduced in low pH.

A strength of this study is that the authors interrogate the data quantitatively. This is a much better and more detailed look at the effects of pH on cell signalling in CTL than has been done previously. The use of a range of KO or inhibitor strategies to actually test relative likelihood of upstream pathways being explanatory for the effect of low pH on the cell signalling and proliferation phenotype versus just looking at correlative changes in expression is a particular strength.

Some areas where the manuscript could be improved are outlined below:

Major comments:

1)

The authors claim that IL2 dependent expression of Myc is impacted at the transcript level by low pH but that a difference in protein degradation rate is needed to explain the full reduction in Myc at a protein level.

Fig 4e is used as a justification that Myc protein and mRNA levels do not directly correspond. However, if I'm reading this correctly, the data plotted here for the pH6.6 condition seems to indicate that there were 3 samples where the Myc mRNA level was 80-100% of the level seen in the pH 7.4 + IL2 200 IU/mL condition ... yet in Fig 4C the average percentage for pH 6.6 is ~60% of the pH7.4 condition for saturating IL2 levels. There is only one data point in 4c that has large enough error bars for the pH 6.6 condition to contain these three data points (this is the 2000 IU/mL condition) though this would mean the 4th value would have to be very low to generate an average of ~60%. This doesn't make sense to me. Has there been some kind of plotting error? Could some of the raw measurements be included in a supplementary figure and/or could shape/size be used to help distinguish how the data points in 4d-e relate back to the IL2 concentration conditions in 4c? These are quite influential data points when it comes to looking at correlation with JAK/STAT signalling parameters and are the data points that give the strongest support to the idea of Myc protein vs mRNA not correlated in high vs low pH.

2)

An issue with regards protein degradation as a required mechanism to generate the pH dependent difference in Myc protein expression - while MG132 does increase Myc protein level in the low pH condition it also does this substantially in the pH7.4 + IL2 and pH7.4 -IL-2 conditions (Fig 4f), the latter of which has substantially less Myc mRNA than the pH 6.6. condition (Fig 4c). It is not clear to me from this experiment that blocking protein degradation has had any impact beyond stabilising the different levels of Myc protein being produced between the three conditions which does seem to somewhat correspond with their underlying mRNA levels? mTORc1 is not the only signalling pathway that impacts protein synthesis downstream of IL2 in CTL so an incomplete effect of mTORc1 on Myc expression doesn't really rule out a difference in protein synthesis between conditions. A better test is required to distinguish if there truly is a difference in Myc degradation rate between high and low pH... one possibility is to measure the rate of Myc loss over time while protein synthesis is blocked by cycloheximide.

3)

A measurement plotted throughout quite a few figures is "mean division number" from a CTV proliferation assay. Labelling of already activated lymphocytes to distinguish division peaks is not an established use for CTV. Clear peak resolution with CTV typically relies on uniform cell size/content (typically achieved by labelling naïve lymphocytes) to generate a narrow initial peak fluorescence or some kind of cell sorting to increase the uniformity of the cells being labelled (narrowing FSC to increase uniformity in activated lymphocytes shows some success for this though still doesn't give great peak resolution). As authors directly show in Fig 1E there are no distinct division peaks detected in their experiments. I don't see how it has been possible for an accurate measurement of mean division number to be made by this method for a labelled CTL culture even if using something like the flowjo fitting tool.

This is not to say that there isn't a difference in proliferation, it just needs to be measured/shown in a different way in order for the results to be robust and properly comparable (in some instances this is definitely required... it seems highly improbable that this level of CTV resolution could distinguish between an average division number of ~0.5 and ~0.1 like are plotted in Fig 5A for the pH 6.6 condition). The methods section indicates that cell number readings were taken during these experiments. Perhaps, the authors could show cell count data combined with some kind of indication of % viability in the cultures to try and deduce approximate culture expansion vs death or CTV gMFI could be plotted as a readout instead since this can be reasonably measured in these experiments (though to note even when cell division numbers are the same CTV fluorescence can be slightly different in cells of a difference size, so this confounding factor means a CTV fluorescence parameter is still not optimal for accurate measurement when proliferative differences are small). Measurements of cell cycle via e.g. EdU/BrdU incorporation for key conditions of interest would be most helpful for conditions where the comparison is important and differences are difficult to distinguish because they are small e.g +/-rapamycin in IL2 cultures with pH 7.4 vs 6.6. as in Fig 5A.

Minor points:

1) The authors should include specific information about what device/method they used to measure the pH of their starting media in this study. Did they measure pH more than once in their cultures? E.g. after the 24 hr TCR retriggering and 2 day TCR-independent expansion phase? It would be interesting/useful (though not essential) to know how much the pH has shifted in these cultures over this timeframe

2) The introduction/start of the results section would benefit from a more clearly explained and cited justification of why pH 6.6 has been chosen for these experiments. Also why pH 7 is included in some experiments? Are these pHs linked to a specific kind of cancer? All solid cancers?

3) The start of the results section would also benefit from a brief description of how the CTL are generated rather than just referring readers to a supplementary figure, given that these are the cell type in which all experiments are being performed.

4) Figures 3 and 5 are miniscule! These panels need to be larger for these figures to be readable at a normal level of magnification (or in a printed format).

Referee #3:

General summary and opinion about principal significance of the study, its questions and findings

This paper looks at how the acidic conditions that may be encountered during CD8 T-cell effector reactivation and activity in tissue affect CTL response to TCR and IL-2 stimulation. This paper adds to previous publications on T-cell function in acidic conditions by looking at how acidity regulates IL2 signalling pathways including ones other than STAT5 phosphorylation (particularly focusing on mTORC1 and Myc) and by looking at the effects of exposure to acidic conditions on amino acid metabolism.

It is shown that T-cells with expressing a low affinity TCR are less able to respond to restimulation in response to antigen in acidic conditions, while T-cells expressing a higher affinity TCR respond normally. It is shown that there is some impairment of IL-2 binding to the high affinity IL2 receptor in acidic conditions in the range that could be encountered in tissue, and signalling in response to IL2 is altered during acidic conditions, with a particular reduction in signalling downstream of mTORC1 and Myc expression. Inhibition of mTORC1 signalling in response to acidified media was independent of PKB, TSC2, RHEB, GATOR1 and AMPK signalling, and not due to a direct inhibition of mTOR kinase activity in acidic conditions. Intracellular levels of glutamine and glutamate were reduced in cells exposed to acidic media, and glutamine uptake was reduced in acidic conditions in full media but not in HBSS.

Specific major concerns essential to be addressed to support the conclusions

This would maybe just require some clarification, but in figure 6 it is shown that reducing extracellular glutamine reduces mTORC1 activity and Myc protein levels, and that glutamine transport is lower in CTL that have been cultured in acidic conditions. However, in the text (line 439), if I've understood correctly, reduced glutamine transport is excluded as a factor in the reduced mTORC1 activity and Myc levels because intracellular glutamine is not detected after a 10-fold reduction in glutamine levels in the media. Is it possible that glutamine is not detected because the assay isn't sensitive enough? It might also be useful to look at how quickly mTORC1 and Myc are affected by glutamine deprivation. Myc loss can happen quite quickly after removal of glutamine, which would maybe bring it into the timescale that mTORC1 and Myc are being affected by reduced pH.

Would it be possible to use your RNA seq data to look at expression of STAT5 targets vs Myc targets? This would strengthen the argument that there is a disproportionate loss of Myc compared to Jak/STAT signalling. I know it looks disproportionate by Western blot, but Western blot is only semi-quantitative and it can be a bit complicated comparing levels of different proteins rather than the same protein in different conditions. In figure 2B it looks like CD25 is induced by addition of IL2 in the normal pH samples (as you would expect as CD25 is a STAT5 target) and it not induced in the IL2 stimulated cells in lower pH media, which could suggest that the expression of STAT5 target genes is reduced. However, the flow is looking at protein abundance at the cell surface, so there are complications from internalisation and translation, so it would be good to look at the RNA levels.

Minor concerns that should be addressed

Non-stimulated control in figure 1E does not appear to be dividing. As this is a re-activation, shouldn't even the non-restimulated cells be dividing? Or have the CTL stopped dividing at this point in the culture?

Line 108: is the b a reference to the figure 1B? It is a bit confusing.

Do you have any way of validating that the Switch IL2 was produced successfully (i.e. sequence data to show that the intended mutations are there)?

In extended data figure 6C, is that band in the empty/ladder lane from your ladder?

In extended data figure 6G, there are two blots labelled pErk. The lower one looks like the total.

In line 462 the figure with the glutamine uptakes is 6k rather than 6j.

I'm not sure that there is enough evidence to say that mTOR is sensing acidification. From the data, it looks more likely that something upstream of mTOR is responding to acidic conditions and then regulating mTOR activity.

In line 201, I think this is supposed to be performed.

Any additional non-essential suggestions for improving the study

Have you tried any of your experiments in HPLM media? It is good to see what the effects are of lower pH with just HCl as this avoids the other effects that you might see with something like lactic acid, but it might be interesting to see if you see the effect

of acid is different when the concentrations of other metabolites are different.

Using your IL2 dose response experiments, could you estimate how much signal for the different proteins you would be expecting in the IL2 stimulated cells in the more acidic media given the estimated reduction in IL2 binding? Then you could estimate how much of the difference still needed to be explained by another mechanism.

Have you looked at SLC1A5 as the sensor for acidic conditions at all? Various SLC transporters are pH sensitive or require H⁺ to transport, so would potentially respond quite quickly to changes in acidity either intracellularly or extracellularly, and mTORC1 signalling and Myc can be regulated by nutrient uptake.

Do you see similar effects of acidity on initial TCR activation, for example inhibition of OT-1 TCR activation in response to low affinity peptides but normal TCR activation in response to the high affinity peptide?

Referee #4:

Acidity is one of the factors in the tumor microenvironment that suppresses T cell response. This study employed various complementary approaches to address how physiologically relevant low pH (acidity) systematically impacts IL-2 signaling and T-cell functionality. The authors did an incredible job providing a comprehensive view of how acidity dampens T-cell response by disrupting T cells' key signaling cascades and interfering with the metabolic landscape. The authors further highlighted an acidity-dependent IL-2 responsiveness as a critical mechanistic insight. Therefore, the current study provides a potentially new explanation of how acidity impacts T-cell functionality. It may also help us understand the immunosuppressive nature of the tumor microenvironment and thus develop new strategies to improve cancer immunotherapy.

I do not have any major concerns, but I have a couple of suggestions for authors to consider improving this study.

1) Conceptually, I encourage authors to explore further how acidity perturbs IL-2 responsiveness in T cells. Notably, recent studies have examined how acidity impacts Treg differentiation and function (PMID: 36788428; PMID: 33589820; etc.). Given the study's particular emphasis on IL-2 responsiveness, I encourage authors to compare the effects of acidity on IL-2 responsiveness in CTL vs. Treg since IL-2 signaling is critically involved in regulating Treg proliferation and functionality. Comparing how CTL and Treg respond to acidity may help define "IL-2 responsiveness".

2) Technically, authors should consider using HCL and lactic acid to adjust the media pH and then compare their effects on IL2 receptors (critical data presented in Figure 2). This may help reconcile their findings with others' results. However, it is not necessary to test the impact of lactic acid on CTLs extensively since several groups have studied the topic.

3) Modulating TSC2 or LKB is a critical effort to reverse acidity's effects on mTOR signaling and CTL proliferation. However, this approach failed to restore the c-Myc expression. A relatively straightforward approach would be overexpressing c-Myc (particularly with specific hot spot mutations that stabilize it) in CTLs and testing its impact on CTLs at low pH.

Referee #5:

The manuscript by de Silly and colleagues delineates the comprehensive mechanisms by which acidity inhibits T-cell function. They demonstrated that low pH induces significant disruptive changes in cytotoxic T lymphocytes (CTLs), impacting their responsiveness to IL-2, intracellular signaling pathways (mTOR and c-Myc), and amino acid metabolism. These changes collectively significantly impede T cell function, including proliferation, cytokine production, and cytotoxicity, particularly in lower affinity TCR T cells. Overall, the study is well-designed, and conclusions are well-supported by the data. Therefore, the study is of interest in the context of insight into the underlying mechanisms of T cell biology in the TME. However, several concerns need further addressing:

1. CTL function relies heavily on glycolysis. Moreover, the authors have shown that low pH inhibits the mTOR signaling pathway. It's worth exploring whether this inhibition could impact intracellular glycolysis in T cells and the potential mechanisms behind it should be addressed.

2. To comprehensively understand the impact of low pH on CTLs, it's essential to provide the global transcriptional profile. This will offer insights into the overall changes occurring in proliferation, signaling pathways, and metabolic processes involved in T cell remodeling under acidic conditions.

3. The low pH could induce epigenetic changes, potentially influencing the differentiation of T cell stemness (PMID: 36717749). Understanding the epigenetic alterations could shed light on how acidosis hampers the function of CTLs. The authors are also encouraged to explore or discuss the potential regulatory mechanisms through which acidosis affects the balance between T cell differentiation into effector and stem cell states.

4. It's important to elucidate how the extracellular acidic pH inhibits Myc transcription and proteasomal degradation. The authors should provide supporting evidence or engage in a discussion regarding potential mechanisms underlying this phenomenon.

Minor concerns:

1. In Fig. 6g, it is essential to indicate which carbon atom of the ^{13}C -glutamine was labeled. The number of specific labeled carbon atom in these indicated metabolites should be included in this figure, with detailed explanations provided in the legend.
2. The authors need to clarify why the low pH did not impact the uptake of ^{13}C -glutamine as depicted in Fig. 6g, whereas this effect was not observed in Fig. 6k.
3. It is recommended to replace the representation of p70S6K in Fig. 5c with a clearer image.
4. There is no description provided for Fig. 6k in the manuscript.
5. The authors cited too many review papers, and certain recent key references about acidity or lactate on T cell differentiation were missing.

Response from the Authors

We sincerely thank all of the referees for their time and careful review of our manuscript. We have comprehensively addressed their key comments with extensive additional experiments during the revision period. We believe that these new data and updated text significantly improve the clarity and quality of our manuscript. We thank you for your consideration of our revised work for publication in the EMBO Journal.

On behalf of the co-authors,

Referee #2 (Report for Author)

The manuscript "Acidity perturbs IL-2 responsiveness, mTORC1 and c-Myc in CD8+ T cells" seeks to examine the impact of reducing culture media pH on in vitro differentiated cytotoxic T lymphocyte (CTL) function, signalling and amino acid content. The motivation behind this investigation was to understand how the specific variable of acidity, which is typically low in solid tumours, impacts CTL signalling, to better inform strategies looking to reinvigorate T cell responses against cancer.

This is not the first study to examine the impact of extracellular pH on T cell function, with most of the key CTL cellular phenotypes reported within this study (reduced IFN γ , IL2 production, reduced cell proliferation and blunted CTL killing, though only under select conditions here) initially reported in previous works on this topic, which is outlined and suitably referenced by the authors in the text. This study does however represent a more detailed and extensive attempt than prior work to measure and test the cell signalling changes downstream of the TCR and IL2 when CTL are placed into reduced pH media.

Investigation into the effect of low pH on IL2 signalling find that: 1) consistent with less detailed previous work, IL2-receptor expression is reduced and JAK/STAT signalling is decreased, 2) show that mTORc1 activity is substantially reduced, which is consistent with what known about the effects of pH on cell signalling but has not to my knowledge been shown in CTL and 3) that cMyc expression is substantially decreased. The authors also find that strong TCR stimulation can over-ride the effects of low pH conditions on Myc expression but not mTORc1 signalling.

The authors try to pull apart how change in mTORc1 activity and Myc expression is being

regulated downstream of IL2. For Myc they ascribe this to a combination of transcript level and protein degradation rate. For mTORc1 they investigate several possible major upstream signalling pathways as well as control of amino acid content. While the authors put in considerable work on this question and are able to rule out a number of signalling pathways as individual mechanisms in their culture system, unfortunately they have not been able to pin down what the mechanism is here for pH control of mTORc1. To close the paper the authors look at changes in amino acid content due to low pH showing an increase in proline and decrease in glutamine/glutamate and find that glutamine uptake is reduced in low pH.

A strength of this study is that the authors interrogate the data quantitatively. This is a much better and more detailed look at the effects of pH on cell signalling in CTL than has been done previously. The use of a range of KO or inhibitor strategies to actually test relative likelihood of upstream pathways being explanatory for the effect of low pH on the cell signalling and proliferation phenotype versus just looking at correlative changes in expression is a particular strength.

We thank you very much for these comments on our work.

Some areas where the manuscript could be improved are outlined below:
Major comments:

1) The authors claim that IL2 dependent expression of Myc is impacted at the transcript level by low pH but that a difference in protein degradation rate is needed to explain the full reduction in Myc at a protein level.

Fig 4e is used as a justification that Myc protein and mRNA levels do not directly correspond. However, if I'm reading this correctly, the data plotted here for the pH6.6 condition seems to indicate that there were 3 samples where the Myc mRNA level was 80-100% of the level seen in the pH 7.4 + IL2 200 IU/mL condition ... yet in Fig 4C the average percentage for pH 6.6 is ~60% of the pH7.4 condition for saturating IL2 levels. There is only one data point in 4c that has large enough error bars for the pH 6.6 condition to contain these three data points (this is the 2000 IU/mL condition) though this would mean the 4th value would have to be very low to generate an average of ~60%. This doesn't make sense to me. Has there been some kind of plotting error? Could some of the raw measurements be included in a supplementary figure and/or could shape/size be used to help distinguish how the data points in 4d-e relate back to the IL2 concentration conditions in 4c? These are quite influential data points when it comes to looking at correlation with JAK/STAT signalling parameters and are the data points that give the strongest support to the idea of Myc protein vs mRNA not correlated in high vs low pH.

We thank the Reviewer for this comment which is correct. Ex-Fig. 4e was combining data from ex-Fig. 4C (now Fig. 5C) together with another set of experiment that

integrates 4 biological replicates from two other independent experiments where IL-2 dose-response was not performed at pH6.6 (only 0 and 200 IU/mL). Here are the graph results of these two set of experiments individually:

Which gives in combo (Set 1 + Set2):

We have removed this graph which we now realize is misleading. We conducted and included another set of experiments (please see Fig. 5E - kinetics of *Myc* mRNA with c-Myc protein levels in parallel) indicating that mRNA transcripts of *Myc* are in fact significantly involved in the lower c-Myc protein pattern observed at low pH. This is further supported by a new experiment in which we overexpressed wild type c-Myc (Fig. 6D; please see the response to reviewer #4 and the corresponding section in the manuscript).

2) An issue with regards protein degradation as a required mechanism to generate the pH dependent difference in Myc protein expression - while MG132 does increase Myc protein level in the low pH condition it also does this substantially in the pH7.4 + IL2 and pH7.4 -IL-2 conditions (Fig 4f), the latter of which has substantially less Myc mRNA than the pH 6.6. condition (Fig 4c). It is not clear to me from this experiment that blocking protein degradation has had any impact beyond stabilising the different levels of Myc protein being produced between the three conditions which does seem to somewhat correspond with their underlying mRNA levels? mTORc1 is not the only signalling

pathway that impacts protein synthesis downstream of IL2 in CTL so an incomplete effect of mTORc1 on Myc expression doesn't really rule out a difference in protein synthesis between conditions. A better test is required to distinguish if there truly is a difference in Myc degradation rate between high and low pH... one possibility is to measure the rate of Myc loss over time while protein synthesis is blocked by cycloheximide.

We thank the Reviewer for this insightful comment and experimental suggestion allowing one to more stringently assess c-Myc half-life by blocking protein synthesis. Indeed, this experiment precludes any bias in transcript amounts and, as a consequence, in translation rates. We first sought to determine the timing at which low pH has maximal impact on c-Myc (Fig. 5E). We assessed *Myc* mRNA and c-Myc protein levels in parallel over time and observed that they followed the same trend (lower *Myc* transcription is involved in lower c-Myc protein levels). Maximal impact on c-Myc levels occurred between 1h30 and 2h upon acidic treatment. Therefore, we conducted the CHX (cycloheximide) experiment upon a pH pre-treatment of CTLs for 1h30. At 1h30 we added CHX and looked at c-Myc levels at 0, 20, 40 and 60 minutes (Fig. 5G). We observed that low pH led to increased degradation rates of c-Myc and thus to lower c-Myc half-life (from 37 minutes to 23 minutes).

3) A measurement plotted throughout quite a few figures is "mean division number" from a CTV proliferation assay. Labelling of already activated lymphocytes to distinguish division peaks is not an established use for CTV. Clear peak resolution with CTV typically relies on uniform cell size/content (typically achieved by labelling naïve lymphocytes) to generate a narrow initial peak fluorescence or some kind of cell sorting to increase the uniformity of the cells being labelled (narrowing FSC to increase uniformity in activated lymphocytes shows some success for this though still doesn't give great peak resolution). As authors directly show in Fig 1E there are no distinct division peaks detected in their experiments. I don't see how it has been possible for an accurate measurement of mean division number to be made by this method for a labelled CTL culture even if using something like the flowjo fitting tool.

This is not to say that that there isn't a difference in proliferation, it just needs to be measured/shown in a different way in order for the results to be robust and properly comparable (in some instances this is definitely required... it seems highly improbable that this level of CTV resolution could distinguish between an average division number of ~0.5 and ~0.1 like are plotted in Fig 5A for the pH 6.6 condition). The methods section indicates that cell number readings were taken during these experiments. Perhaps, the authors could show cell count data combined with some kind of indication of % viability in the cultures to try and deduce approximate culture expansion vs death or CTV gMFI could be plotted as a readout instead since this can be reasonably measured in these experiments (though to note even when cell division numbers are the same CTV

fluorescence can be slightly different in cells of a difference size, so this confounding factor means a CTV fluorescence parameter is still not optimal for accurate measurement when proliferative differences are small). Measurements of cell cycle via e.g. EdU/BrdU incorporation for key conditions of interest would be most helpful for conditions where the comparison is important and differences are difficult to distinguish because they are small e.g. +/-rapamycin in IL2 cultures with pH 7.4 vs 6.6. as in Fig 5A.

It is true that CTV peaks are not well resolved with CTLs as they comprise cells of various sizes. Unfortunately, we did not assess absolute cell number for “proliferation” experiments, but for “expansion” experiments. Nevertheless, we believe the CTV experiments we show provide important and reliable read-outs, even if not providing a precise and absolute cell division number. They enable us to uncouple cell division from cell death (something that cannot be done properly while looking at overall cell expansion even when adding a % cell death readout). Notably, the decrease in cell proliferation at low pH is confirmed many times later on in the manuscript when we investigate total cell expansion and cell death using various KD and KO.

Concerning the methodology, we do an estimation of number of cell division based on the unstimulated condition that gives the maximal fluorescence (cf. picture below, empty histograms: 10^5): then we gate 1st division for cells starting at value of fluorescence below this value down to half this value (one division should dilute fluorescence at maximum twice- cf. picture below: $10^5 \div 2 = 5 \times 10^4$; therefore cells with fluorescence that falls between 5×10^4 and 10^5 are considered as in 1st division), then we start the gating for the second division using the same principle (cf. picture below: $5 \times 10^4 \div 2 = 2.5 \times 10^4$; therefore cells with fluorescence that falls between 2.5×10^4 and 5×10^4 are considered as in 2nd division), etc. From these fractions, we calculate the mean cell division number.

As a side note, but still important, the purpose of ex-Figure 5A (now Fig. EV3A) was to show that inhibiting mTORC1 can lead to the same proliferation defects observed at pH6.6. Even if we show stars/statistical significance for pH7.4+rapa vs pH6.6 or pH6.6 vs pH6.6+rapa, we are not claiming that this represents a biologically significant difference.

We want to highlight that the goal of these experiments was not to precisely determine cell divisions, but to compare whether the cells divide to the same extent at pH7.4 versus pH6.6, which is not the case.

Because adding extra CTV MFI figures represents an extra layer of complexity for the reader and it is easier/clearer to understand with cell division number figures, what we can propose is to warn readers about the limitation of our cell number division estimation in the Methods section. We state: "Importantly, *since CTLs do not represent a uniform cell size population, CTV incorporation does not give well resolved peaks. Therefore, we want to emphasize the fact that our estimation of cell division number gives an approximate (not precise) cell division number.*"

Minor

points:

1) The authors should include specific information about what device/method they used to measure the pH of their starting media in this study. Did they measure pH more than once in their cultures? E.g. after the 24 hr TCR retriggering and 2 day TCR-independent expansion phase? It would be interesting/useful (though not essential) to know how much the pH has shifted in these cultures over this timeframe

We used a classical pH meter. This information has been added in the Methods section. We agree this would have been an interesting result, but we did not measure pH repetitively in the supernatants.

2) The introduction/start of the results section would benefit from a more clearly explained and cited justification of why pH 6.6 has been chosen for these experiments. Also why pH 7 is included in some experiments? Are these pHs linked to a specific kind of cancer? All solid cancers?

The rationale of using pH6.6 is somewhat arbitrary (even if frequently used in other studies), but is supported by the fact that one can encounter this value in tumors (Tannock and Rotin, Cancer Res 1989). Importantly however, it allows one to observe a clear impact of acidity on the CTLs without leading to massive cell death. At the start of the study, pH7 and pH6.2 were included in order to establish a dose-response to pH. We briefly explained the use of these pH in the manuscript at the

beginning of the Results section:” *Tumor pH is highly variable (Feng et al, 2024; Tannock & Rotin, 1989) and we thus set out to assess the impact of pH7.4 as a physiological control, pH7, and pH6.6, a level of acidity that can be found in tumors (and often explored in pH studies). “*

3) The start of the results section would also benefit from a brief description of how the CTL are generated rather than just referring readers to a supplementary figure, given that these are the cell type in which all experiments are being performed.

Thank you for this suggestion, which we have followed. We stated: “*Briefly, naïve CD8⁺ T cells were activated for two days with anti-CD3 and anti-CD28 antibodies and expanded over several days in the presence of high-dose murine IL-2 (200 IU/mL) to favor effector differentiation (Ross & Cantrell, 2018).*”

4) Figures 3 and 5 are miniscule! These panels need to be larger for these figures to be readable at a normal level of magnification (or in a printed format).

Indeed, the quality should have been better- the submission process lowered the quality of the figures.

Given that we had to format the figure panels to comply with the journal guidelines, we carried out a deep reformatting of the figure panels. This allowed us to split these big panels and to increase the size the graphs. We believe the figures are now much easier to read.

Referee #3 (Report for Author)

This paper looks at how the acidic conditions that may be encountered during CD8 T-cell effector reactivation and activity in tissue affect CTL response to TCR and IL-2 stimulation. This paper adds to previous publications on T-cell function in acidic conditions by looking at how acidity regulates IL2 signalling pathways including ones other than STAT5 phosphorylation (particularly focusing on mTORC1 and Myc) and by looking at the effects of exposure to acidic conditions on amino acid metabolism.

It is shown that T-cells with expressing a low affinity TCR are less able to respond to restimulation in response to antigen in acidic conditions, while T-cells expressing a higher affinity TCR respond normally. It is shown that there is some impairment of IL-2 binding to the high affinity IL2 receptor in acidic conditions in the range that could be encountered in tissue, and signalling in response to IL2 is altered during acidic conditions, with a particular reduction in signalling downstream of mTORC1 and Myc expression. Inhibition of mTORC1 signalling in response to acidified media was independent of PKB, TSC2, RHEB, GATOR1 and AMPK signalling, and not due to a direct inhibition of mTOR kinase activity in acidic conditions. Intracellular levels of glutamine and glutamate were reduced in cells exposed to acidic media, and glutamine uptake was reduced in acidic conditions in full media but not in HBSS.

Specific major concerns essential to be addressed to support the conclusions

This would maybe just require some clarification, but in figure 6 it is shown that reducing extracellular glutamine reduces mTORC1 activity and Myc protein levels, and that glutamine transport is lower in CTL that have been cultured in acidic conditions. However, in the text (line 439), if I've understood correctly, reduced glutamine transport is excluded as a factor in the reduced mTORC1 activity and Myc levels because intracellular glutamine is not detected after a 10-fold reduction in glutamine levels in the media. Is it possible that glutamine is not detected because the assay isn't sensitive enough?

The assay by mass spectrometry is in fact very sensitive and we do not think that the glutamine absence we observed upon 10-fold reduction in extracellular glutamine is due to issues with sensitivity.

Notably, we observe that the decrease in intracellular glutamine content at low pH is not as profound as the one observed when we lower extracellular glutamine by 10-fold. In spite of this, Myc levels and mTORC1 activity is much higher upon 10-fold reduction in extracellular glutamine than at low pH.

It might also be useful to look at how quickly mTORC1 and Myc are affected by glutamine deprivation. Myc loss can happen quite quickly after removal of glutamine, which would maybe bring it into the timescale that mTORC1 and Myc are being affected by reduced pH.

We agree this could be of interest. However, since we ruled out glutamine involvement in c-Myc and mTORC1 patterns at low pH, we respectfully do not think this experiment would be relevant to decipher pH impact (it would provide glutamine-centric information which is beyond the scope of the study).

Would it be possible to use your RNA seq data to look at expression of STAT5 targets vs Myc targets? This would strengthen the argument that there is a disproportionate loss of Myc compared to Jak/STAT signalling. I know it looks disproportionate by Western blot, but Western blot is only semi-quantitative and it can be a bit complicated comparing levels of different proteins rather than the same protein in different conditions. In figure 2B it looks like CD25 is induced by addition of IL2 in the normal pH samples (as you would expect as CD25 is a STAT5 target) and it not induced in the IL2 stimulated cells in lower pH media, which could suggest that the expression of STAT5 target genes is reduced. However, the flow is looking at protein abundance at the cell surface, so there are complications from internalisation and translation, so it would be good to look at the RNA levels.

In ex-Extended Data Figure 4d (now Appendix Fig. S5B), we already performed a gene set enrichment analysis (GSEA) to determine pathways that are up- or down-regulated in an unbiased fashion, and the Myc pathway came out but not Jak/STAT. It is important to highlight that GSEA is a more robust and comprehensive method for analyzing pathway activity compared to manually checking the expression levels of individual targets. GSEA considers the entire set of genes within a pathway and evaluates their collective behavior, which provides a more accurate representation of pathway activation or repression. This approach mitigates the bias and variability that can arise from focusing on individual genes and allows for the detection of subtle yet coordinated changes in gene expression that might be missed with a gene-by-gene analysis. Therefore, the GSEA results we presented offer a stronger and more reliable evidence of the differential regulation of Myc and Jak/STAT signaling pathways in our study. However, we believe Jak/STAT still is disturbed with time, given the results we show concerning their direct phosphorylation state. We did not compare levels of two different proteins, but their lowering (% reduction) due to pH and/or IL-2 amounts.

Notably, we revised significantly our conclusions concerning the importance of Myc mRNA in c-Myc protein levels at low pH given the new Myc mRNA and c-Myc protein level kinetics we carried out and display in Fig. 5E (please see response to reviewer #2 and the corresponding section in the manuscript), and the results we obtained by overexpressing wild type c-Myc (Fig. 6D; please see response to

reviewer #4 and the corresponding section in the manuscript). We are now convinced that *Myc* mRNA levels have a significant role in the final c-Myc protein levels obtained at low pH.

Minor concerns that should be addressed

Non-stimulated control in figure 1E does not appear to be dividing. As this is a re-activation, shouldn't even the non-restimulated cells be dividing? Or have the CTL stopped dividing at this point in the culture?

Indeed, at this point (at least 5 days after initial priming), the CTLs depend on exogenous cytokine to proliferate/divide. In non-stimulated control, there is no cytokine added and no re-activation stimulus.

Line 108: is the b a reference to the figure 1B? It is a bit confusing.

Thank you for noticing this - it has been fixed.

Do you have any way of validating that the Switch IL2 was produced successfully (i.e. sequence data to show that the intended mutations are there)?

The plasmid used to produce the cytokine was indeed sequenced. Notably, we have a high level of experience in producing IL-2 and variants in our lab (please refer to our BioRxiv manuscript: doi.org/10.1101/2023.05.24.541283).

In extended data figure 6C, is that band in the empty/ladder lane from your ladder?

Yes, it is. The band seen in the "empty/ladder" lane is coming from the ladder. We usually add a ladder in these lanes because they are often distorted in the 26well precast gel that we are using. It often happens that a band appears in the ladder lane for some primary antibodies.

In extended data figure 6G, there are two blots labelled pErk. The lower one looks like the total.

Thank you for noticing this - it has been fixed.

In line 462 the figure with the glutamine uptakes is 6k rather than 6j.

Thank you for noticing this - it has been fixed.

I'm not sure that there is enough evidence to say that mTOR is sensing acidification. From the data, it looks more likely that something upstream of mTOR is responding to acidic conditions and then regulating mTOR activity.

It is indeed possible – even if we ruled out most of the major regulator in mTORC1 activity and given the rapidity to which acidity lowers mTORC1 activity.

We believe the problematic sentences, which are probably a bit too much an overstatement, are these ones in the discussion:

"Hence, we conclude that along with its well-known roles in sensing nutrient and energy sufficiency, mTORC1 acts as a sensor of pH" and "Importantly, we have demonstrated a role for mTORC1 as a bona fide sensor of acidity in primary T cells."

Therefore, the second sentence has been removed and the first sentence has been changed to this one:

"Hence, we conclude that along with its well-known roles in sensing nutrient and energy sufficiency, our data suggests that mTORC1 also acts as a sensor of pH".

In line 201, I think this is supposed to be performed.
Thank you for noticing this - it has been fixed.

Any additional non-essential suggestions for improving the study

Have you tried any of your experiments in HPLM media? It is good to see what the effects are of lower pH with just HCl as this avoids the other effects that you might see with something like lactic acid, but it might be interesting to see if you see the effect of acid is different when the concentrations of other metabolites are different.

We did not try any of our experiments in HPLM. However, the medium we are using contains a lower amount of glucose (which is more physiological) as compared to typical DMEM/RPMI media.

Use of lactic acid or assessment of pH impact when concentrations of other metabolites are different would be of interest. However, the purpose of this study was really to analyze the "pure" impact of low pH on T cells. We believe our study could serve as a basis to modify other parameters in conjunction to pH lowering in future studies.

Using your IL2 dose response experiments, could you estimate how much signal for the different proteins you would be expecting in the IL2 stimulated cells in the more acidic media given the estimated reduction in IL2 binding? Then you could estimate how much of the difference still needed to be explained by another mechanism.

Indeed, this could represent an interesting way of showing the results.

However, we believe it is easier/better for the reader to not go into too much modeling as we already did it with interpretations from correlation curves from ex-Fig3e (Now Fig. EV2D). It is quite compelling to observe that together with the 70% binding left as measured in Fig. 2D that is completely in line with the lower JAK/STAT phosphorylation pattern observed in ex-Fig. 3D (now Fig. 3C).

Have you looked at SLC1A5 as the sensor for acidic conditions at all? Various SLC transporters are pH sensitive or require H⁺ to transport, so would potentially respond quite quickly to changes in acidity either intracellularly or extracellularly, and mTORC1 signalling and Myc can be regulated by nutrient uptake.

We tried to look at SLC1A5 expression by western blot, but we were not able to detect it.

In fact, we believe SLC1A5 is the major transporter that is disturbed by low pH, but we only suggest its involvement as we do not have direct proof. That is why we wrote in the result section: *"Interestingly, we observed that increasing doses of extracellular glutamine lowered intracellular levels of serine and threonine (Appendix Fig. S10A) which might reflect CTL reliance on the neutral amino acid transporter alanine serine cysteine transporter 2 (ASCT2), a transporter which imports neutral amino acids such as alanine and glutamine (its primary role) in exchange for intracellular amino acids."*

Also, in the discussion, we stated: *"We also found that increased glutamine levels in the culture media was associated with lower intracellular levels of serine and threonine suggesting that low pH may disrupt substrate specificity and/or activity of ASCT2 (SLC1A5)"*.

Do you see similar effects of acidity on initial TCR activation, for example inhibition of OT-1 TCR activation in response to low affinity peptides but normal TCR activation in response to the high affinity peptide?

This would be interesting, but we did not try.

Referee #4 (Report for Author)

Acidity is one of the factors in the tumor microenvironment that suppresses T cell response. This study employed various complementary approaches to address how physiologically relevant low pH (acidity) systematically impacts IL-2 signaling and T-cell functionality. The authors did an incredible job providing a comprehensive view of how acidity dampens T-cell response by disrupting T cells' key signaling cascades and interfering with the metabolic landscape. The authors further highlighted an acidity-dependent IL-2 responsiveness as a critical mechanistic insight. Therefore, the current study provides a potentially new explanation of how acidity impacts T-cell functionality. It may also help us understand the immunosuppressive nature of the tumor microenvironment and thus develop new strategies to improve cancer immunotherapy.

Many thanks for your kind comments on our study.

I do not have any major concerns, but I have a couple of suggestions for authors to consider improving this study.

1) Conceptually, I encourage authors to explore further how acidity perturbs IL-2 responsiveness in T cells. Notably, recent studies have examined how acidity impacts Treg differentiation and function (PMID: 36788428; PMID: 33589820; etc.). Given the study's particular emphasis on IL-2 responsiveness, I encourage authors to compare the effects of acidity on IL-2 responsiveness in CTL vs. Treg since IL-2 signaling is critically involved in regulating Treg proliferation and functionality. Comparing how CTL and Treg respond to acidity may help define "IL-2 responsiveness".

We agree that exploring Tregs would be very interesting but think that this goes beyond the scope of our comprehensive study focused on the impact of low pH on CTLs. (Exploring the impact of low pH on Tregs would be an interesting follow-up study requiring major experimental work.)

2) Technically, authors should consider using HCL and lactic acid to adjust the media pH and then compare their effects on IL2 receptors (critical data presented in Figure 2). This may help reconcile their findings with others' results. However, it is not necessary to test the impact of lactic acid on CTLs extensively since several groups have studied the topic.

It is an interesting point raised by the Reviewer. It is true that many studies claiming lactic acid impact do not clearly rule out that the effect observed is purely pH-mediated. However, we believe that untangling the impact of lactic acid is not

the focus of our study and would require that many parameters beyond IL-2R expression be assessed.

3) Modulating TSC2 or LKB is a critical effort to reverse acidity's effects on mTOR signaling and CTL proliferation. However, this approach failed to restore the c-Myc expression. A relatively straightforward approach would be overexpressing c-Myc (particularly with specific hot spot mutations that stabilize it) in CTLs and testing its impact on CTLs at low pH.

This is a very good suggestion. Notably, we do not know which residues are linked to the increased c-Myc degradation at low pH (we demonstrated that it does not appear linked to T58/S62 phosphorylation), hence we decided to overexpress wild type c-Myc. However, since we show that mTORC1 and c-Myc defects do not appear to be connected, we explored enhancing mTORC1 activity by knocking out TSC2 and to improve c-Myc levels by overexpressing it at the same time. To do so we used a single retroviral vector in OT-I x CRISPR/Cas9 CTLs which encodes a TSC2 sgRNA under a U6 promoter, wild-type c-Myc under a PGK promoter and a Thy1.1 reporter gene (to select transduced CTLs) downstream of an IRES (our attempts to overexpress polycistronically c-Myc and Thy1.1 with a 2A skipping site instead of an IRES was leading to issues in c-Myc size and activity). We looked at mTORC1 activity (p-p70S6K), c-Myc levels (mRNA and protein) as shown in Fig. 6D, and at cell expansion/viability (Fig. 6E). c-Myc overexpression significantly improved CTL expansion at neutral pH, but it was not sufficient to restore CTL expansion at low pH, even when combined to mTORC1 enhancement via TSC2 knockout. In fact, combination of c-Myc overexpression and TSC2 knockout led to a lower CTL viability. Notably, these experiments subtly change the conclusion that we had concerning the mechanism by which low pH alter c-Myc protein levels as we were able to overexpress wild type c-Myc at low pH. However, one still has to consider one limitation: the level at which *Myc* was overexpressed as compared to endogenous induction of *Myc* by IL-2. It is likely that post-transcriptional regulation is hindered in these artificial/non-physiological settings by saturating c-Myc degradation events.

Referee #5 (Report for Author)

The manuscript by de Silly and colleagues delineates the comprehensive mechanisms by which acidity inhibits T-cell function. They demonstrated that low pH induces significant disruptive changes in cytotoxic T lymphocytes (CTLs), impacting their responsiveness to IL-2, intracellular signaling pathways (mTOR and c-Myc), and amino acid metabolism. These changes collectively significantly impede T cell function, including proliferation, cytokine production, and cytotoxicity, particularly in lower affinity TCR T cells. Overall, the study is well-designed, and conclusions are well-supported by the data. Therefore, the study is of interest in the context of insight into the underlying mechanisms of T cell biology in the TME. However, several concerns need further addressing:

We thank the reviewer for these favorable comments.

1. CTL function relies heavily on glycolysis. Moreover, the authors have shown that low pH inhibits the mTOR signaling pathway. It's worth exploring whether this inhibition could impact intracellular glycolysis in T cells and the potential mechanisms behind it should be addressed.

Indeed, this is of interest. However, we believe it goes beyond the scope of our manuscript, given that we have focused to identify the cause(s) and not the consequences of mTORC1/c-Myc downregulation, which can be broad. Furthermore, it is already supported by the literature that mTORC1 promotes glycolysis, and Wu et al Nat Commun 2020 shows that acidity lowers glycolysis.

2. To comprehensively understand the impact of low pH on CTLs, it's essential to provide the global transcriptional profile. This will offer insights into the overall changes occurring in proliferation, signaling pathways, and metabolic processes involved in T cell remodeling under acidic conditions.

In fact, this is one of the first experiment we carried out in order to guide our investigation. Disturbed pathways are shown in Appendix Fig. S5B. However, the changes are so strong (which makes sense given the broad impact of IL-2R signaling, mTORC1 and c-Myc) that we were not able to identify something of interest with this data, except the confirmation that acidity impacted c-Myc targets and mTORC1 pathways.

3. The low pH could induce epigenetic changes, potentially influencing the differentiation of T cell stemness (PMID: 36717749). Understanding the epigenetic alterations could shed light on how acidosis hampers the function of CTLs. The authors are also encouraged to explore or discuss the potential regulatory mechanisms through which

acidosis affects the balance between T cell differentiation into effector and stem cell states.

Indeed, we missed this paper which has now been made mention to in our discussion. We stated in the discussion: “Notably, it has been recently reported that acidity preserves stemness (Cheng et al, 2023), in line with the implicated roles of IL-2R signaling, c-Myc and mTORC1 in effector cell versus stem cell/memory differentiation (Ross & Cantrell, 2018; Verbist et al, 2016), all of which were found to be perturbed in our study investigating changes in CTLs under acidic conditions. The impact of low pH on T cells may be even more pronounced in the presence of lactate (Feng et al, 2022), a metabolite often secreted at the same time as protons, and is of interest to explore in future studies.”

4. It's important to elucidate how the extracellular acidic pH inhibits Myc transcription and proteasomal degradation. The authors should provide supporting evidence or engage in a discussion regarding potential mechanisms underlying this phenomenon.

This is indeed what we aimed to elucidate, even if we did not find the direct targets involved concerning proteasome degradation.

For transcription, it appears to be due to lower IL-2R signaling. Concerning the increased proteasomal degradation, it might be a consequence of an increased activity of the proteasome at lower pH, given that intracellular pH is acidified. We added a sentence in the Discussion section: “Our data indicate that acidity might improve proteasome activity, as previously proposed by others (Rackova & Csekes, 2020; Zund et al, 1997)”

Minor concerns:

1. In Fig. 6g, it is essential to indicate which carbon atom of the ¹³C-glutamine was labeled. The number of specific labeled carbon atom in these indicated metabolites should be included in this figure, with detailed explanations provided in the legend.

Thank you for this comment. As suggested, we added this graph in Fig. EV4E.

2. The authors need to clarify why the low pH did not impact the uptake of ¹³C-glutamine as depicted in Fig. 6g, whereas this effect was not observed in Fig. 6k.

Even if the uptake of glutamine is decreased at low pH, extracellular glutamine is still up taken. Since the whole pool (almost 100% as you can see on the Figs. 8D and EV4E) of intracellular glutamine is derived from extracellular glutamine, it is not surprising that even at low pH, 100% of glutamine is ¹³C-labelled (even if it represents 60-70% of the totality up taken at neutral pH).

3. It is recommended to replace the representation of p70S6K in Fig. 5c with a clearer image.

As suggested, we increased the contrast of this blot.

The blots shown here are all coming from the same sample. It often happens that one staining looks less good. We decided to pick this sample because it is representative of most if not all of the molecules we looked at in these sets of experiments. Unfortunately, the p70S6K staining is not as beautiful as we would have liked, but we believe it is not problematic given there are other total protein showing protein loading was identical (which is the goal of the p70S6K staining). Furthermore, we believe results concerning p-p70S6K (%) are clear and even confirmed in another set of experiments (Fig. EV3C, sgCTRL+Akti).

4. There is no description provided for Fig. 6k in the manuscript.

Thank you for noticing it. There was a typo that we corrected.

5. The authors cited too many review papers, and certain recent key references about acidity or lactate on T cell differentiation were missing.

Thank you for pointing this out. We believe we have now included most of the relevant literature for the scope of our study. It is true we missed out some recent articles that are now added, including Feng et al., Nat Commun 2022, Chen et al., Nat Metab 2023, Feng et al. Nat Biomed Eng 2024.

Dear Dr. Vuillefroy de Silly,

Thank you for the submission of your revised manuscript to The EMBO Journal and for your patience during peer review. It has now been seen by the four original referees who previously assessed the earlier version of your manuscript, and we have received the complete set of their comments. While the referees recognize that the majority of their previously raised concerns have been successfully addressed in the revised manuscript, there are still a few remaining points by referees #2 and #3 that we would like you to fully address in a minor revision before we can accept the manuscript for publication in The EMBO Journal. Please also include in your resubmission a detailed point-by-point response to the referee comments explaining all changes to the manuscript.

From the editorial side, there are also a few changes and corrections that we need from you:

- Please note that no more than 5 keywords can be listed after the Abstract.
- The Materials and Methods need to be described in the manuscript using our "Structured Methods" format, which is now required for all research articles. According to this format, the Materials and Methods section includes a "Reagents and Tools Table" -listing key reagents, experimental models, software and relevant equipment and including their sources and relevant identifiers- followed by a "Methods and Protocols" section describing the methods using a step-by-step protocol format. The aim is to facilitate adoption of the methodologies across labs. More information on this format as well as a template (.docx) for the "Reagents and Tools Table" can be found in our author guide:
- The author contributions statement should be removed from the manuscript file. Instead, we now use CRediT to specify the contributions of each author in the journal submission system. Please feel free to use the free text box to provide more detailed descriptions during submission. See also our guide to authors for more information:
<https://www.embopress.org/page/journal/14602075/authorguide#authorshipguidelines>.
- As per our journal's policy, "data not shown" (stated on page 11 of your manuscript) is not permitted. All data referred to in the paper should be displayed in the main or Expanded View figures, or in the Appendix. Please add these data or change the text accordingly if these data are not central to the study and its conclusions, or properly cite the respective published sources if these data can be found elsewhere.
- Please make sure that all text is legible in your synopsis image, at its final dimensions (width: 550 pixels, the height is variable in the range 300-600 pixels).
- During our standard Figure checks, we detected possible blot re-use between Figure EV2 and Appendix Figure S3 (GAPDH and actin bands). Please clarify and correct the figures if necessary (and describe any changes in your cover letter). Please note that if the particular experimental setup justifies the re-use, it must still be explicitly stated in the figure legends.
- Please define the annotated p values ***** as well as provide the exact p-values for the same in the legend of Figure 5b, as appropriate.
- Please note that the exact p values are not provided in the legends of Figures 1c-h; 2a-d; 3a-b; 5f-g; 6a-b, e; 7d; 8a, h; EV 1f; EV 3a, c-e; EV 4b.

Please also note that as part of the EMBO publications' Transparent Editorial Process, The EMBO Journal publishes online a Peer Review File along with each accepted manuscript. This File will be published in conjunction with your paper and will include the referee reports, your point-by-point response and all pertinent correspondence relating to the manuscript. You can opt out of this by letting the editorial office know (contact@embojournal.org). If you do opt out, the Peer Review File link will point to the following statement: "No Peer Review File is available with this article, as the authors have chosen not to make the review process public in this case."

We look forward to seeing a final version of your manuscript as soon as possible. Please use this link to submit your revision:
<https://emboj.msubmit.net/cgi-bin/main.plex>

Best regards,

Ioannis

Ioannis Papaioannou, PhD

Referee #2:

I'm happy that the authors have nicely addressed major suggestions 1-2 from my previous review and all my minor comments. Some of the new experiments, particularly those looking at Myc regulation really clear up some questions from the previous version of the manuscript. The manuscript has improved in both scientific quality and readability during the revision process.

Ongoing point of concern:

I'm still not satisfied that it's appropriate to graph CTV fluorescence as a parameter called "mean division number" throughout the paper. To be clear, I agree with the authors CTV-style assays are needed to pull apart survival vs proliferation questions. I also agree that the extent of CTV dilution in this instance should be indicative of the extent of proliferation (with one caveat mentioned below) and I understand this lines up with cell counts later in the paper - I think this is all fine and I believe that acidity is impacting cell proliferation.

My issue is the use of the term 'mean division number' versus what has actually been gated/calculated/plotted/described within the manuscript. More specifically:

- 1) the authors don't actually describe how they set their gates and calculate 'mean division number' in their methods or anywhere else in the manuscript. The term 'mean division number' normally refers to one of two fairly specific things with the precise definitions depending on whether the person calculating it decides to correct for the effect of cell expansion or not. Both the specifics of how the gating has been done and how the parameter has been calculated is necessary in order to interpret the data presented.
- 2) The estimated placement of the division gates explained in the author response is not quite correctly calculated - CTV dilution is never quite perfectly half of the prior peak level due to the impact of cellular autofluorescence. To mathematically estimate CTV dilution gate positions based on halving of dye intensity you technically require a non-labelled background sample for every condition assayed. The calculation for the halving dye intensity = $(\text{undivided fluorescence} - \text{background fluorescence}) / 2^{\text{division number}} + \text{background fluorescence}$.
- 3) This non-labelled control is actually something I neglected to mention in my previous review and is the caveat I mentioned above. I think it would be valuable to show at least once that acidity doesn't impact the background fluorescence of CTL in the CTV channel in order to demonstrate that the CTV fluorescence intensity between conditions can be reasonably compared as an indication of dye dilution/extent of proliferation - otherwise shifts in CTV fluorescence between conditions could always be attributable to an effect of acidity on cell autofluorescence rather than the extent of cell division.

Given these points, my suggestion for how to represent the data is to plot CTV fluorescence of the treatment conditions as a fraction of the unstimulated/undivided condition. If you want it to look similar to the mean division number plots it could be plotted as $-\log_2$ of the fraction, with the axis label something along the lines of "estimated proliferation ($-\log_2$ CTV dilution)". In this way the data could be quantitatively plotted and compared in easy to read graphs without having to make any assumptions about where the division gates should be.

Minor points:

1. I'm confused about line 91-92. It says that pH 6.6 doesn't preferentially block CTL in a cell cycle stage but Appendix Fig S1D does show a higher proportion of cells in G0/G1 in low pH when cultured in IL2? Wouldn't this mean the cells are preferentially blocked in G0/G1?
2. In line 260 - it mentions a drop in aspartate and references Fig 8A and EV4A but I can't see aspartate in either figure? Is there a graph missing or a figure reference that has been left out?

Referee #3:

Review - Acidity perturbs IL-2 responsiveness, mTORC1 and c-Myc in CD8+ T cells

General summary

This paper focuses on how acidic conditions affect the function of CD8 T-cells. It was found that cytotoxicity was reduced in response to lower affinity antigen/TCR binding, and that CD8 T cells cultured in a low pH had reduced cytokine production in response to stimulation and proliferation was reduced. There were changes in IL2 receptor protein abundance and IL2 binding. Signalling through the mTORC1 pathway was reduced at low pH; and levels of Myc protein were reduced. Reduction in mTORC1 signalling could not be attributed to known upstream regulators. Reduction in Myc protein involved changes in transcription and degradation. Metabolic changes including reduction in glutamine and glutamate were also seen in CTL exposed to low pH.

Major concerns

My main concern is relying on the Western blot data to compare activity through different signalling pathways. As different antibodies are being used for the different proteins, has anything been done to show that a similar reduction in a protein produces a similar drop in signal for the things you are comparing?

Also, with the GSEA, I would have thought that the Jak-STAT pathway is a lot broader than the Myc pathway one, as it includes many different cytokines with various functions, so not all of the proteins in the Jak-STAT pathway are going to be involved in IL-2 signalling.

Minor concerns

No additional minor concerns.

Non-essential suggestions

No additional non-essential suggestions.

Referee #4:

I am satisfied with the revised manuscript and have no further comments.

Referee #5:

The authors have addressed most of my concerns.

Referee #2:

I'm happy that the authors have nicely addressed major suggestions 1-2 from my previous review and all my minor comments. Some of the new experiments, particularly those looking at Myc regulation really clear up some questions from the previous version of the manuscript. The manuscript has improved in both scientific quality and readability during the revision process.

We thank the reviewer for these comments.

Ongoing point of concern:

I'm still not satisfied that it's appropriate to graph CTV fluorescence as a parameter called "mean division number" throughout the paper. To be clear, I agree with the authors CTV-style assays are needed to pull apart survival vs proliferation questions. I also agree that the extent of CTV dilution in this instance should be indicative of the extent of proliferation (with one caveat mentioned below) and I understand this lines up with cell counts later in the paper - I think this is all fine and I believe that acidity is impacting cell proliferation.

We believe there might have been a misunderstanding in the way that we are calculating "mean cell division number". We are not using CTV MFI for the calculation. Rather we are using gates that encompass a fluorescence of x and $x/2$ (gate 1 = 1 division), $x/2$ and $x/4$ (gate 2 = 2 divisions), etc. x is given by unstimulated cells without stimulation: it represents the value of fluorescence where most of the cells (>80%) have a higher value (which means 0 division). Then we recover the percentage of cells upon stimulation that are in each of these gates: e.g., 20% were in gate 0 (fluorescence above x ; 0 division), 20% in gate 1, 50% in gate 2, and 10% in gate 3. Then we calculate mean cell division number: $(20\% \times 0) + (30\% \times 1) + (40\% \times 2) + (10\% \times 3) = 1.4$ divisions.

My issue is the use of the term 'mean division number' versus what has actually been gated/calculated/plotted/described within the manuscript. More specifically:

1) the authors don't actually describe how they set their gates and calculate 'mean division number' in their methods or anywhere else in the manuscript. The term 'mean division number' normally refers to one of two fairly specific things with the precise definitions depending on whether the person calculating it decides to correct for the effect of cell expansion or not. Both the specifics of how the gating has been done and how the parameter has been calculated is necessary in order to interpret the data presented.

Thank you for noting this. It is true that we did not describe the way in which we are gating and calculating 'mean division number' in the manuscript. A paragraph has been added in the Methods section, including a figure that shows the gating strategy to infer the fraction (%) of cells at a defined, yet estimated, division number.

2) The estimated placement of the division gates explained in the author response is not quite correctly calculated - CTV dilution is never quite perfectly half of the prior peak level due to the impact of cellular autofluorescence. To mathematically estimate CTV dilution gate positions based on halving of dye intensity you technically require a non-labelled background sample for every condition assayed. The calculation for the halving dye intensity = $(\text{undivided fluorescence} - \text{background fluorescence}) / 2^{\text{division number}} + \text{background fluorescence}$.

We agree that the way in which we are calculating cell division number is not perfect. Indeed, since we start with CTLs that are heterogenous in size, since asymmetric division occurs (Bocharov et al, "Asymmetry of Cell Division in CFSE-Based Lymphocyte Proliferation Analysis", Front Immunol 2013), and given protein (to which CTV is covalently linked) turnover, CTV fluorescence (and any other CFSE-like dyes) will never be perfectly diluted by 2 at each round of division.

The formula given is interesting and indeed more accurate. However, we do not have the background fluorescence value for each experiment. Nevertheless, given the strength (100-1000x stronger than background fluorescence when there is no division) of CTV fluorescence (please see the comment below), we consider that background fluorescence and its impact on the results are negligible.

3) This non-labelled control is actually something I neglected to mention in my previous review and is the caveat I mentioned above. I think it would be valuable to show at least once that acidity doesn't impact the background fluorescence of CTL in the CTV channel in order to demonstrate that the CTV fluorescence intensity between conditions can be reasonably compared as an indication of dye dilution/extent of proliferation - otherwise shifts in CTV fluorescence between conditions could always be attributable to an effect of acidity on cell autofluorescence rather than the extent of cell division.

Since we never reach a point in which CTV is close to being absent due to extensive cell proliferation, there should not be any impact/bias of cell autofluorescence on cell division number estimations given how strong CTV fluorescence is as compared to autofluorescence (100-1000x stronger when there is no division). Furthermore, whereas one might expect autofluorescence to be higher when cells are bigger in size leading to an underestimation in cell proliferation, low pH leads to smaller T cells, with lower amount of cell proliferation. We think that the most important control to make sure shifts in CTV fluorescence upon pH conditioning are indeed due to differences in cell division number is CTV fluorescence of unstimulated cells at pH7.4, pH7 and pH6.6: as shown in Fig. 1E, these conditions result in similar CTV fluorescence.

Given these points, my suggestion for how to represent the data is to plot CTV fluorescence of the treatment conditions as a fraction of the unstimulated/undivided condition. If you want it to look similar to the mean division number plots it could be plotted as $-\log_2$ of the fraction, with the axis label something along the lines of "estimated proliferation ($-\log_2$ CTV dilution)". In this way the data could be quantitatively plotted and compared in easy to read graphs without having to make any assumptions about where the division gates should be.

As previously mentioned, we believe that there may have been a misunderstanding in the way in which we are calculating “mean cell division number” as we are not using CTV MFI for the calculation. Although we agree that the methodology is not perfect, we think that it is still objective and allows one to estimate cell division number. Furthermore, in the experiments in which we used CTV, the differences we are identifying/claiming are rather black & white (we are not analyzing differences of 0.2-0.5 divisions, rather at least > 1): misinterpretation due to artifactual/flow issues should thus be minimized. This is even more supported by the fact that most of the results are confirmed later on in the manuscript with a more global, and mostly indisputable parameter, cell expansion.

We could indeed plot total CTV MFI on a log₂ axis. However, we think it would add another level of complexity to the manuscript. Furthermore, not plotting the final parameter of interest that was initially sought (number of division) while leading the readership to infer it artificially with a log₂ axis is not what we really want.

Nevertheless, we thank you for your thoughtful comments as we agree that we should have added explanations on the methodology we used. This has now been included in the methods section where we also highlight the limits of the methodology used. Since the terminology “Mean cell division number” appears to be problematic and potentially misleading because we are not giving the exact number of division but rather an estimation based on inference, we have changed the terminology to “Estimated division number” throughout the figures.

Minor points:

1. I'm confused about line 91-92. It says that pH 6.6 doesn't preferentially block CTL in a cell cycle stage but Appendix Fig S1D does show a higher proportion of cells in G₀/G₁ in low pH when cultured in IL2? Wouldn't this mean the cells are preferentially blocked in G₀/G₁?

We conclude to a general blockade because the G₀/G₁/S/M pattern is the same +/- IL-2 at pH6.6, and is almost identical to time 0: it looks that cells are “frozen” upon conditioning at pH6.6.

2. In line 260 - it mentions a drop in aspartate and references Fig 8A and EV4A but I can't see aspartate in either figure? Is there a graph missing or a figure reference that has been left out?

Thank you for noticing it. It has been corrected in the manuscript: aspartate was not detected in Figure 8A and EV4A. In contrast, we were able to detect it in the other ones, starting at Figure 8C. We added this sentence in the manuscript (line 269): “Whereas not detected in previous assays, it is worth noting that intracellular aspartate content was also lowered by acidity and was dependent upon extracellular glutamine levels (Fig. 8C).”

Referee #3:

Review - Acidity perturbs IL-2 responsiveness, mTORC1 and c-Myc in CD8+ T cells

General summary

This paper focuses on how acidic conditions affect the function of CD8 T-cells. It was found that cytotoxicity was reduced in response to lower affinity antigen/TCR binding, and that CD8 T cells cultured in a low pH had reduced cytokine production in response to stimulation and proliferation was reduced. There were changes in IL2 receptor protein abundance and IL2 binding. Signalling through the mTORC1 pathway was reduced at low pH; and levels of Myc protein were reduced. Reduction in mTORC1 signalling could not be attributed to known upstream regulators. Reduction in Myc protein involved changes in transcription and degradation. Metabolic changes including reduction in glutamine and glutamate were also seen in CTL exposed to low pH.

Major concerns

My main concern is relying on the Western blot data to compare activity through different signalling pathways. As different antibodies are being used for the different proteins, has anything been done to show that a similar reduction in a protein produces a similar drop in signal for the things you are comparing?

It is true that each antibody has his own binding characteristic. However, we believe there has been a misunderstanding: we are not comparing activity through different pathways by direct comparison of different protein levels (directly coming from the antibody signal) per pH/condition. Rather we are comparing the relative signal of one protein at a time (relative because the value is always normalized to a housekeeping protein or to the total protein of interest in case of a phosphorylation read-out) upon treatment (e.g., pH6.6) vs control (e.g., pH7.4). Therefore, we are comparing signals obtained from the same antibodies.

A drop estimated by Western blot does not necessarily reflect a drop to the same extent in reality: Western blot is indeed known as a semi-quantitative technique but it can lead to quantitative results when performed properly (Taylor et al., "A critical path to producing high quality, reproducible data from quantitative western blot experiments." Scientific reports 2022) and is still used as a gold standard to decipher cell signaling. Furthermore, the consequences of the drop we observed/inferred at low pH in mTORC1 and c-Myc were confirmed by GSEA.

Also, with the GSEA, I would have thought that the Jak-STAT pathway is a lot broader than the Myc pathway one, as it includes many different cytokines with various functions, so not all of the proteins in the Jak-STAT pathway are going to be involved in IL-2 signalling.

We acknowledge that the Jak-STAT pathway can be considered broader than the Myc pathway. Indeed, Myc is even included within the Jak-STAT gene set. However, it is important to note that the Gene Set Enrichment Analysis (GSEA) accounts for gene set size effects. The enrichment score (ES) in GSEA is calculated by walking down the ranked list of genes, increasing the score when a gene is in the gene set and decreasing it when it is not. The size of the gene set influences this walk and the resulting ES. To prevent size-related bias, GSEA normalizes the enrichment score to account for gene set size, resulting in the normalized enrichment score (NES). This normalization makes the enrichment scores more comparable across gene sets of different sizes, mitigating some of the biases introduced by gene set size.

Our results suggest that the Myc signaling gene set is significantly affected by low pH, whereas the Jak-STAT signaling pathway is arguably less affected than Myc signaling, as the enrichment of Jak-STAT elements in up-regulated genes is not statistically significant. However, as the reviewer mentioned, it seems like not all elements of the Jak-STAT pathway are affected by IL-2 signaling. Furthermore, given the results we show concerning their direct phosphorylation state and *Myc* transcription, the Jak-STAT pathway appears to be impaired - that is why we did not conclude in the manuscript to the absence of a pH-mediated impact on the Jak-STAT pathway based on GSEA.

The GSEA data are mostly presented here to confirm that the extent to which c-Myc levels are lowered at low pH (as measured by Western blot) has indeed consequences on c-Myc transcriptional targets.

Minor concerns

No additional minor concerns.

Non-essential suggestions

No additional non-essential suggestions.

Referee #4:

I am satisfied with the revised manuscript and have no further comments.

Referee #5:

The authors have addressed most of my concerns.

Dear Dr. Vuillefroy de Silly,

Congratulations on an excellent manuscript, I am very pleased to inform you that it has been accepted for publication in The EMBO Journal. Thank you very much for your comprehensive responses to the referee concerns and for addressing all editorial requests.

Your manuscript will now be processed for publication by EMBO Press. It will be copy edited and you will receive page proofs prior to publication. Please note that you will be contacted by Springer Nature Author Services to complete licensing and payment information.

If you have any questions, please do not hesitate to contact the Editorial Office. Thank you for your contribution to The EMBO Journal. Working with you has been a pleasure!

Best wishes,

Ioannis
